# THERE AND BACK AGAIN:
# ON THE RELATION BETWEEN NOISE AND IMAGE INVERSIONS IN DIFFUSION MODELS

**Łukasz Staniszewski**[1,2*] **Łukasz Kuciński**[3,4,5] **Kamil Deja**[1,2]
[1]Warsaw University of Technology  [2]IDEAS Research Institute  [3]University of Warsaw
[4]Institute of Mathematics, Polish Academy of Sciences  [5]IDEAS NCBR

## ABSTRACT

Diffusion Models achieve state-of-the-art performance in generating new samples but lack a low-dimensional latent space that encodes the data into editable features. Inversion-based methods address this by reversing the denoising trajectory, transferring images to their approximated starting noise. In this work, we thoroughly analyze this procedure and focus on the relation between the initial *noise*, the *generated samples*, and their corresponding *latent encodings* obtained through the DDIM inversion. First, we show that latents exhibit structural patterns in the form of less diverse noise predicted for smooth image areas (e.g., plain sky). Through a series of analyses, we trace this issue to the first inversion steps, which fail to provide accurate and diverse noise. Consequently, the DDIM inversion space is notably less manipulative than the original noise. We show that prior inversion methods do not fully resolve this issue, but our simple fix, where we replace the first DDIM Inversion steps with a forward diffusion process, successfully decorrelates latent encodings and enables higher quality editions and interpolations. The code is available at https://github.com/luk-st/taba.

## 1 INTRODUCTION

Diffusion-based probabilistic models (DMs), (Sohl-Dickstein et al., 2015), have achieved state-of-the-art results in many generative domains including image (Dhariwal & Nichol, 2021), speech (Popov et al., 2021), video (Ho et al., 2022), and music (Liu et al., 2021) synthesis. Nevertheless, one of the significant drawbacks that distinguishes diffusion-based approaches from other generative models like Variational Autoencoders (Kingma & Welling, 2014) is the lack of an implicit latent space that encodes the images into low-dimensional, interpretable, or editable representations.

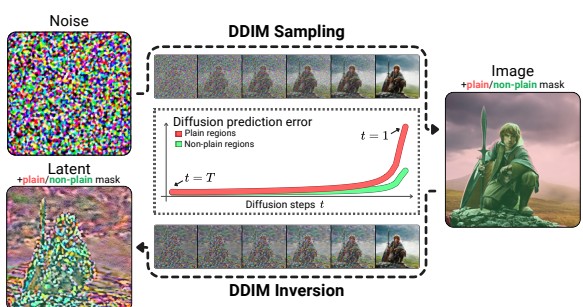

Figure 1: **DDIM inversion produces latent encodings that exhibit less diverse noise in the smooth image areas than in the non-plain one.** We attribute this problem to the errors of noise prediction in the first inversion steps.

To mitigate this issue, several works seek meaningful relations in the approximated starting noise used for generations. This method, known as an inversion technique, was introduced by Song et al. (2021) with Denoising Diffusion Implicit Models (DDIM), and led to the proliferation of works (Garibi et al., 2024; Mokady et al., 2023; Huberman-Spiegelglas et al., 2024; Samuel et al., 2025; Hong et al., 2024; Parmar et al., 2023). The core idea is to use the noise predicted by the Diffusion Model and add it to the image instead of subtracting it. Repeating this process effectively traces the backward diffusion trajectory, approximating the *noise* that could have generated the *image*. However, due to approximation errors and biases introduced by the trained model, discrepancies arise between the original noise and its reconstruction – *latent* representation.

---

*Corresponding author: `luks.staniszewski@gmail.com`

While recent works (Garibi et al., 2024; Mokady et al., 2023; Parmar et al., 2023; Huberman-Spiegelglas et al., 2024; Zheng et al., 2024) try to improve the inversion procedure from the perspective of tasks such as image reconstruction, editing, or interpolation, in this work, we focus on the inversion process itself and analyze the errors DDIM inversion introduces. To that end, we analyze the relation between sampled noise, generated images, and their inverted latent encodings. First, we review existing studies and conduct additional analyses demonstrating that the reverse DDIM technique produces latent representations with pixel correlations that deviate significantly from a Normal distribution. As presented in Figure 1, we experimentally show that this deviation manifests as lower diversity in latents, particularly in regions corresponding to smooth image surfaces. We further attribute this discrepancy to the noise approximations in the first few inversion steps.

To highlight the consequences of the observed divergence, we show that the DDIM-inversion-based latent space is less *manipulative* than the ground truth noise. This limitation is particularly noticeable in lower-quality image interpolations and less expressive edits, especially in smooth input image regions. Furthermore, we demonstrate that prior inversion methods, although designed to improve image reconstruction, fail to preserve the Gaussian properties of the latents. Based on our analyses, we propose a simple fix, replacing the first steps of the DDIM inversion process with a forward diffusion. Finally, we show that such an approach successfully decorrelates the resulting latents, mitigating the observed limitations, and can be combined with state-of-the-art image editing and style transfer engines to further improve their performance. Our main contributions can be summarized as follows:

- We show that DDIM latents deviate from the Gaussian distribution, mostly because of less diverse noise predictions for the plain image surfaces during the first inversion steps.

- We show that, consequently, the DDIM latents are less manipulative, leading to the lower quality of image interpolations and edits.

- We demonstrate that prior inversion methods do not address this issue and propose a simple and effective fix by substituting early inversion steps with a forward diffusion.

## 2 BACKGROUND AND RELATED WORK

**Denoising Diffusion Implicit Models.** The training of DMs consists of forward and backward diffusion processes, where, in the context of Denoising Diffusion Probabilistic Models (DDPMs, Ho et al. (2020)), the former one with training image $x_0$ and a variance schedule $\{\beta_t\}_{t=1}^T$, can be expressed as $x_t = \sqrt{\bar{\alpha}_t}x_0 + \sqrt{1 - \bar{\alpha}_t}\epsilon_t$, with $\alpha_t = 1 - \beta_t$, $\bar{\alpha}_t = \prod_{s=1}^t \alpha_s$, and $\epsilon_t \sim \mathcal{N}(0, \mathcal{I})$.

In the backward process, the noise is gradually removed starting from a random noise $x_T \sim \mathcal{N}(0, \mathcal{I})$ for $t = T \dots 1$, with intermediate steps defined as:

$$x_{t-1} = \sqrt{\bar{\alpha}_{t-1}} \cdot \underbrace{(x_t - \sqrt{1 - \bar{\alpha}_t} \cdot \epsilon_\theta^{(t)}(x_t, c))/\sqrt{\bar{\alpha}_t}}_{x_0 \text{ prediction}} + \underbrace{\sqrt{1 - \bar{\alpha}_{t-1} - \sigma_t^2} \cdot \epsilon_\theta^{(t)}(x_t, c)}_{\text{direction pointing to } x_t} + \sigma_t z_t, \quad (1)$$

where $\epsilon_\theta^{(t)}(x_t, c)$ is an output of a neural network (such as U-Net), and can be expressed as a combination of clean image ($x_0$) prediction, a direction pointing to previous denoising step ($x_t$), and a stochastic factor $\sigma_t z_t$, where $\sigma_t = \eta\sqrt{\beta_t(1 - \bar{\alpha}_{t-1})/(1 - \bar{\alpha}_t)}$ and $z_t \sim \mathcal{N}(0, \mathcal{I})$. In the standard DDPM model, the $\eta$ parameter is set to $\eta = 1$. However, changing it to $\eta = 0$ makes the whole process a deterministic Denoising Diffusion Implicit Model (DDIM, Song et al. (2021)), a class of DMs we target in this work.

One of the advantages of DDIM is that by making the process deterministic, we can encode images back to the noise space. The inversion can be obtained by rewriting Eq. (1) as

$$x_t = \sqrt{\alpha_t}x_{t-1} + (\sqrt{1 - \bar{\alpha}_t} - \sqrt{\alpha_t - \bar{\alpha}_t}) \cdot \epsilon_\theta^{(t)}(x_t, c). \quad (2)$$

However, due to circular dependency on $\epsilon_\theta^{(t)}(x_t, c)$, Dhariwal & Nichol (2021) propose to approximate this equation by assuming the local linearity between directions ($x_{t-1} \to x_t$) and ($x_t \to x_{t+1}$), so that the model's prediction in $t$-th inversion step can be approximated using $x_{t-1}$ as an input, i.e.,

$$\epsilon_\theta^{(t)}(x_t, c) \approx \epsilon_\theta^{(t)}(x_{t-1}, c). \quad (3)$$

While such approximation is often sufficient to obtain good reconstructions of images, it introduces the error dependent on the difference $(x_t - x_{t-1})$, which can be detrimental for models that sample images with a few diffusion steps or use the classifier-free guidance (Ho & Salimans, 2021; Mokady et al., 2023) for better prompt adherence. As a result, also noticed by recent works (Garibi et al., 2024; Parmar et al., 2023), latents resulting from DDIM inversion do not follow the definition of Gaussian noise because of the existing correlations. In this work, we empirically study this phenomenon and explain its origin. We discuss the relations between the following three variables: **Gaussian noise** ($\mathbf{x_T}$, an input to generate an image through a backward diffusion process), **image sample** ($\mathbf{x_0}$, the outcome of the diffusion model generation process), and **latent encoding** ($\mathbf{\hat{x}_T}$, the result of the DDIM inversion procedure as introduced in Eq. (2)).

**Image-to-noise inversion techniques.** The DDIM inversion, despite the noise approximation errors, forms the foundation for many applications, including inpainting (Zhang et al., 2023a), interpolation (Dhariwal & Nichol, 2021; Zheng et al., 2024), and edition (Su et al., 2022; Kim et al., 2022a; Hertz et al., 2022; Ceylan et al., 2023; Deja et al., 2023). Several works (Mokady et al., 2023; Garibi et al., 2024; Huberman-Spiegelglas et al., 2024; Samuel et al., 2025; Hong et al., 2024; Miyake et al., 2023; Han et al., 2024; Cho et al., 2024; Dong et al., 2023; Zhang et al., 2023b; Parmar et al., 2023; Tang et al., 2024; Wallace et al., 2023; Pan et al., 2023; Wang et al., 2024; Brack et al., 2024; Lin et al., 2024; Ju et al., 2024) aim to reverse the denoising process in text-to-image models, where prompt embeddings can strongly affect the final latent representation through Classifier-free-guidance (CFG) (Ho & Salimans, 2021). Null-text inversion (Mokady et al., 2023) extends the DDIM inversion with additional null-embedding optimization, reducing the image reconstruction error. Other techniques improve inversion for image editing through seeking embeddings (Miyake et al., 2023; Han et al., 2024; Dong et al., 2023) or leveraging DDIM latents (Cho et al., 2024) for guidance. On the other hand, some works leverage additional numerical methods (Samuel et al., 2025; Pan et al., 2023; Garibi et al., 2024) to minimize inversion error. In particular, Renoise (Garibi et al., 2024) iteratively improves the estimation of the next point along the diffusion trajectory by averaging multiple noise predictions, incorporating an additional patch-level regularization term that penalizes correlations between pixel pairs to ensure the editability of the latents. Huberman-Spiegelglas et al. (2024) followed by Brack et al. (2024) propose inversion methods for DDPMs, enabling the creation of various image edition results via inversion. Finally, to reduce the discrepancy between DDIM latents and Gaussian noises, Parmar et al. (2023) propose to additionally regularize final DDIM Inversion outputs for better image editing, Lin et al. (2024) introduce an alternative noise scheduler to improve inversion stability, while Hong et al. (2024) propose an exact inversion procedure for higher-order DPM-Solvers, solving the optimization problem at each step.

## 3 ANALYSIS

In our experiments, we employ six different diffusion models, which we compare in Table 1. For both generation and inversion processes, we use the DDIM sampler with, unless stated otherwise, $T = 100$ steps. We provide more details on the number of diffusion steps in Appendix N.1.

| Model | Diffusion Space | Resolution Image | Latent | Training Dataset | Cond? | Arch |
|---|---|---|---|---|---|---|
| ADM-32 | Pixel | 32x32 | - | CIFAR-10 | ✗ | U-Net |
| ADM-64 | Pixel | 64x64 | - | ImageNet | ✗ | U-Net |
| ADM-256 | Pixel | 256x256 | - | ImageNet | ✗ | U-Net |
| LDM | Latent | 256x256 | 3x64x64 | CelebA | ✗ | U-Net |
| DiT | Latent | 256x256 | 4x32x32 | ImageNet | ✓ | DiT |
| IF | Pixel | 64x64 | - | LAION-A | ✓ | U-Net |
| SDXL | Latent | 1024x1024 | 4x128x128 | - | ✓ | U-Net |

Table 1: **Overview of diffusion models used for our experiments.** We study both unconditioned and conditioned models, operating in pixel and latent spaces. More details on models are provided in Appendix J.

### 3.1 LATENTS VS. NOISE

The inversion process provides the foundation for practical methods in many applications, with the underlying assumption that by encoding the *image* back with a denoising model, we can obtain the original *noise* that can be used for reconstruction. However, this assumption is not always fulfilled, which leads to the question:

> **Research Question 1:** Are there any differences between sampled Gaussian noise and latents calculated through the DDIM inversion?

**Prior work and findings.** This question relates to several observations from the existing literature, which highlight that outputs of the DDIM inversion differ from the standard Gaussian noise (Parmar et al., 2023; Garibi et al., 2024) and that the difference can be attributed to the approximation error (Hong et al., 2024; Wallace et al., 2023). While these works notice the divergence between noise and latent encodings, they do not validate them or study the source of this issue.

**Experiments.** First, we consolidate existing observations on the presence of correlations in the latent encodings $(\hat{\mathbf{x}}_\mathbf{T})$ and validate them by running an initial experiment that compares latents to images $(\mathbf{x_0})$ and noises $(\mathbf{x_T})$ across diverse diffusion architectures. In Table 2, we calculate a mean of top-20 Pearson correlation coefficients (their absolute values) inside $C \times 8 \times 8$ pixel patches, where $C$ is the number of channels (pixel RGB colors or latent space dimensions for latent models). The results confirm that latent representations have significantly more correlated pixels than the noise. In Fig. 2, we show how the measured correlations visually manifest themselves in the latents. For pixel models such as Deepfloyd IF, we observe clear groups of correlated pixels as presented in Fig. 2a. For latent diffusion models, we can highlight the inversion error by plotting the difference between the latent and the noise, as presented in Fig. 2b. This property also holds for LDMs with a 4-channel latent space, with the use of PCA (Fig. 2c).

| Model | Noise ($\mathbf{x_T}$) | Latent ($\hat{\mathbf{x}}_\mathbf{T}$) | Sample ($\mathbf{x_0}$) |
|---|---|---|---|
| ADM-32 | $0.039_{\pm.003}$ | $0.382_{\pm.010}$ | $0.964_{\pm.022}$ |
| ADM-64 | $0.039_{\pm.003}$ | $0.126_{\pm.008}$ | $0.925_{\pm.021}$ |
| ADM-256 | $0.039_{\pm.003}$ | $0.161_{\pm.013}$ | $0.960_{\pm.008}$ |
| IF | $0.039_{\pm.003}$ | $0.498_{\pm.025}$ | $0.936_{\pm.019}$ |
| LDM | $0.039_{\pm.003}$ | $0.045_{\pm.014}$ | $0.645_{\pm.099}$ |
| DiT | $0.041_{\pm.003}$ | $0.103_{\pm.021}$ | $0.748_{\pm.064}$ |
| SDXL | $0.036_{\pm.002}$ | $0.155_{\pm.044}$ | $0.637_{\pm.064}$ |

Table 2: **Mean of top-20 Pearson correlation coefficients inside $8 \times 8$ patches for random Gaussian noises, latent encodings, and generations.** DDIM Latents are much more correlated than noises.

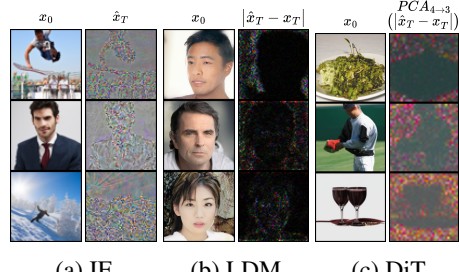

(a) IF    (b) LDM    (c) DiT

Figure 2: **Latent encodings exhibit image patterns.** For small pixel-space models (a), we observe correlations directly in the inversion results. For larger models (e.g., LDMs), the same patterns can be observed in the absolute errors between the latent and noise (b). This observation also holds for LDM models operating on 4-channels, where we use PCA for visualization (c).

**Conclusion.** Our initial experiments numerically validate observations from recent studies and demonstrate that latent representations computed using the DDIM inversion deviate from the expected characteristics of independent Gaussian noise. Specifically, both visual analysis and quantitative evaluations reveal significant correlations between the neighboring pixels.

## 3.2 LOCATION OF LATENT ENCODINGS SPACE

To delve deeper, we first propose to empirically analyze the nature of this issue, posing a question:

**Research Question 2:** How do DDIM inversion latents differ from the Gaussian noise?

**Experiments.** To answer this question, we first geometrically investigate the location of the latents with respect to the generation trajectory. We analyze the distance between the following steps $\{x_t\}_{t=T\dots1}$ of the backward diffusion process and intermediate points on the linear interpolation path between the noise and the DDIM latent. The results of this experiment are presented in Fig. 3, where each pixel, with coordinates $(t, \lambda)$, is colored according to the $l_2$ distance between the intermediate trajectory step $x_t$ and the corresponding interpolation step. This distance can be expressed as $\|(1-\lambda)\mathbf{x_T}+\lambda\hat{\mathbf{x}}_\mathbf{T}-x_t\|_2$. For better clarity, we normalize the distances column-wise.

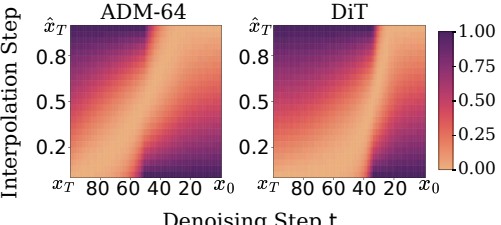

Figure 3: $l_2$-**distances between intermediate denoising steps** $x_t$ **and points on the linear interpolation path from noise $\mathbf{x_T}$ to the inverted latents $\hat{\mathbf{x}}_\mathbf{T}$.** The consecutive intermediate generation steps along the sampling trajectory consequently approach the latent.

We observe that, while moving from the initial noise ($\mathbf{x_T}$) towards the final sample ($\mathbf{x_0}$), the intermediate steps $x_t$ approach the DDIM inversion latent ($\hat{\mathbf{x}}_\mathbf{T}$), while after the transition point around 50-70% (timesteps 50-30) of the generative trajectory, the distance to the latent becomes lower than the distance to the starting noise. **This observation reveals that latents retain some characteristics of the original samples and contain information about the source generation.**

Similar observation can be made on the basis of visualizations in Fig. 2, where we can distinguish the coarse shape of the original objects in the latents. In particular, while the pixels associated with the objects have high diversity, the areas related to the background are much smoother. This leads to the hypothesis that *the most significant difference between the initial noises and latent encodings is the limited variance in the areas corresponding to the background of the generated images.* To validate it, we compare the properties of the latents between plain and non-plain areas in the image. We determine binary masks $\mathcal{M}_p$ by calculating the absolute difference between neighboring pixels. Pixels where this local variation falls below a fixed threshold ($\tau = 0.025$) across all channels are classified as

| Model | Absolute Error | | Standard Deviation | |
|---|---|---|---|---|
| | Plain | Non-plain | Plain | Non-plain |
| Noise (def.) | – | – | 1.0 | 1.0 |
| ADM-32 | 0.49 | 0.43 | 0.34 | 0.46 |
| ADM-64 | 0.22 | 0.15 | 0.49 | 0.64 |
| ADM-256 | 0.39 | 0.29 | 0.53 | 0.66 |
| IF | 0.56 | 0.40 | 0.46 | 0.72 |
| LDM | 0.13 | 0.03 | 0.45 | 0.59 |
| DiT | 0.12 | 0.06 | 0.43 | 0.54 |
| SDXL | 0.30 | 0.26 | 0.87 | 0.96 |

Table 3: **Average *per-pixel* absolute error (between noise and latent) and standard deviation of the latents' pixels corresponding to the plain and non-plain image areas.** The error for plain pixels, where latents are less diverse, is higher than for other regions.

plain regions, effectively isolating low-texture areas (see Appendix H for more details). This procedure selects areas, such as sky, sea, plain backgrounds, or surfaces. In Table 3, we show that the error between the starting noise $\mathbf{x_T}$ and the latent $\hat{\mathbf{x}}_\mathbf{T}$ resulting from the DDIM inversion is higher for pixels corresponding to the plain areas. Across the models, this trend goes along with a decrease in the standard deviation of the latents' pixels related to those regions. It suggests that DDIM inversion struggles with reversing the plain image areas, bringing them to mean (0) and reducing their diversity. Additionally, in Appendix P, we present that correlations and reduced latent diversity for plain image regions can be similarly observed within Flow Matching (Lipman et al., 2023) models.

**Conclusion.** Latent encodings resulting from the DDIM Inversion deviate from the Gaussian noise towards zero values. This is especially true for parts of the latents corresponding to the plain image surfaces. This observation reveals that inversion outputs retain some characteristics of the original input samples and contain information about the source generation.

### 3.3 ORIGIN OF THE DIVERGENCE

Given the observation from the previous section, we now investigate the source of the correlations occurring in latent encodings, posing the question:

> **Research Question 3:** What causes the spatial correlations observed in DDIM latents?

**Experiments.** We recall that the DDIM Inversion error can be attributed to the approximation of the diffusion model's output at step $t \in 1 \ldots T$ with the output from step $t - 1$ (see Eq. (3)). Hence, we can define the inversion approximation error for step $t$ as the difference between DM's output for the current and previous inversion timesteps $t$ and $t - 1$ as:

$$\xi(t) = |\underbrace{\epsilon_\theta^{(t)}(x_{t-1}, c)}_{\mathcal{E}_t^I} - \underbrace{\epsilon_\theta^{(t)}(x_t, c)}_{\mathcal{E}_t^S}|, \tag{4}$$

where $\mathcal{E}_t^S$ is the true model prediction at step $t$, and $\mathcal{E}_t^I$ is the Dhariwal & Nichol (2021) approximation using the previous step as an input. Based on the observations from the previous section, we propose to investigate how the approximation error $\xi(t)$ varies across pixels in plain and non-plain image areas throughout the inversion process. To that end, we average the approximation errors for 4000 images for each of the $T = 50$ diffusion timesteps. To measure the error solely for the analyzed step $t$, we start the inversion procedure from the exact latent from step $(t - 1)$ (cache from the sampling path). We split the latent pixels into plain and non-plain areas based on the masks calculated for clean images. More details on this setup can be found in Appendix I.

In Fig. 4a, we present the visualization of calculated differences for each inversion step. There is a significant difference in prediction errors between plain and non-plain areas, especially in the initial steps of the inversion process. Notably, for pixels associated with plain image areas, the error predominantly accumulates within the first 10% of the inversion steps. Additionally, in Fig. 4b, we present that this error discrepancy is strongly connected to a decrease in the diversity of diffusion models' predictions. More precisely, we calculate a ratio of the predictions' standard deviations between the sampling and inversion processes for the associated timesteps. We show that, for plain image regions, there is a significant decrease in the fraction of predictions' variance preserved during the first inversion steps. Those observations can be related to recent works (Lee et al., 2023; Lin et al., 2024) analyzing why numerical solvers incur significant errors during the earliest diffusion steps (as $t \rightarrow 0$). Specifically, Lee et al. (2023) trace the error to the $1/t$ curvature blow-up of the reverse-time ODE trajectory, whereas Lin et al. (2024) attribute the predominance of early DDIM inversion approximations to a singularity arising from commonly used noise schedules. Our experiments extend those studies by showing that the error can be mainly attributed to the plain regions in the original images.

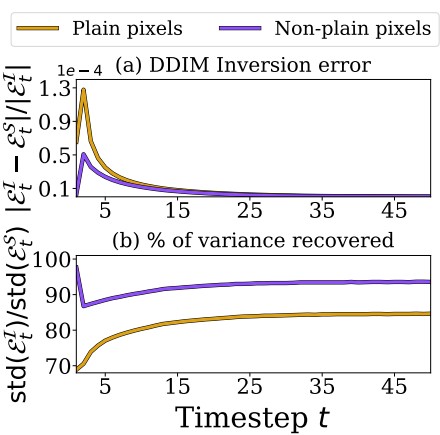

Figure 4: **Discrepancy of the DDIM noise predictions for plain and non-plain image pixels.** We show that, in the first inversion steps, the approximations are significantly (a) more erroneous and (b) less diverse for plain regions than for the rest of the image.

**Conclusion.** Early approximations during the DDIM Inversion procedure result in unequally distributed errors across pixels related to the plain and non-plain image areas, which are the origin of the structural patterns and correlations in the latents.

## 4    CONSEQUENCES OF THE DIVERGENCE AND HOW TO MITIGATE THEM?

After identifying the differences between the noises and latents, and highlighting the origin of this phenomenon, we finally pose the last question:

> **Research Question 4:** What are the practical consequences of the divergence between noises and inverse DDIM latents, and how can they be mitigated?

Our findings in Section 3.3 indicate that the initial inversion steps predominantly contribute to the divergence between DDIM latent variables and Gaussian noise, in the form of insufficiently diverse approximations of diffusion model predictions. Therefore, as a simple fix to this issue, we propose to replace the first inversion steps with random noise, as in a forward diffusion process. The rationale behind this decision is twofold:

- Substituting initial steps with Gaussian noise allows us to recover the noise variance exactly when the DDIM inversion fails to do so.
- Recent studies (Deja et al., 2022; Liu et al., 2025; Li & Chen, 2024; Fesl et al., 2025) have shown that final steps of the backward diffusion do not contribute additional generative information, instead functioning as a data-agnostic denoising process. Therefore, the exact inversion of those steps is less important for accurate reconstruction.

The proposed inversion step is therefore defined as follows (see Appendix F for pseudocode):

$$x_t = \begin{cases} \sqrt{\bar{\alpha}_t}x_0 + \sqrt{1 - \bar{\alpha}_t}\epsilon, & \text{if } t \leq t' \quad \text{(forward diffusion)} \\ \sqrt{\alpha_t}x_{t-1} + (\sqrt{1 - \bar{\alpha}_t} - \sqrt{\alpha_t - \bar{\alpha}_t}) \cdot \epsilon_\theta^{(t)}(x_{t-1}, c), & \text{if } t > t' \quad \text{(DDIM inversion)} \end{cases} \quad (5)$$

Before moving to practical applications, we first evaluate the effectiveness of this approach using $N = 10000$ images generated with $T = 50$ steps using DiT and IF models. For such generations, we first noise them with a forward diffusion to the intermediate step $t'$, followed by the DDIM inversion for $T - t'$ steps, ending up with an approximation of the initial noise. In Table 4, we show that

| Inversion steps replaced by forward (%T) | DiT | | | IF | | |
|---|---|---|---|---|---|---|
| | Pixel Corr. | Reconstruction Absolute Error | KL Div. $\times 10^{-3}$ | Pixel Corr. | Reconstruction Absolute Error | KL Div. $\times 10^{-3}$ |
| Noise $\mathbf{x_T}$ | 0.04 | 0.00 | 0.20 | 0.05 | 0.00 | 0.40 |
| DDIM latent $\hat{\mathbf{x}}_T$ | 0.16 | 0.05 | 11.57 | 0.64 | 0.07 | 608.25 |
| 1 (2%) | 0.04 | 0.05 | 0.29 | 0.06 | 0.07 | 0.98 |
| 1…2 (4%) | 0.04 | 0.07 | 0.25 | 0.05 | 0.07 | 0.48 |
| 1…5 (10%) | 0.04 | 0.12 | 0.49 | 0.05 | 0.08 | 0.42 |
| 1…10 (20%) | 0.04 | 0.15 | 0.45 | 0.05 | 0.10 | 0.40 |

| Model | Region | Different prompt generations from: | | |
|---|---|---|---|---|
| | | Noise (baseline) | Latent DDIM Inv. | Latent w/ Forward 4% |
| IF | Plain | 17.90 | 14.92 (+16.7%) | 17.42 (+2.7%) |
| | Non-plain | 18.60 | 17.35 (+ 6.7%) | 18.16 (+2.4%) |
| DiT | Plain | 13.64 | 11.95 (+12.4%) | 13.27 (+2.7%) |
| | Non-plain | 16.49 | 15.34 (+ 7.0%) | 16.18 (+1.9%) |

Table 4: **Structures can be removed from DDIM latents by replacing inversion steps with forward diffusion.** Using forward diffusion instead of the first $4\%$ of inversion steps brings the resulting latents closer to Gaussian noise without a major degradation in the image reconstruction.

Table 5: **PNG bit-rate (bits / pixel) after saving only the masked pixels.** Higher compression (lower bpp) means less local variability in the pixel stream. Values in parentheses are the percentage change with respect to the noise baseline.

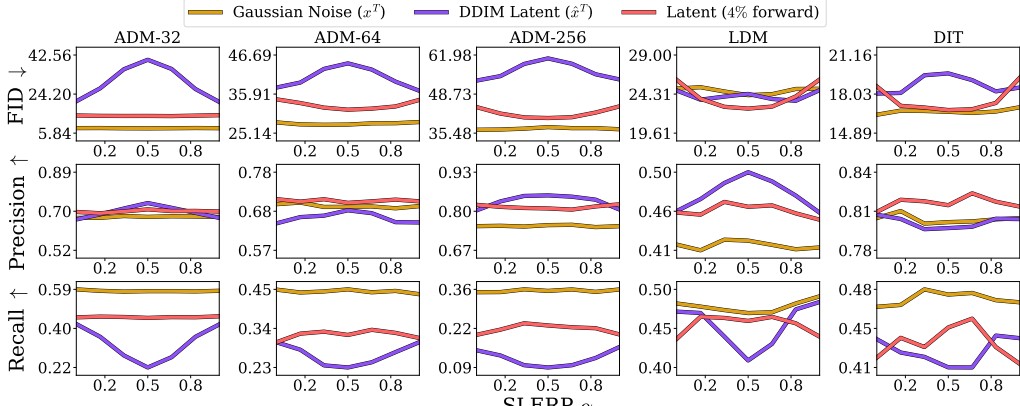

Figure 5: **The quality and diversity of images generated from interpolations of latents $\hat{\mathbf{x}}_T$ deteriorate along the path, as indicated by, accordingly, the FID peak and Recall decrease.** In contrast, the quality of generations from noise $\mathbf{x_T}$ interpolations remains stable. Our simple fix, which is replacing $4\%$ of the first inversion steps with forward diffusion, alleviates this issue.

by replacing just $4\%$ of the inversion steps, we can completely remove correlations in the latents, up to the level of random Gaussian noise. This replacement percentage allows us to navigate the trade-off between reconstruction fidelity and latent manipulability. We observe that this trade-off is highly favorable: replacing the first few steps ($t' \leq 4\%$) restores the Gaussian properties required for diverse editing, while maintaining a reconstruction error that remains within the perceptual noise floor (see Appendix N.2 for detailed analysis). To further evaluate this effect, in Table 5 we measure the sizes of the different parts of images (plain vs non-plain) after saving them with PNG lossless compression. Compression is most effective on parts related to plain images generated from DDIM latents, what inclines a low diversity in their values. At the same time, replacing only $4\%$ of inversion steps significantly reduces this issue. In the following sections, we further demonstrate that the divergence between noise and DDIM latents has significant practical consequences, and show that our approach for removing correlations consistently mitigates them.

## 4.1 INTERPOLATION QUALITY

We start with the task of image interpolation, where the goal, for two given images, is to generate a sequence of semantically meaningful intermediary frames. Numerous methods (Dhariwal & Nichol, 2021; Song et al., 2021; Samuel et al., 2023; Zheng et al., 2024; Zhang et al., 2024b; Bodin et al., 2025) employ Diffusion Models with DDIM inversion technique to obtain latent encodings from input images, interpolate them, and regenerate back to images. Song et al. (2021) propose to use the spherical linear interpolation (SLERP, Shoemake (1985)), that, for two objects $x$ and $y$, with a coefficient $\lambda \in [0; 1]$, is defined as $z^{(\lambda)} = \frac{\sin(1-\lambda)\theta}{\sin\theta}x + \frac{\sin\lambda\theta}{\sin\theta}y$, where $\theta = \arccos\left((x \cdot y)/(\|x\|\|y\|)\right)$.

In our experiment, we compare the quality of interpolations in the noise and latent spaces. To this end, we sample $N = 20$k noises, use DDIM with $T = 50$ diffusion steps to generate images, and invert those images back into their latents. Next, we randomly assign pairs, which we interpolate with SLERP for $\lambda \in \{0, \frac{1}{6}, \frac{1}{3}, \frac{1}{2}, \frac{2}{3}, \frac{5}{6}, 1\}$ and denoise. In Fig. 5, we show that, by calculating FID-10k, generations starting from interpolations between random noises (in orange) preserve consistent

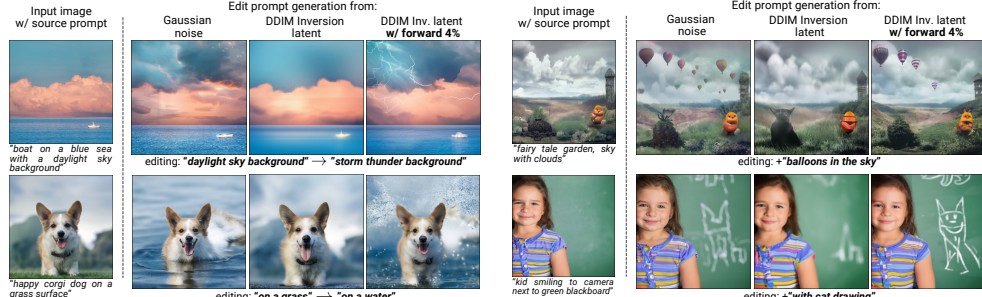

Figure 6: **Image editing by reversing the original image with a source prompt and reconstructing it with a target one.** DDIM Inversion produces less diverse image changes when the manipulation is related to plain regions in source images, contrary to when using ground-truth Gaussian noise. Replacing the first inversion steps with forward diffusion leads to more editable latents.

quality along the entire path, meaning that all interpolated images fall into the real images manifold. In contrast, this property collapses for the setting with DDIM latents as inputs (in purple). In the following rows, we show that worse interpolation results for latents stem from the decline in the variety of produced generations, as indicated by a lower recall (Kynkäänniemi et al., 2019), especially when getting closer to the middle of the interpolation path. Nevertheless, as presented in Fig. 5, using our fix for the first 4% of steps mitigates this issue, enabling higher-quality interpolations with better diversity of the intermediate points. In Appendix O.1, we present qualitatively that the proposed fix leads to more diverse interpolations, especially in the image background.

## 4.2 DIVERSITY AND QUALITY OF IMAGE EDITION

Apart from image interpolations, text-to-image diffusion models with DDIM inversion are often used for text-based edition, where a source image is first inverted and then reconstructed with a different target prompt (Hertz et al., 2022; Mokady et al., 2023; Garibi et al., 2024; Huberman-Spiegelglas et al., 2024). However, knowing that DDIM latents are less diverse in plain areas, we hypothesize that using them as a starting point might reduce the diversity and quality of the edited samples. To evaluate this, we use DiT (Peebles & Xie, 2023) and IF (StabilityAI, 2023) as conditional DMs with $T = 50$ diffusion steps. For each model, we construct two sets of 1280 randomly selected (1) source prompts $P_S$, used during the generation and inversion, and (2) target prompts $P_T$, used for edition. Using source prompts $P_S$, we generate images $I_S$ from Gaussian noise $\mathbf{x_T}$ and invert them back into the latents $\hat{\mathbf{x}}_\mathbf{T}$. Next, we regenerate images $\hat{I}_T$ from the latents, changing the conditioning to the target prompts $P_T$. We compare the edits with ground truth targets $I_T$ generated from the original noise $\mathbf{x_T}$ with $P_T$. In Fig. 6, we present the drawback of leveraging latents as starting points for the denoising, where the structures for $I_S$ images' backgrounds limit editing performance in $\hat{I}_T$ target images.

In Table 6, we quantitatively measure this effect. First, we calculate the diversity of target generations $(I_T, \hat{I}_T)$ against source images $(I_S)$. We use DreamSim distance (Fu et al., 2023), LPIPS (Zhang et al., 2018), SSIM (Wang et al., 2004), and cosine similarity of DINO features (Darcet et al., 2024) to measure the distance between two sets of generations. The experiment shows that edits resulting from latent encodings $\hat{I}_T$ are characterized by higher similarity (SSIM, DINO) and lower diver-

| Property | Metric | DiT | | Deepfloyd IF | |
|---|---|---|---|---|---|
| | | Noise $\mathbf{x_T}$ | Latent $\hat{\mathbf{x}}_\mathbf{T}$ | Noise $\mathbf{x_T}$ | Latent $\hat{\mathbf{x}}_\mathbf{T}$ |
| Diversity against $I_S$ | DreamSim ↑ | $0.71_{\pm0.12}$ | $0.68_{\pm0.13}$ | $0.67_{\pm0.10}$ | $0.61_{\pm0.11}$ |
| | LPIPS ↑ | $0.59_{\pm0.12}$ | $0.56_{\pm0.12}$ | $0.38_{\pm0.11}$ | $0.33_{\pm0.11}$ |
| | SSIM ↓ | $0.23_{\pm0.13}$ | $0.26_{\pm0.14}$ | $0.34_{\pm0.15}$ | $0.41_{\pm0.15}$ |
| | DINO ↓ | 0.17 | 0.22 | 0.34 | 0.42 |
| Text alignment | CLIP-T ($P_S$) ↓ | 0.465 | 0.480 | 0.273 | 0.353 |
| | CLIP-T ($P_T$) ↑ | 0.681 | 0.662 | 0.649 | 0.614 |
| | Directional ↑ | 0.570 | 0.541 | 0.776 | 0.676 |

Table 6: **Diversity of editions (generations from noise $\mathbf{x_T}$ and latents $\hat{\mathbf{x}}_\mathbf{T}$, conditioned on target prompt) in relation to source images $I_S$ and their alignment with source, target, and directional prompts.** The arrows (↑/↓) indicate greater generation diversity and higher text alignment to the target prompt.

sity (DreamSim, LPIPS) relative to starting images $I_S$ than the one resulting from noises $I_T$. At the same time, in the bottom rows of Table 6, we show that the correlations occurring in the latent encodings induce lower performance in text-alignment to target prompts $P_T$, which we measure by calculating cosine similarity between text embeddings and resulting image embeddings, both obtained with the CLIP (Radford et al., 2021) encoder. Additionally, to better assess image editing quality, we use directional CLIP similarity (Gal et al., 2022).

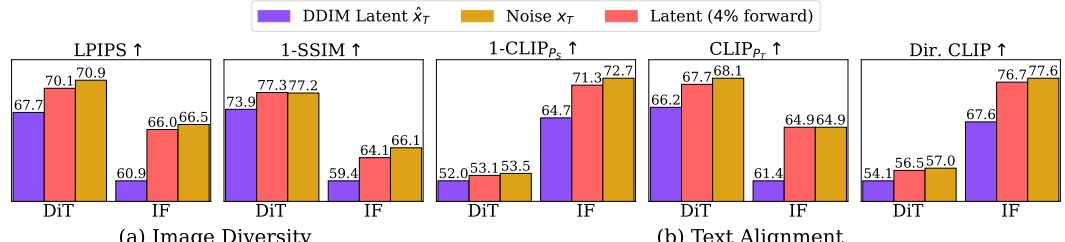

Figure 7: **Replacing first DDIM inversion steps with forward diffusion improves editions' pixel diversity (a) and prompt-alignment (b).** For IF and DiT models, diversity of generations can be improved by leveraging the forward diffusion process in the first inversion steps, and denoising resulting latents with a different prompt. Additionally, we observe a boost in generations' alignment to the target prompts, as indicated by the larger Directional CLIP Similarity, larger CLIP-T value for $P_T$, and the smaller one for $P_S$. We present details of the experiment in Appendix K.

We further evaluate how replacing the first inversion steps with the forward diffusion affects the diversity and text alignment of generated outputs. As shown in Fig. 7a, swapping the first 4% of DDIM inversion steps with forward diffusion improves the diversity of images generated from latents almost to the level of the samples from original noise. At the same time, as presented in Fig. 7b, replacing the first inversion steps leads to a significant decrease in generations' alignment to the source prompt ($P_S$) and an increase in similarity to the target one ($P_T$). This can also be observed in visualizations (Fig. 6). Moreover, because of the random noise added as the initial inversion steps, as presented in Appendix O.3, our approach enables stochastic image editing, producing diverse manipulations of the same image.

## 4.3 DO EXISTING INVERSION METHODS FIX THE CORRELATIONS?

So far, we have demonstrated the issues of the classical DDIM inversion method. In this section, we investigate whether novel inversion methods introduced in prior work resolve the issue of selectively reduced diversity in latents. We employ the Stable Diffusion XL (Podell et al., 2024) model and, using 2000 generations from COCO-30K prompts, measure the normality and editability of the resulting inversions, as well as image reconstruction performance. For fair comparison, we use the same number of NFEs. Results in Table 7 indicate that methods based on predicted noise regularization, such as Pix2Pix-Zero (Parmar et al., 2023) and ReNoise (Garibi et al., 2024), while slightly improving the latents' quality, do not offer significantly better editability, while being two times slower than DDIM. On the other hand, replacing DDIM approximation (Eq. (3)) with reverse DPMSolver (Lu et al., 2022) leads to decorrelated latents at the cost of high image reconstruction error. We show that our fix offers the best editability of latents with minimal reconstruction loss, all at the lowest computational cost. To be more precise, thanks to the fact that selected inversions steps replaced with the randomly sampled noise are the least important in terms of accurate reconstruction, we can observe no increase in the reconstruction error when replacing 2% of forward steps (up to the 2nd decimal point of MAE), while for 4% of steps the additional error is around 1% of pixel deviations - a value below the threshold usually employed by adversarial attacks as being not noticeable by a human eye (Madry et al., 2017). This replacement percentage allows us to navigate the trade-off between reconstruction fidelity and latent manipulability.

| Prior | NFE | Normality | | Image Reconstruction | | | CLIP Text Alignment | | | Inv. time |
| | | Corr. ↓ | KL ↓ | MAE ↓ | LPIPS ↓ | PSNR ↑ | Source ↓ | Target ↑ | Direct. ↑ | [s/image] ↓ |
|---|---|---|---|---|---|---|---|---|---|---|
| Gaussian Noise | — | $0.08_{\pm.01}$ | 0.10 | – | – | – | $31.88_{\pm11.66}$ | $73.34_{\pm9.65}$ | $80.62_{\pm16.96}$ | – |
| DDIM Inv. | 50 | $0.16_{\pm.02}$ | 0.89 | **0.03** | $\underline{0.10}_{\pm.05}$ | **27.58** | $34.99_{\pm11.36}$ | $69.58_{\pm10.12}$ | $75.59_{\pm17.86}$ | $7.17_{\pm.01}$ |
| Pix2Pix-Zero | 50 | $0.15_{\pm.02}$ | 0.85 | **0.03** | $\underline{0.10}_{\pm.05}$ | $\underline{27.35}$ | $34.86_{\pm11.39}$ | $69.73_{\pm10.12}$ | $75.83_{\pm17.87}$ | $22.07_{\pm1.84}$ |
| ReNoise (T=50, K=1) | 50 | $0.14_{\pm.02}$ | 0.73 | $\underline{0.04}$ | $0.09_{\pm.05}$ | 25.64 | $34.89_{\pm11.46}$ | $69.87_{\pm10.07}$ | $76.47_{\pm18.12}$ | $19.86_{\pm.22}$ |
| ReNoise (T=25, K=2) | 50 | $0.14_{\pm.02}$ | 0.59 | $\underline{0.04}$ | $0.09_{\pm.04}$ | 24.81 | $35.21_{\pm11.61}$ | $69.68_{\pm10.09}$ | $76.17_{\pm18.15}$ | $15.36_{\pm.51}$ |
| ReNoise (T=17, K=3) | 51 | $\underline{0.13}_{\pm.02}$ | 0.47 | 0.06 | $0.13_{\pm.05}$ | 22.35 | $35.79_{\pm11.65}$ | $69.04_{\pm9.98}$ | $75.20_{\pm17.96}$ | $14.31_{\pm.49}$ |
| DPMSolver-1 (T=50) | 50 | $0.09_{\pm.01}$ | 0.50 | 0.06 | $0.30_{\pm.10}$ | 22.55 | $34.81_{\pm11.40}$ | $70.26_{\pm10.42}$ | $74.76_{\pm18.02}$ | $7.06_{\pm.00}$ |
| DPMSolver-2 (T=25) | 50 | $0.09_{\pm.01}$ | **0.26** | 0.06 | $0.14_{\pm.07}$ | 24.76 | $34.69_{\pm11.55}$ | $\underline{71.24}_{\pm9.91}$ | $76.17_{\pm18.10}$ | $7.06_{\pm.00}$ |
| **Ours (forward 2%)** | 49 | $0.14_{\pm.02}$ | 0.71 | **0.03** | $\underline{0.10}_{\pm.05}$ | 27.12 | $\underline{34.32}_{\pm11.49}$ | $70.21_{\pm10.13}$ | $\underline{76.76}_{\pm17.79}$ | $\underline{7.00}_{\pm.00}$ |
| **Ours (forward 4%)** | 48 | $0.09_{\pm.01}$ | $\underline{0.38}$ | $\underline{0.04}$ | $0.14_{\pm.04}$ | 25.68 | $33.62_{\pm11.63}$ | $72.17_{\pm9.94}$ | $78.91_{\pm17.49}$ | $6.86_{\pm.01}$ |

Table 7: **Evaluation of inversion methods across multiple metrics: latents normality, image reconstruction, prompt alignment, and speed.** DDIM with the proposed fix offers a good trade-off between latent editability and image reconstruction, while increasing the inversion speed.

## 4.4 IMPROVING STATE-OF-THE-ART EDITING ENGINES WITH OUR FIX

Finally, we evaluate the possibility of combining our simple fix with existing methods designed for real image manipulation. We adapt StyleAligned (Hertz et al., 2024), the state-of-the-art method for transferring a style from a reference image to new generations, and MasaCtrl (Cao et al., 2023), a complex editing engine for text-based real image editing. As these methods employ Naïve DDIM Inversion to find starting noise for input images, we can directly apply our simple fix to those techniques, without changing their generation procedure.

| Inversion Method | CLIP Prompt Alignment ↑ | Set Consistency (DINO) ↑ | Set Consistency (CSD) ↑ | Style Similarity (DINO) ↑ | Style Similarity (CSD) ↑ |
|---|---|---|---|---|---|
| Naïve DDIM | **0.795** | **0.476** | 0.552 | 0.505 | 0.690 |
| Ours (forward 4%) | **0.795** | 0.471 | **0.554** | **0.510** | **0.697** |

Table 8: **Style transfer from reference image with StyleAligned (Hertz et al., 2024) incorporating Naïve DDIM Inversion and version with our fix.** Our fix improves similarity to input style.

We evaluate style transfer by measuring generations' alignment to the prompt, set consistency (pairwise cosine similarities of DINO (Darcet et al., 2024) and CSD (Somepalli et al., 2025) embeddings), and style consistency with the reference image (cosine similarities of DINO and CSD embeddings). The Table 8 compares the performance of vanilla StyleAligned and the version with our fix in style transfer from StyleDrop (Sohn et al., 2023) images. As presented, our approach improves the alignment with the target style. Additionally, in Fig. 8, we present a qualitative comparison of both inversion algorithms when combined with StyleAligned (1) for style transfer and MasaCtrl (2) for real image editing. More examples can be found in Appendices O.4 and O.5.

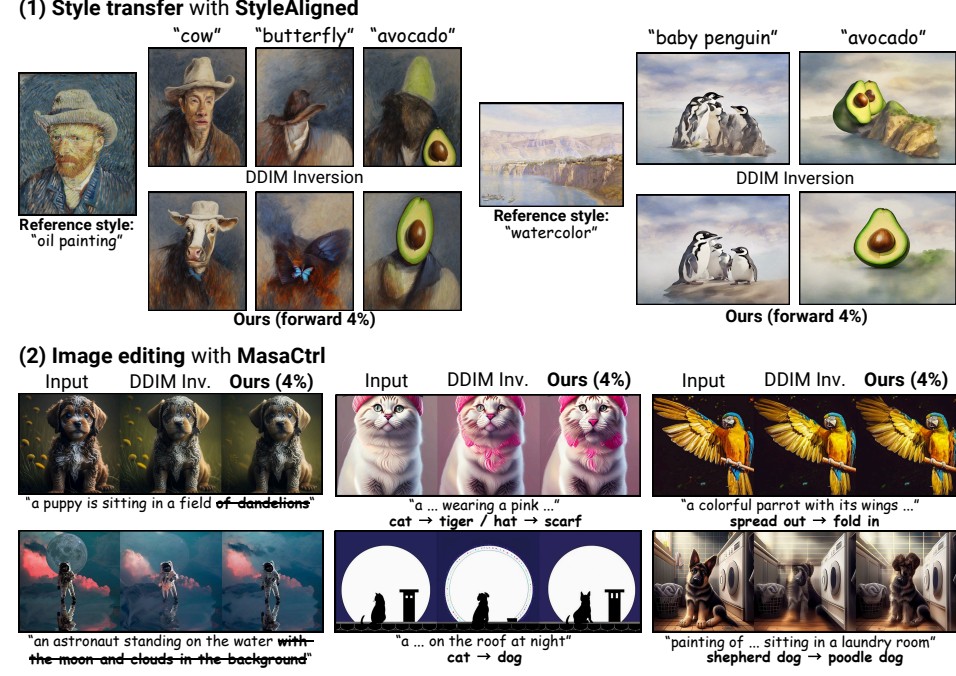

Figure 8: **DDIM Inversion with our fix, when merged to popular image manipulation engines, improves (1) style transfer with StyleAligned and (2) image editing with MasaCtrl.** We use real images from, accordingly, StyleDrop (Sohn et al., 2023) and PIEBench (Ju et al., 2024) benchmarks.

## 5 CONCLUSIONS

This work demonstrates that DDIM inversion errors cause latent representations to systematically deviate from a Gaussian distribution, particularly in smooth regions of the source image. We trace this to high inversion error and insufficiently diverse noise during the early noising steps, and demonstrate that this divergence degrades the quality of image editing and interpolation. We propose a simple fix by replacing initial inversion steps with forward diffusion, which successfully decorrelates the latents and significantly improves sample quality in practical applications.

## ACKNOWLEDGMENTS

We thank (in random order) Wojciech Masarczyk, Bartosz Cywiński, Katarzyna Zaleska, and Paweł Skierś for insightful feedback and discussions throughout the project. This work was funded by the National Science Centre, Poland, grant no UMO-2023/51/B/ST6/03004. The computing resources were provided by the PL-Grid Infrastructure, grant no.: PLG/2025/018390 and PLG/2025/018424. This paper has been supported by the Horizon Europe Programme (HORIZON-CL4-2022-HUMAN-02) under the project "ELIAS: European Lighthouse of AI for Sustainability", GA no. 101120237.

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

# APPENDIX

## CONTENTS

In the Appendix, we first outline the limitations **(A)** of our experiments, LLM usage during writing **(B)**, discuss the broader impact **(C)** of this work, and list the computational resources **(D)** used. Following, we describe, in detail, the DDIM approximation error **(E)**, and the fix we introduce (pseudocode) in this work **(F)**. Next, we describe our experiments: measuring the noise–image–latent triangles **(G)**, methodology for identifying plain-regions pixels **(H)** in the image, and computing the inversion error **(I)**. In **(J)**, we demonstrate how we condition the models, and in **(K)**, we present more details on image diversity and prompt alignment during editing. Next, we compare Gaussian noise and DDIM latents in their mappings to images **(L)** and track how these relationships evolve during DM training **(M)**. In section **(N)** we discuss the impact of different parameters' values: number of inversion steps used, number of inversion steps replaced with forward diffusion, and the impact of guidance scale. In **(O)**, we include additional qualitative results for image interpolation, reconstruction, editing, and style transfer. Finally, in section **(P)**, we present that the investigated issue with latents' correlations also exists in Flow Matching models.

# A  LIMITATIONS

In this work, we analyzed the relation between the random noise, image generations and their latent encodings obtained through DDIM Inversion. While our studies focused on DDIM approximation error from Song et al. (2021), there exist other solvers and inversion methods, as described in Section 2, bringing their own advantages and limitations. The error of DDIM inversion strongly depends on the number of steps with which it is performed. In particular, performing the process very granularly, e.g., using $T = 1000$ steps, can result in strong suppression of the correlation. Nevertheless, the default number of steps we have chosen, i.e., 100, is, according to previous works (Hong et al., 2024; Garibi et al., 2024; Kim et al., 2022b), a practical choice as a good balance between the reconstruction error and the speed of the algorithm. In Appendix N.1, we present that the latents exhibit correlations when using 1000 sampling steps, and that the proposed fix can also help in such cases.

The observations from our analytical experiments (correlations in Table 2, interpolations in Fig. 5) generalize well to all tested diffusion models, but are less evident in the LDM model trained on the CelebA-HQ images. We attribute this exemption to the specificity of the dataset on which the model was trained - photos with centered human faces, usually with uniform backgrounds. We believe that, unlike models trained on a larger number of concepts, the process of generating faces with uniform backgrounds is more stable and introduces little detail in subsequent steps, making the difference in approximation error for plain and non-plain areas less significant.

As mentioned in Section 2, the DDIM inversion error can be detrimental when using a small number of steps. Even though the solution proposed in this work (involving forward diffusion in first inversion steps) drastically removes correlations in latents and, thus, improves image interpolation and editing, it does not improve the numerical inversion error resulting from using a small number of steps. In our experiments with 50 steps that are commonly used for editing, we show no significant drawbacks. However, in the extreme cases, using our fix in even a single step might result in the loss of information necessary for correct image reconstruction, hence it may then be less preferred than standard DDIM inversion. In Appendix N.1, we present failure cases for the introduced solution.

# B  LLM USAGE

Throughout the preparation of this manuscript, we employed a large language model (LLM) as a writing assistant. Its use was focused on improving the clarity and readability of the text, correcting grammar, and refining sentence structure. The authors carefully reviewed, edited, and take full responsibility for all content, ensuring the scientific integrity and accuracy of the final paper.

# C  BROADER IMPACT

As our work is mostly analytical, we do not provide new technologies that might have a significant societal impact. However, our solution for improving the accuracy of DDIM inversion has potential implications that extend beyond technical advancements in diffusion models. As our fix enables more prompt-aligned image editing, it could be combined with various editing engines and misused to advance image manipulation techniques. The enhanced interpolation quality could make synthetic content more convincing and harder to detect. The authors do not endorse using the method for deceptive or malicious purposes, and discourage any application that could erode trust or cause harm.

# D  COMPUTE RESOURCES

For the experiments, we used a scientific cluster consisting of 110 nodes with CrayOS operating system. Each node is powered by 288 CPU cores, stemming from 4 NVIDIA Grace processors, each with 72 cores and a clock speed of 3.1 GHz. The nodes are equipped with substantial memory, featuring 480 GB of RAM per node. For GPU acceleration, each node in the cluster consists of 4 NVIDIA GH200 96GB GPUs with 120 GB of RAM and 72 CPUs per GPU.

Almost all the experiments we perform are based on performing a sampling process using a diffusion model from noise, performing DDIM inversion, and, possibly, image reconstruction from the latent, where each of these processes takes the same number of steps, hence the same number of GPU-hours on average. As our experiments differ in terms of number of sampling steps and images to generate, we provide average GPU time **per one sampling step** per batch (with $B$ denoting batch size) for each model as following: ADM-32 ($B = 256$): 0.054s, ADM-64 ($B = 128$): 0.093s, ADM-256 ($B = 64$): 0.901s, LDM ($B = 128$): 0.273s, DiT ($B = 128$): 0.104s, IF ($B = 64$): 0.609s. Note that some experiments, such as analyzing inversion approximation errors per step (Fig. 4) or sampling from noise interpolations (Fig. 5), involves performing the procedures several times. Taking into consideration all the experiments described in the main text of this work, fully reproducing them takes roughly 110 GPU hours. However, considering the prototyping time, preliminary and failed experiments, as well as the fact that most of the experiments must be performed sequentially (e.g., inversion after image generation, image reconstruction after inversion), the overall execution time of the entire research project is multiple times longer.

## E    APPROXIMATION ERROR IN DDIM INVERSION

Denoising Diffusion Probabilistic Models (DDPMs, Ho et al. (2020)) generate samples by reversing the forward diffusion process, modeled as a Markov Chain, where a clean image $x_0$ is progressively transformed to white Gaussian noise $x_T$ in $T$ diffusion steps. A partially noised image $x_t$, which serves as an intermediate object in this process, is expressed as $x_t = \sqrt{\bar{\alpha}_t}x_0 + \sqrt{1 - \bar{\alpha}_t}\epsilon_t$ where $\bar{\alpha}_t = \prod_{s=1}^{t} \alpha_s$, $\alpha_t = 1 - \beta_t$, $\epsilon_t \sim \mathcal{N}(0, \mathcal{I})$, and $\{\beta_t\}_{t=1}^{T}$ is a variance schedule, controlling how much of the noise is contained in the image at the specific step $t$.

To enable sampling clean images from clean Gaussian noises, the neural network $\epsilon_\theta$ is trained to predict the noise added to a clean image $x_0$ for a given intermediate image $x_t$. Such a trained model is further utilized in the backward diffusion process by iteratively transferring a more noisy image ($x_t$) to the less noisy one ($x_{t-1}$) as

$$x_{t-1} = \frac{1}{\sqrt{\alpha_t}}(x_t - \frac{\beta_t}{\sqrt{1 - \bar{\alpha}_t}}\epsilon_\theta^{(t)}(x_t)) + \sigma_t z, \tag{6}$$

with $z \sim \mathcal{N}(0, \mathcal{I})$ being a noise portion added back for denoising controlled by $\sigma_t = \sqrt{\beta_t(1 - \bar{\alpha}_{t-1})/(1 - \bar{\alpha}_t)}$.

Song et al. (2021) reformulate the diffusion process as a non-Markovian, which leads to a speed-up of the sampling process. While previously obtaining a less noisy image at $x_t$ required all past denoising steps from $T$ till $(t+1)$, this approach allows skipping some steps during sampling. More precisely, the backward diffusion process is defined as a combination of predictions of image $x_0$, next denoising step $x_t$, and random noise (with $\sigma_t = \eta\sqrt{\beta_t(1 - \bar{\alpha}_{t-1})/(1 - \bar{\alpha}_t)}$ and $z_t \sim \mathcal{N}(0, \mathcal{I})$):

$$x_{t-1} = \sqrt{\bar{\alpha}_{t-1}}\left(\frac{\mathbf{x}_t - \sqrt{1 - \bar{\alpha}_t}\,\epsilon_\theta^{(t)}(x_t)}{\sqrt{\bar{\alpha}_t}}\right) + \sqrt{1 - \bar{\alpha}_{t-1} - \sigma_t^2} \cdot \epsilon_\theta^{(t)}(x_t) + \sigma_t z_t. \tag{7}$$

While setting $\eta = 1$ makes Eq. (7) equivalent to Eq. (6), leading to a Markovian probabilistic diffusion model, setting $\eta = 0$ removes the random component from the equation, making it a Denoising Diffusion Implicit Model (DDIM), which prominent ability is to perform a deterministic mapping from given noise ($x_T$) to image ($x_0$). One of the potential benefits of implicit models is the possibility of reversing the backward diffusion process to transfer images back to the original noise. Operating in such a space by modifying resulting inversions unlocks numerous image manipulation capabilities, i.a., image editing (Hertz et al., 2022; Mokady et al., 2023; Huberman-Spiegelglas et al., 2024; Parmar et al., 2023; Rout et al., 2025; Miyake et al., 2023; Brack et al., 2024; Tang et al., 2024; Hong et al., 2024; Wallace et al., 2023; Samuel et al., 2025; Garibi et al., 2024; Pan et al., 2023; Dong et al., 2023), image interpolation (Zheng et al., 2024; Zhang et al., 2024b; Samuel et al., 2023; Dhariwal & Nichol, 2021), or stroke-to-image synthesis (Meng et al., 2022; Rout et al., 2025). The inversion process can be derived from Eq. (7), leading to the formula for transferring a less noisy image $x_{t-1}$ to a more noisy one $x_t$:

$$x_t = \sqrt{\alpha_t}x_{t-1} + (\sqrt{1 - \bar{\alpha}_t} - \sqrt{\alpha_t - \bar{\alpha}_t}) \cdot \epsilon_\theta^{(t)}(x_t) \tag{8}$$

Unfortunately, a perfect image-to-noise inversion is infeasible. Due to circular dependency on $x_t$ within Eq. (8), Dhariwal & Nichol (2021) propose to approximate this equation by assuming that the model's prediction in $t$-th step for $x_t$ is locally equivalent to the decision for $x_{t-1}$: $\epsilon_\theta^{(t)}(x_t) \approx \epsilon_\theta^{(t)}(x_{t-1})$. The inverted trajectory is determined in multiple steps. Hence, the error propagates further away from the image, leading to the latents that significantly deviate from the starting noise. This flaw is presented in Fig. 9.

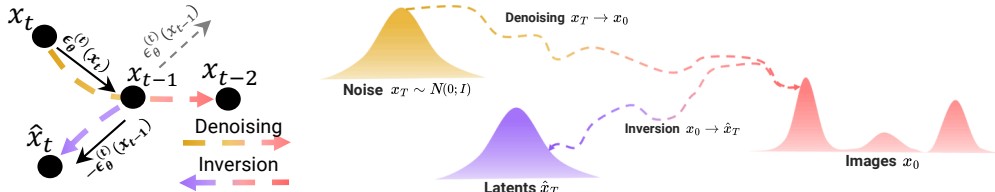

Figure 9: The DDIM inversion error, derived from approximating DM's prediction for $x_t$ with the output for $x_{t-1}$ (left), propagates with the next inversion steps, leading to a distribution of latents $\hat{x}^T$ that deviates significantly from the expected noise distribution $x^T$ (right).

## F    PSEUDOCODE FOR DDIM INVERSION WITH FORWARD DIFFUSION

We present in Algorithm 1 the pseudocode for the proposed solution to the decorrelation of latent encodings by replacing the first $t'$ inversion steps with a forward diffusion process. In Appendices N.1 to N.3, we analyze the sensitivity of this fix – how it performs for: different number of inversion steps $T$ (Appendix N.1), different percentage of inversion steps replaced with forward diffusion $f$ (Appendix N.2), and when classifier-free guidance is applied (Appendix N.3).

---

**Algorithm 1** Finding decorrelated DDIM latent encoding $\hat{\mathbf{x}}_\mathbf{T}$

---

**Require:** image $\mathbf{x_0}$; diffusion model $\epsilon_\theta$; noise schedules $\{\alpha_t\}_{t=1}^T$, $\{\bar{\alpha}_t\}_{t=1}^T$; number of inversion steps $T$; forward-replacement timestep $f$
**Ensure:** decorrelated latent encoding $\hat{\mathbf{x}}_\mathbf{T}$
1:  sample $\tilde{\epsilon} \sim \mathcal{N}(0, \mathcal{I})$
2:  $\hat{\mathbf{x}}_\mathbf{f} \leftarrow \sqrt{\bar{\alpha}_f} \cdot \mathbf{x_0} + \sqrt{1 - \bar{\alpha}_f} \cdot \tilde{\epsilon}$                                          ▷ forward diffusion
3:  **for** $t \leftarrow f + 1, \ldots, T$ **do**
4:      $\hat{\mathbf{x}}_\mathbf{t} \leftarrow \sqrt{\alpha_t} \cdot \hat{\mathbf{x}}_{\mathbf{t-1}} + (\sqrt{1 - \bar{\alpha}_t} - \sqrt{\alpha_t}) \cdot \epsilon_\theta^{(t)}(\hat{\mathbf{x}}_{\mathbf{t-1}})$         ▷ DDIM Inversion step
5:  **end for**
6:  **return** $\hat{\mathbf{x}}_\mathbf{T}$

---

# G  MOST PROBABLE TRIANGLES

In Section 3.2, we analyze where the latent encodings are distributed in relation to the initial noise and generated samples. Here, we determine the most probable angles formed by noises $\mathbf{x_T}$, samples $\mathbf{x_0}$, and latents $\hat{\mathbf{x}}_\mathbf{T}$ for each model.

## G.1  METHODOLOGY

First, we determine the vectors going from each vertex to the other vertices of the noise-sample-latent ($\mathbf{x_T}$-$\mathbf{x_0}$-$\hat{\mathbf{x}}_\mathbf{T}$) triangle. For sample $\mathbf{x_0}$, we obtain a vector leading to noise $\overrightarrow{\mathbf{x_0}\mathbf{x_T}} = \mathbf{x_T} - \mathbf{x_0}$ and to latent $\overrightarrow{\mathbf{x_0}\hat{\mathbf{x}}_\mathbf{T}} = \hat{\mathbf{x}}_\mathbf{T} - \mathbf{x_0}$, and calculate the angle between them using cosine similarity as

$$\angle\mathbf{x_0} = \arccos \frac{\overrightarrow{\mathbf{x_0}\mathbf{x_T}} \cdot \overrightarrow{\mathbf{x_0}\hat{\mathbf{x}}_\mathbf{T}}}{\|\overrightarrow{\mathbf{x_0}\mathbf{x_T}}\|\|\overrightarrow{\mathbf{x_0}\hat{\mathbf{x}}_\mathbf{T}}\|}, \tag{9}$$

and convert resulting radians to degrees. Similarly, we obtain the angle next to the noise $\angle\mathbf{x_T}$ and latent $\angle\hat{\mathbf{x}}_\mathbf{T}$.

Next, we determine histograms for each angle, approximating the probability density function for every angle ($p_{\angle\mathbf{x_T}}, p_{\angle\mathbf{x_0}}, p_{\angle\hat{\mathbf{x}}_\mathbf{T}}$) binned up to the precision of one degree, see Fig. 11. Finally, for all angles triples candidates (where $\angle\mathbf{x_T} + \angle\mathbf{x_0} + \angle\hat{\mathbf{x}}_\mathbf{T} = 180°$), we calculate the probability of a triangle as the product of the probabilities and choose the triplet maximizing such joint probability:

$$\underset{(\angle\mathbf{x_T}, \angle\mathbf{x_0}, \angle\hat{\mathbf{x}}_\mathbf{T})}{\arg\max} \ p_{\angle\mathbf{x_T}} \cdot p_{\angle\mathbf{x_0}} \cdot p_{\angle\hat{\mathbf{x}}_\mathbf{T}}. \tag{10}$$

## G.2  RESULTS

Results of the experiment in Table 9 show that the angle located at the image and noise vertices (accordingly $\angle\mathbf{x_0}$ and $\angle\mathbf{x_T}$) are always acute and, in almost every case, the angle by the latent vertex ($\angle\hat{\mathbf{x}}_\mathbf{T}$) is obtuse. This property implies that, due to approximation errors in the reverse DDIM process, latents reside in proximity to, but with a measurable offset, from the shortest-path trajectory between the noise distribution and the generated image.

| Model | $T$ | $\angle\mathbf{x_0}$ | $\angle\mathbf{x_T}$ | $\angle\hat{\mathbf{x}}_\mathbf{T}$ |
|---|---|---|---|---|
| ADM | 10 | 44 | 16 | 120 |
| | 100 | 29 | 28 | 123 |
| $32 \times 32$ | 1000 | 20 | 45 | 115 |
| ADM | 10 | 30 | 31 | 119 |
| | 100 | 11 | 60 | 109 |
| $64 \times 64$ | 1000 | 6 | 79 | 95 |
| ADM | 10 | 24 | 50 | 106 |
| | 100 | 24 | 73 | 83 |
| $256 \times 256$ | 1000 | 23 | 73 | 84 |
| LDM | 10 | 23 | 53 | 104 |
| | 100 | 2 | 76 | 102 |
| $256 \times 256$ | 1000 | 1 | 83 | 96 |
| DiT | 10 | 27 | 47 | 106 |
| | 100 | 4 | 66 | 110 |
| $256 \times 256$ | 1000 | 1 | 80 | 99 |

Table 9: **Impact of the number of diffusion steps $T$ on angles in the noise $\mathbf{x_T}$, image $\mathbf{x_0}$, and latent $\hat{\mathbf{x}}_\mathbf{T}$ triangle.** Regardless of the number of diffusion steps, latents appear between Gaussian noise and generations.

In Fig. 10, we provide example triangles for ADM-32 (a), ADM-256 (b), and LDM (c), which we calculate using $N = 1000$ images with $T = 100$ diffusion steps.

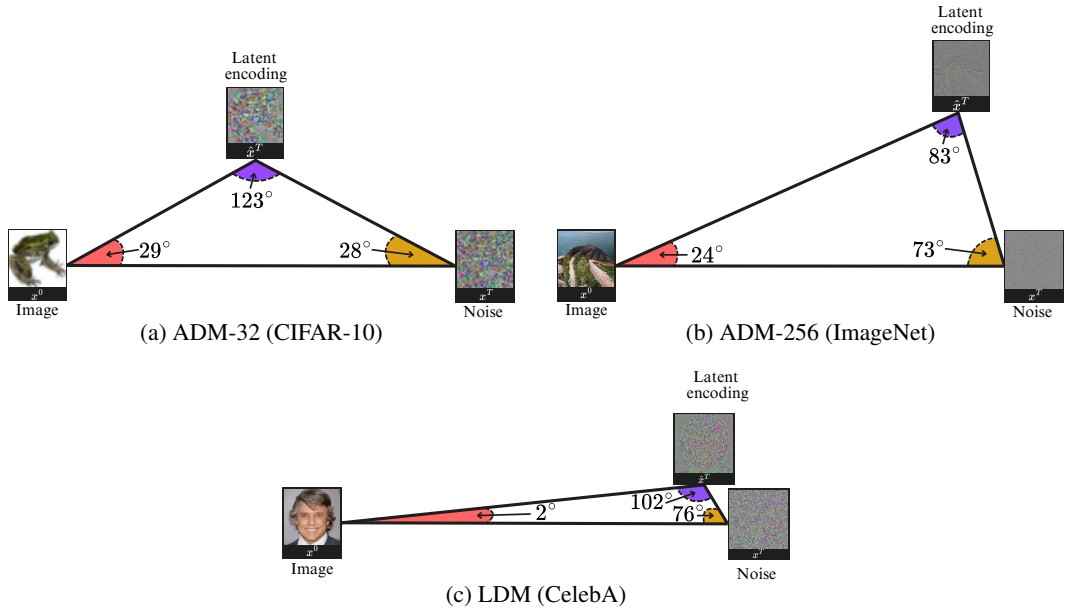

Figure 10: **Most probable triangles formed from random Gaussian noise ($x_T$), the images ($x_0$) generated, and latents ($\hat{x}_T$) recovered with DDIM Inversion procedure.**

### G.3 EXAMPLE HISTOGRAMS

In Fig. 11 we present histograms approximating probabiliy density functions for noise $x_T$, image $x_0$, and DDIM latent $\hat{x}_T$ angles.

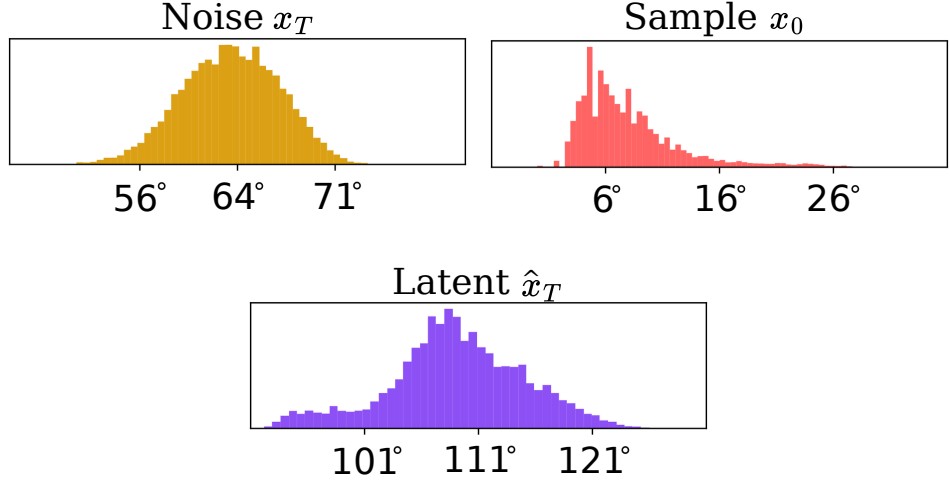

Figure 11: **Histograms approximating probability density functions of angles values for noise, sample, and latent vertices.** Example calculated for DiT model using $T = 100$ diffusion steps.

# H PLAIN SURFACE THRESHOLDING

During our experiments, we determine binary mask $\mathcal{M}$ to identify pixels corresponding to the plain areas in the images. We describe this process in this section.

Let $\mathbf{I} \in \mathbb{R}^{C \times H \times W}$ be the input image. For each pixel value across every channel $(c, h, w)$ in $\mathbf{I}$, we compute the absolute difference to point in the next row $D_H(c, h, w) = |I_{c,h+1,w} - I_{c,h,w}|$ and to the pixel in the next column $D_W(c, h, w) = |I_{c,h,w+1} - I_{c,h,w}|$. We obtain $D_H \in \mathbb{R}^{C \times H - 1 \times W}$ and $D_W \in \mathbb{R}^{C \times H \times W - 1}$, which we pad with zeros (last row for $D_H$ and last column for $D_W$), making them of shape $C \times H \times W$. The difference matrix $D$, representing how a point varies from its neighbors, is computed as $D = (D_W + D_H)/2$.

Finally, we determine a binary mask $\mathcal{M}_c$ per each channel $c$, by applying threshold $\tau$ to $D$ as

$$\mathcal{M}_c(h, w) = \begin{cases} 1, & \text{if } D_c(h, w) < \tau \\ 0, & \text{otherwise} \end{cases} \tag{11}$$

During the experiments, we set $\tau = 0.025$.

The final mask $\mathcal{M} \in \{0, 1\}^{W \times H}$ can be derived by evaluating the logical AND across all channels for each pixel location as

$$\mathcal{M}(h, w) = \prod_c \mathcal{M}_c(h, w). \tag{12}$$

After obtaining the final mask for plain pixels, which we denote as $\mathcal{M}_p$, the according mask for non-plain image surfaces can be obtained by applying logical NOT to the mask as $\mathcal{M}_n = \neg \mathcal{M}_p$. In Fig. 12, we present example masks determined using our methodology.

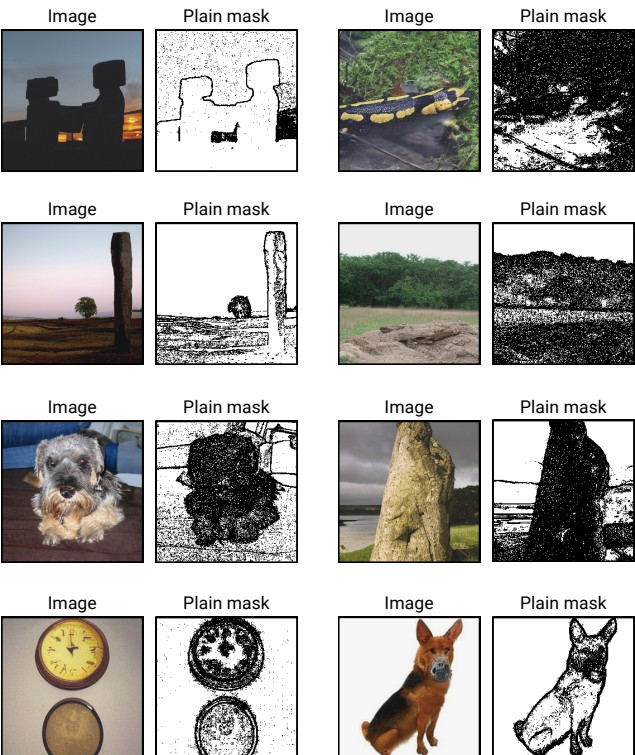

Figure 12: **Examples of samples together with their masks indicating plain areas (white).** Images generated using DiT model.

## I  CALCULATING INVERSION ERROR

In Section 3.3, we study how the inversion error differs for pixels related to plain and non-plain samples' regions and investigate it across diffusion steps. In this section, we describe the method for determining the error.

First, we generate $N = 4k$ images with $T = 50$ diffusion steps with both ADM-64 and DiT models, saving intermediate noise predictions $\{\mathcal{E}_t^S\}_{t=50...1}$ during sampling. Next, we collect diffusion model outputs during the inversion process $\{\mathcal{E}_t^I\}_{t=1...50}$ assuming all the previous steps were correct. While for $t = 1$, $\mathcal{E}_1^I$ can be set to the diffusion model prediction from the first inversion step $\epsilon_\theta(x_0)$, for further steps, more advanced methodology is necessary. To this end, for each $t' = 2...50$, we map the images to latents with DDIM Inversion, replacing in $t = 1...t'-1$ the predicted noise $\epsilon_\theta(x_t)$ with the ground truth prediction $\mathcal{E}_t^S$ from denoising process, and collect the model output for the $t'$ step as $\mathcal{E}_{t'}^I := \epsilon_\theta(x_{t'})$. For step $t$, this methodology is equivalent to starting the inversion process from $(t+1)$ denoising step and collecting the first diffusion model prediction.

In the next step, we calculate the absolute error between outputs during inversion and ground truth predictions as $\mathcal{E}_t^E = |\mathcal{E}_t^I - \mathcal{E}_t^S|$. This way, we obtain the inversion approximation error for each timestep.

Further, for the images in the dataset, we obtain binary masks $\mathcal{M}_p$ and $\mathcal{M}_n$ indicating, respectively, plain ($p$) and non-plain ($n$) pixels in the image, according to the procedure described in Appendix H. To ensure that the level of noise that DM predicts in each step does not bias the results, we divide the absolute errors in each step by $l_1$-norm of model outputs, calculated separately for each diffusion step. For plain ($p$) pixels, it can be described as

$$\mathcal{E}_t^{Ep} = 1/\|\mathcal{E}_t^I\|_1 \sum \left(\mathcal{E}_t^E \odot \mathcal{M}_p\right), \tag{13}$$

and adequately for non-plain ($n$) pixels as

$$\mathcal{E}_t^{En} = 1/\|\mathcal{E}_t^I\|_1 \sum \left(\mathcal{E}_t^E \odot \mathcal{M}_n\right). \tag{14}$$

In Table 10, we present the error differences for plain and non-plain areas and how the first 10% of the diffusion steps contribute to the total inversion error. To calculate this error for plain regions in steps $t_s, \ldots, t_e$, we sum errors for timesteps from a given interval as

$$\bar{\mathcal{E}}_{(t_s,t_e)}^{Ep} = \sum_{t'=t_s}^{t_e} \mathcal{E}_t^{Ep}. \tag{15}$$

| Pixel area | Diffusion steps | | Model | |
|---|---|---|---|---|
| | | | ADM-64 | DiT |
| Plain | $1,2\ldots50$ | 100% | 15.11 | 5.67 |
| Non-plain | | | 12.43 | 3.16 |
| Plain | $1,2\ldots5$ | 10% | 9.23 | 3.41 |
| Non-plain | | | 6.80 | 1.42 |
| Plain | $6,7\ldots50$ | 90% | 5.87 | 2.23 |
| Non-plain | | | 5.63 | 1.74 |

Table 10: **Total per-pixel inversion error (normalized) over different timestep ranges** $\bar{\mathcal{E}}_{(t_s,t_e)}^E$ **for plain and non-plain areas.** Inversion approximation error is higher for pixels related to plain image areas, especially in the first 10% of inversion steps.

## J    CONDITIONING FOR DIFFUSION MODELS

For a thorough analysis, we employ both unconditional (ADM-32, ADM-64, ADM-256) and conditional (DiT, IF, SDXL) diffusion models. For conditioning, we take prompts from the Recap-COCO-30K dataset (Li et al., 2025) for IF and Stable Diffusion XL and ImageNet class names for DiT. However, as noted by Mokady et al. (2023), Classifier-Free Guidance introduces additional errors to the DDIM inversion. To focus solely on the inversion approximation error, in the experiments from the main part of this work, we disable CFG by setting the guidance scale to $w = 1$. However, in Appendix N.3, we show that the proposed fix can also improve DDIM Inversion when CFG is enabled ($w > 1$).

## K    DETAILS ON DIVERSITY AND ALIGNMENT OF EDITION

In this section, we provide results for measuring the diversity (against source images $I_S$) and alignment to conditioning prompts (source $P_S$ and target $P_T$) of images generated from Gaussian noise $\mathbf{x_T}$, DDIM latents $\hat{\mathbf{x}}_T$, or the latents produced by our fix (described in Algorithm 1) with modified prompts. We use distance-based metrics (LPIPS (Zhang et al., 2018), DreamSim (Fu et al., 2023)), similarity metrics (SSIM (Wang et al., 2004), DINO (Darcet et al., 2024)) to measure the diversity between source (input) images and target (edited) images, as well as the similarity between embeddings produced by the CLIP (Radford et al., 2021) model to calculate the alignment between the results and prompts. The results in for introduced latent decorrelation procedure are obtained with varying percentages of the first DDIM inversion steps replaced with forward diffusion. In Table 11, we show that by selecting a small fraction of inversion steps to replace with forward diffusion (from $2\%$ up to $6\%$), the resulting latents are more editable.

| Inversion steps replaced | Diversity against $I_S$ | | | | CLIP Alignment | |
| by forward diff. (%T) | DreamSim ↑ | LPIPS ↑ | SSIM ↓ | DINO ↓ | $P_S$ ↓ | $P_T$ ↑ |
|---|---|---|---|---|---|---|
| Noise $\mathbf{x_T}$ | 0.709 | 0.591 | 0.228 | 0.174 | 0.465 | 0.681 |
| DDIM Latent $\hat{\mathbf{x}}_T$ | 0.677 | 0.564 | 0.261 | 0.223 | 0.480 | 0.662 |
| 1 (2%) | 0.696 | 0.580 | 0.239 | 0.187 | 0.470 | 0.676 |
| 1 . . . 2 (4%) | 0.701 | 0.587 | 0.227 | 0.179 | 0.469 | 0.677 |
| 1 . . . 3 (6%) | 0.705 | 0.594 | 0.216 | 0.173 | 0.468 | 0.678 |
| 1 . . . 5 (10%) | 0.711 | 0.605 | 0.198 | 0.165 | 0.466 | 0.679 |
| 1 . . . 50 (100%) | 0.805 | 0.801 | 0.013 | 0.085 | 0.465 | 0.680 |

(a) Diffusion Transformer

| Inversion steps replaced | Diversity against $I_S$ | | | | CLIP Alignment | |
| by forward diff. (%T) | DreamSim ↑ | LPIPS ↑ | SSIM ↓ | DINO ↓ | $P_S$ ↓ | $P_T$ ↑ |
|---|---|---|---|---|---|---|
| Noise (100%) | 0.665 | 0.380 | 0.339 | 0.344 | 0.273 | 0.649 |
| DDIM Latent $\hat{\mathbf{x}}_T$ | 0.609 | 0.328 | 0.406 | 0.416 | 0.353 | 0.614 |
| 1 (2%) | 0.666 | 0.376 | 0.359 | 0.335 | 0.281 | 0.646 |
| 1 . . . 2 (4%) | 0.660 | 0.369 | 0.359 | 0.351 | 0.287 | 0.649 |
| 1 . . . 3 (6%) | 0.661 | 0.370 | 0.353 | 0.349 | 0.285 | 0.650 |
| 1 . . . 5 (10%) | 0.662 | 0.372 | 0.341 | 0.346 | 0.282 | 0.649 |
| 1 . . . 50 (100%) | 0.733 | 0.521 | 0.004 | 0.272 | 0.274 | 0.649 |

(b) Deepfloyd IF

Table 11: **Impact of first DDIM inversion errors on diversity and text-alignment of generations from resulting latent as an input.** For both DiT (a) and IF (b) models, replacing the first inversion steps and denoising leads to more diverse generations against the source images $I_S$. Additionally, we show using forward diffusion in first steps improves the alignment between generation and target prompts, which the generation process is conditioned by, as indicated by the larger CLIP-T value for $P_T$ and the smaller one for $P_S$.

## L  NOISE-TO-IMAGE MAPPING

We showcase an additional study showing the differences that occur between noise and latent encodings, from the perspective of their mapping to the images. Several works investigate interesting properties between the initial random noise and generations that result from the training objective of DDPMs and score-based models. Kadkhodaie et al. (2024) show that due to inductive biases of denoising models, different DDPMs trained on similar datasets converge to almost identical solutions. This idea is further explored by Zhang et al. (2024a), observing that even models with different architectures converge to the same score function and, hence, the same noise-to-image mapping. Khrulkov & Oseledets (2022) show that diffusion models' encoder map coincides with the optimal transport (OT) map when modeling simple distributions. However, other works (Kim & Milman, 2011; Lavenant & Santambrogio, 2022) contradict this finding.

### L.1  SMALLEST $l_2$ MAPPING

Diffusion models converge to the same mapping between the Gaussian noise ($\mathbf{x_T}$) and the generated images ($\mathbf{x_0}$) independently on the random seed, dataset parts (Kadkhodaie et al., 2024), or the model architecture (Zhang et al., 2024a). We further investigate this property from the noise-sample and latent-sample mapping perspective.

In our experimental setup, we start by generating $N = 2000$ images $\mathbf{x_0}$ from Gaussian noise $\mathbf{x_T}$ with a DDIM sampler and invert the images to latents $\hat{\mathbf{x}}_\mathbf{T}$ with naïve DDIM inversion. Next, we predict resulting images for the starting noise samples ($\mathbf{x_T} \rightarrow \mathbf{x_0}$) by iterating over all the $N$ noises, and for each of them, we calculate its pixel distances to all the $N$ generations. For given noise, we select the image to which such $l_2$-distance is the smallest. Similarly, we investigate image-to-noise ($\mathbf{x_0} \rightarrow \mathbf{x_T}$), image-to-latent ($\mathbf{x_0} \rightarrow \hat{\mathbf{x}}_\mathbf{T}$) and latent-to-image ($\hat{\mathbf{x}}_\mathbf{T} \rightarrow \mathbf{x_0}$) mappings. We calculate the distance between two objects as $l_2$ norm of the matrix of differences between them (with $C \times H \times W$ being the dimensions of either pixel or latent space of diffusion model) as $||x - y||_2 = \sqrt{\Sigma_i^C \Sigma_j^H \Sigma_j^W (x_{i,j,k} - y_{i,j,k})^2}$.

In Table 12, we investigate the accuracy of the procedure across varying numbers of diffusion steps $T$ for both image↔noise (a) and image↔latent (b) assignments. We show that assigning initial noise to generations ($\mathbf{x_0} \rightarrow \mathbf{x_T}$) through the distance method can be successfully done regardless of diffusion steps. For the reverse assignment, which is noise-to-image ($\mathbf{x_T} \rightarrow \mathbf{x_0}$) mapping, we can observe high accuracy with a low number of generation timesteps ($T = 10$), but the results deteriorate quickly with the increase of this parameter. The reason for this is that greater values of $T$ allow the generation of a broader range of images, including the ones with large plain areas of low pixel variance. When it comes to mappings between images and latents resulting from DDIM Inversion, assignment in both directions is infeasible for pixel diffusion, regardless of $T$.

| T | ADM-32 | | ADM-64 | | ADM-256 | | LDM | | DiT | |
|---|---|---|---|---|---|---|---|---|---|---|
| | $\mathbf{x_0} \rightarrow \mathbf{x_T}$ | $\mathbf{x_T} \rightarrow \mathbf{x_0}$ | $\mathbf{x_0} \rightarrow \mathbf{x_T}$ | $\mathbf{x_T} \rightarrow \mathbf{x_0}$ | $\mathbf{x_0} \rightarrow \mathbf{x_T}$ | $\mathbf{x_T} \rightarrow \mathbf{x_0}$ | $\mathbf{x_0} \rightarrow \mathbf{x_T}$ | $\mathbf{x_T} \rightarrow \mathbf{x_0}$ | $\mathbf{x_0} \rightarrow \mathbf{x_T}$ | $\mathbf{x_T} \rightarrow \mathbf{x_0}$ |
| 10 | $90.3_{\pm6.3}$ | $94.0_{\pm2.6}$ | $99.4_{\pm0.0}$ | $100_{\pm0.0}$ | $100_{\pm0.0}$ | $39.2_{\pm6.2}$ | $100_{\pm0.0}$ | $100_{\pm0.0}$ | $100_{\pm0.0}$ | $93.7_{\pm7.2}$ |
| 100 | $98.9_{\pm1.2}$ | $50.4_{\pm1.9}$ | $100_{\pm0.0}$ | $59.0_{\pm7.1}$ | $100_{\pm0.0}$ | $23.2_{\pm4.8}$ | $100_{\pm0.0}$ | $100_{\pm0.0}$ | $100_{\pm0.0}$ | $90.7_{\pm10.1}$ |
| 1000 | $99.1_{\pm1.0}$ | $46.8_{\pm3.0}$ | $99.8_{\pm0.2}$ | $44.6_{\pm6.3}$ | $100_{\pm0.0}$ | $25.0_{\pm4.4}$ | $100_{\pm0.0}$ | $100_{\pm0.0}$ | $100_{\pm0.0}$ | $96.7_{\pm4.6}$ |
| 4000 | $99.1_{\pm1.0}$ | $46.4_{\pm3.0}$ | $99.5_{\pm0.3}$ | $43.3_{\pm6.7}$ | - | - | - | - | - | - |

(a) Assigning noise to the corresponding generated image ($\mathbf{x_0} \rightarrow \mathbf{x_T}$) and vice-versa ($\mathbf{x_T} \rightarrow \mathbf{x_0}$).

| T | ADM-32 | | ADM-64 | | ADM-256 | | LDM | | DiT | |
|---|---|---|---|---|---|---|---|---|---|---|
| | $\mathbf{x_0} \rightarrow \hat{\mathbf{x}}_\mathbf{T}$ | $\hat{\mathbf{x}}_\mathbf{T} \rightarrow \mathbf{x_0}$ | $\mathbf{x_0} \rightarrow \hat{\mathbf{x}}_\mathbf{T}$ | $\hat{\mathbf{x}}_\mathbf{T} \rightarrow \mathbf{x_0}$ | $\mathbf{x_0} \rightarrow \hat{\mathbf{x}}_\mathbf{T}$ | $\hat{\mathbf{x}}_\mathbf{T} \rightarrow \mathbf{x_0}$ | $\mathbf{x_0} \rightarrow \hat{\mathbf{x}}_\mathbf{T}$ | $\hat{\mathbf{x}}_\mathbf{T} \rightarrow \mathbf{x_0}$ | $\mathbf{x_0} \rightarrow \hat{\mathbf{x}}_\mathbf{T}$ | $\hat{\mathbf{x}}_\mathbf{T} \rightarrow \mathbf{x_0}$ |
| 10 | $66.4_{\pm1.7}$ | $38.2_{\pm5.1}$ | $64.4_{\pm7.1}$ | $100.0_{\pm0.0}$ | $0.7_{\pm0.2}$ | $30.8_{\pm4.3}$ | $100_{\pm0.0}$ | $100_{\pm0.0}$ | $99.8_{\pm0.6}$ | $95.1_{\pm6.4}$ |
| 100 | $16.4_{\pm6.1}$ | $33.4_{\pm2.7}$ | $8.6_{\pm9.3}$ | $57.5_{\pm7.3}$ | $4.1_{\pm1.4}$ | $23.9_{\pm5.0}$ | $100_{\pm0.0}$ | $100_{\pm0.0}$ | $99.5_{\pm1.7}$ | $90.7_{\pm10.3}$ |
| 1000 | $3.6_{\pm2.2}$ | $40.9_{\pm2.7}$ | $1.7_{\pm1.3}$ | $44.7_{\pm6.5}$ | $23.9_{\pm5.2}$ | $25.4_{\pm4.4}$ | $100_{\pm0.0}$ | $100_{\pm0.0}$ | $100.0_{\pm0.2}$ | $96.6_{\pm4.6}$ |
| 4000 | $2.8_{\pm2.2}$ | $41.9_{\pm3.0}$ | $1.9_{\pm1.4}$ | $43.5_{\pm6.5}$ | - | - | - | - | - | - |

(b) Assigning images to the resulting latent encodings ($\hat{\mathbf{x}}_\mathbf{T} \rightarrow \mathbf{x_0}$) and vice-versa ($\mathbf{x_0} \rightarrow \hat{\mathbf{x}}_\mathbf{T}$).

Table 12: **Accuracy of the $l_2$-distance based assignment for both image↔noise (a) and image↔latent (b) mappings across varying number of diffusion steps $T$.** For pixel DMs, only the image-to-noise ($\mathbf{x_0} \rightarrow \mathbf{x_T}$) mapping is feasible. For the latent space models, we correctly predict assignments in all directions.

For both noise $x^T$ and latents $\hat{x}^T$, their assignment to images in both directions can be successfully done when the denoising is performed in the latent space, as shown for DiT and LDM models. We hypothesize that this fact is connected with the KL regularization term that imposes a slight penalty towards a standard normal distribution $\mathcal{N}(0, I)$ on the latent during training (Rombach et al., 2021).

## L.2 ASYMMETRY OF NOISE-TO-IMAGE MAPPING

Results in Table 12a indicate that, even though the $l_2$-distance is symmetrical, the mapping cannot be done in both directions. The reason behind this is that image and noise assignments are not the same due to the one-directional many-to-one relation, e.g., there might be several noises pointing towards the same closest image.

We present examples of wrong noise-to-image ($\mathbf{x_T} \rightarrow \mathbf{x_0}$) assignments in Fig. 13A for ADM-64. In Fig. 13B, we present the singular generations that lead to incorrect noise-to-image classification (noise attractors), along with the number of noises for which they are the closest. Interestingly, in Fig. 13C, we sort all the generations used in the experiment by the variance of pixels and show 8 least variant images. We observe that the set of singular generations leading to misclassification partially overlaps with lowest-variance generations. In Fig. 14 we observe similar properties for experiment with ADM-32 model.

When assigning images to the initial noises, there are singular generations (with large plain areas) located close to the mean of the random Gaussian noise in the set of generated images. Such generations tend to be the closest (in $l_2$-distance) for the majority of the noises in our experiments.

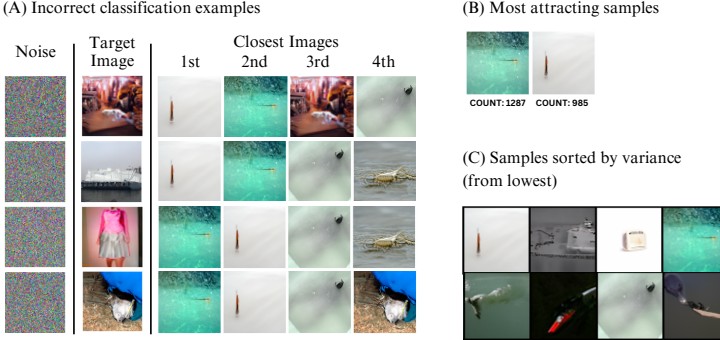

Figure 13: Examples of incorrect assignments of initial noises to resulting images (A), two most noise-attracting images (B), and samples sorted in ascending order by variance of pixels for ADM-64 model trained on the **ImageNet dataset**.

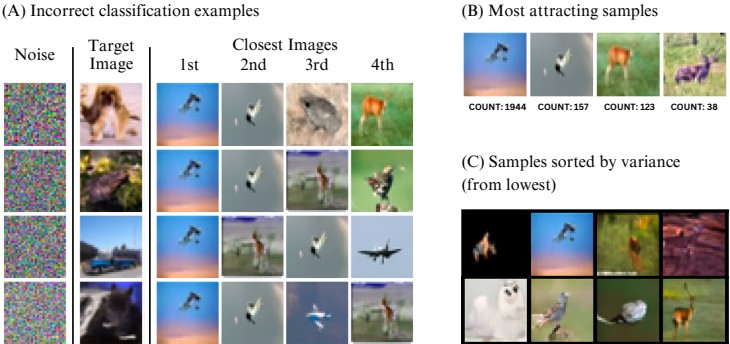

Figure 14: Examples of incorrect assignments of initial noises to resulting images (A), two most noise-attracting images (B), and samples sorted in ascending order by variance of pixels for ADM-32 model trained on the **CIFAR-10 dataset**.

**Conclusion.** Those findings, connected with the reduced diversity of latents (Table 3), suggest that the DDIM latents, unlike noise, cannot be accurately assigned to samples, as the error brings them towards the mean, reducing their diversity and making them closest to most of the images.

## M    ON NOISE-IMAGE-LATENT RELATIONS DURING DIFFUSION TRAINING

To further explore the relationships that exist between noises, generations, and latents, we study how the relationships between them change with the training of the diffusion model. We train two diffusion models from scratch and follow the setup from Nichol & Dhariwal (2021) for two unconditional ADMs for the ImageNet ($64 \times 64$) and CIFAR-10 ($32 \times 32$) datasets. The CIFAR-10 model is trained for 700K steps, while the ImageNet model – for 1.5M steps, both with a batch size of 128. Models employ a cosine scheduler with 4K diffusion steps.

### M.1    SPATIAL RELATIONS OF NOISE AND LATENTS OVER TRAINING TIME

We conclude our latent localization experiments (Section 3.2) by showing that our observations are persistent across the diffusion model training process. We generate $N = 2048$ images with the final model, using implicit sampling with $T = 100$ steps, and invert them to the corresponding latents using checkpoints saved during the training. In Fig. 15, we show that both the angle adjacent to the noise $\angle \mathbf{x_T}$ and the distance between the latent and noise $\|\hat{\mathbf{x}}_{\mathbf{T}} - \mathbf{x_T}\|_2$ quickly converge to the point that remains unchanged through the rest of the training, indicating that the relation between noises, latents, and samples is defined at the very early stage of the training. Additionally, we observe that the noise reconstruction error in DDIM Inversion does not degrade with the training progress.

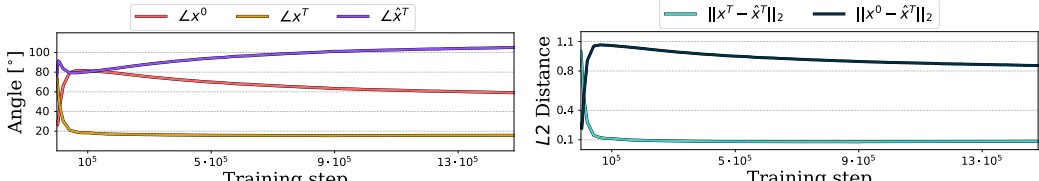

Figure 15: **We investigate spatial relations between the noise $\mathbf{x_T}$, latents $\hat{\mathbf{x}}_{\mathbf{T}}$, and generated images $\mathbf{x_0}$ over training process of diffusion model.** We show that those relations are defined at the early stage of the training.

### M.2    IMAGE-TO-NOISE DISTANCE MAPPING OVER TRAINING TIME

We analyze the image-noise mapping with $l_2$-distance from Appendix L over diffusion model training time. We sample $N = 2000$ Gaussian noises and generate images from them using ADM models with $T = 100$ diffusion steps, calculating the accuracy of assigning images to corresponding noises (and vice versa) using the smallest $l_2$-distance. In Fig. 16, we can observe, for both models, that the distance between noises and their corresponding generations accurately defines the assignment of initial noises given the generated samples ($\mathbf{x_0} \rightarrow \mathbf{x_T}$) from the beginning of the training till the end. At the same time, the accurate reverse assignment ($\mathbf{x_T} \rightarrow \mathbf{x_0}$) can only be observed at the beginning of the training when the trained model is not yet capable of generating properly formed images. Already in the beginning phase of model training, the quality of noise to image mapping rapidly drops and does not change until the end of training.

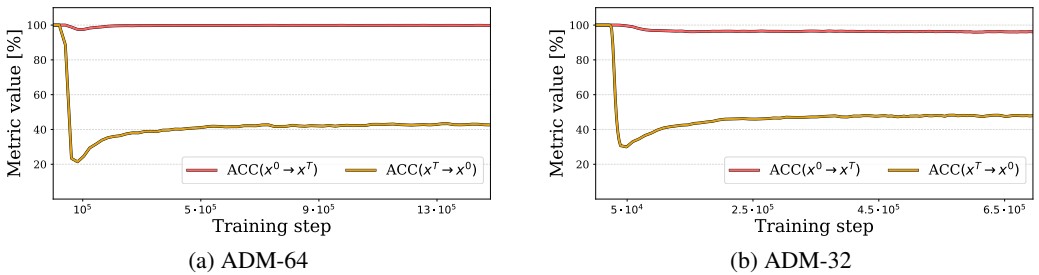

(a) ADM-64                                          (b) ADM-32

Figure 16: **Accuracy of assigning initial noise given the generated sample ($\mathbf{x_0} \rightarrow \mathbf{x_T}$) and sample given the initial noise ($\mathbf{x_T} \rightarrow \mathbf{x_0}$) when training the diffusion model.** We can observe that from the very beginning of training, we can assign initial noise with a simple L2 distance while the accuracy of the reverse assignment rapidly drops.

### M.3 IMAGE ALIGNMENT OVER TRAINING TIME

Inspired by the noise-image mapping experiment, we investigate how the generations resulting from the same noise visually change over DM training time. Thus, for each training step $n \in \{1 \dots 700K\}$ for CIFAR-10 and $\{1 \dots 1.5M\}$ for ImageNet, we generate $2048$ samples $\{\mathbf{x}^{\mathbf{0}}_{i,n}\}^{2048}_{i=1}$ from *the same random noise* $\mathbf{x}^{\text{fixed}}_{\mathbf{T}} \sim \mathcal{N}(0, \mathcal{I})$, and compare them with generations obtained for the fully trained model. We present the visualization of this comparison in Fig. 17 using CKA, DINO, SSIM, and SVCCA as image-alignment metrics. We notice that image features rapidly converge to the level that persists until the end of the training. This means that prolonged learning does not significantly alter how the data is assigned to the Gaussian noise after the early stage of the training. It is especially visible when considering the SVCCA metric, which measures the average correlation of top-10 correlated data features between two sets of samples. We can observe that this quantity is high and stable through training, showing that generating the most important image concepts from a given noise will not be affected by a longer learning process. For visual comparison, we plot the generations sampled from the model trained with different numbers of training steps in Fig. 17 (right).

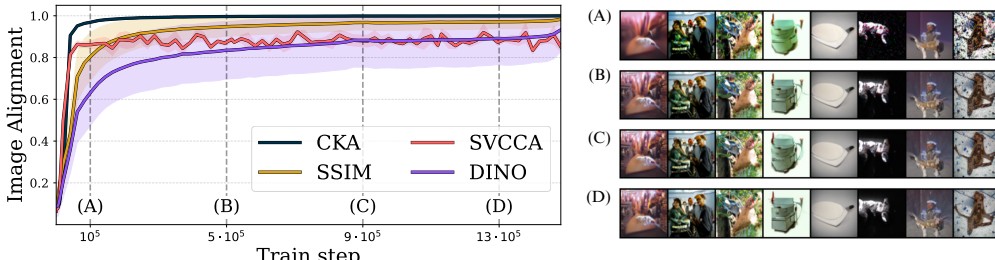

Figure 17: **Similarity of the generations sampled from the same random noise at different stages of diffusion model's training to the final outputs for ADM-64 model.** Only after a few epochs does the model already learn the mapping between Gaussian noise and generations. Prolonged training improves the quality of samples, adding high-frequency features without changing their content. This can be observed through different image alignment metrics (left) and visual inspection (right).

In Fig. 18, we visualize how the diffusion model learns the low-frequency features of the image already at the beginning of the training when comparing generations from the next training steps against the generations after finishing training for the ADM-32 model trained on the CIFAR-10 dataset. In Fig. 19 (ADM-64) and Fig. 20 (ADM-32), we show additional examples illustrating how generations evolve over training for the same Gaussian noise $\mathbf{x}_{\mathbf{T}}$ using a DDIM sampler. Initially, low-frequency features emerge and remain relatively stable, while continued training improves generation quality by refining only the high-frequency details.

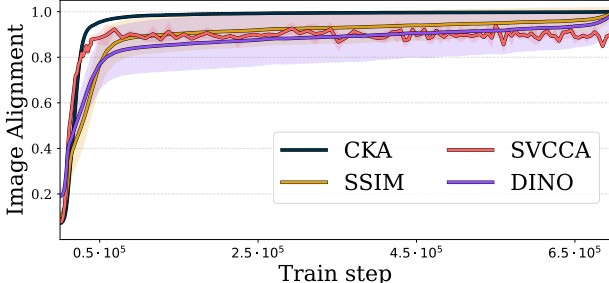

Figure 18: **Similarity of the generations sampled from the same random noise at different stages of the diffusion model's training to the final outputs for ADM-32 (CIFAR-10).** We plot CKA, SVCCA, SSIM, and DINO image alignment metrics and show that the diffusion model already learns the mapping between Gaussian noise and generations at the beginning of the training.

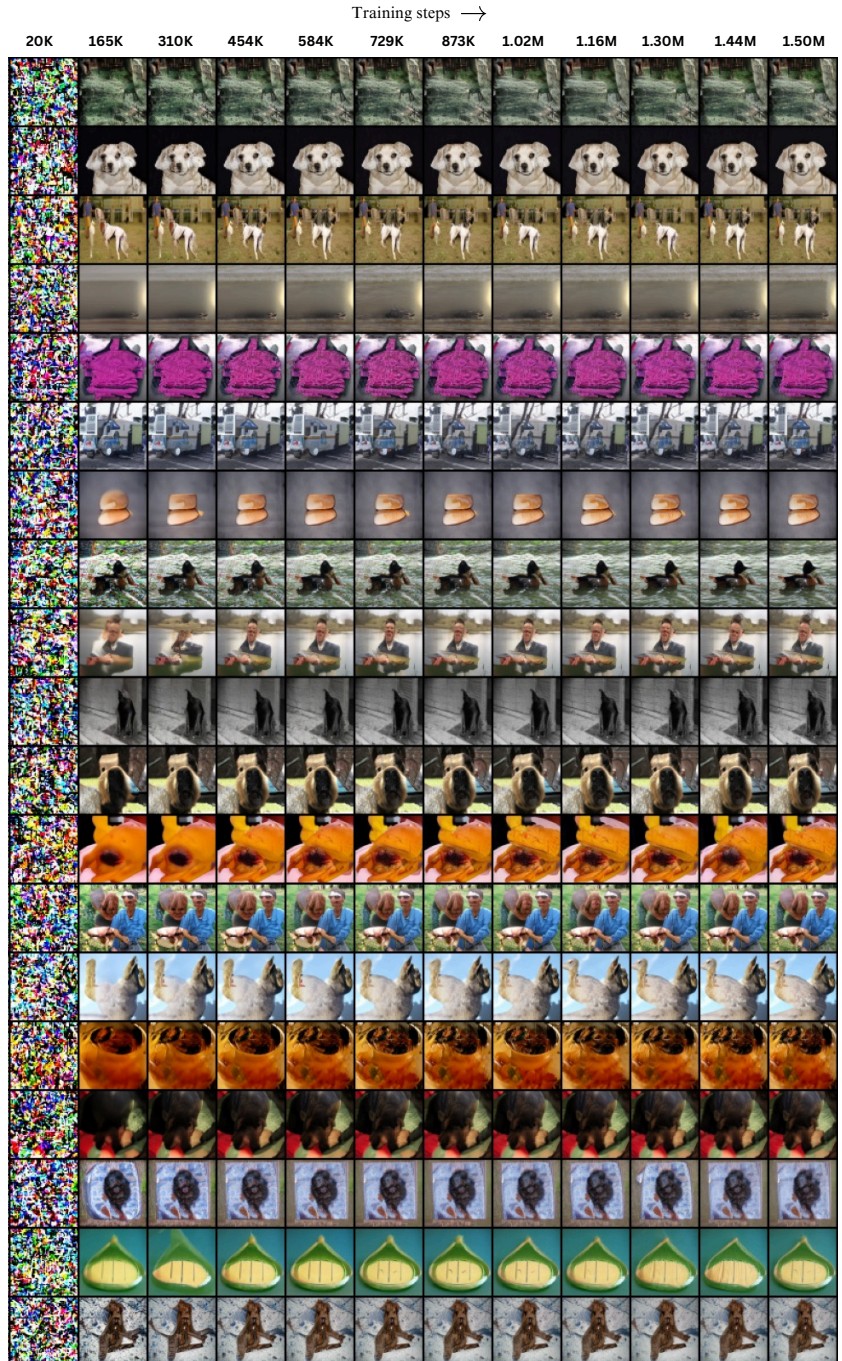

Figure 19: **Examples of images sampled using DDIM scheduler from the same noise during the training process for the ADM-64 model trained on the ImageNet dataset.**

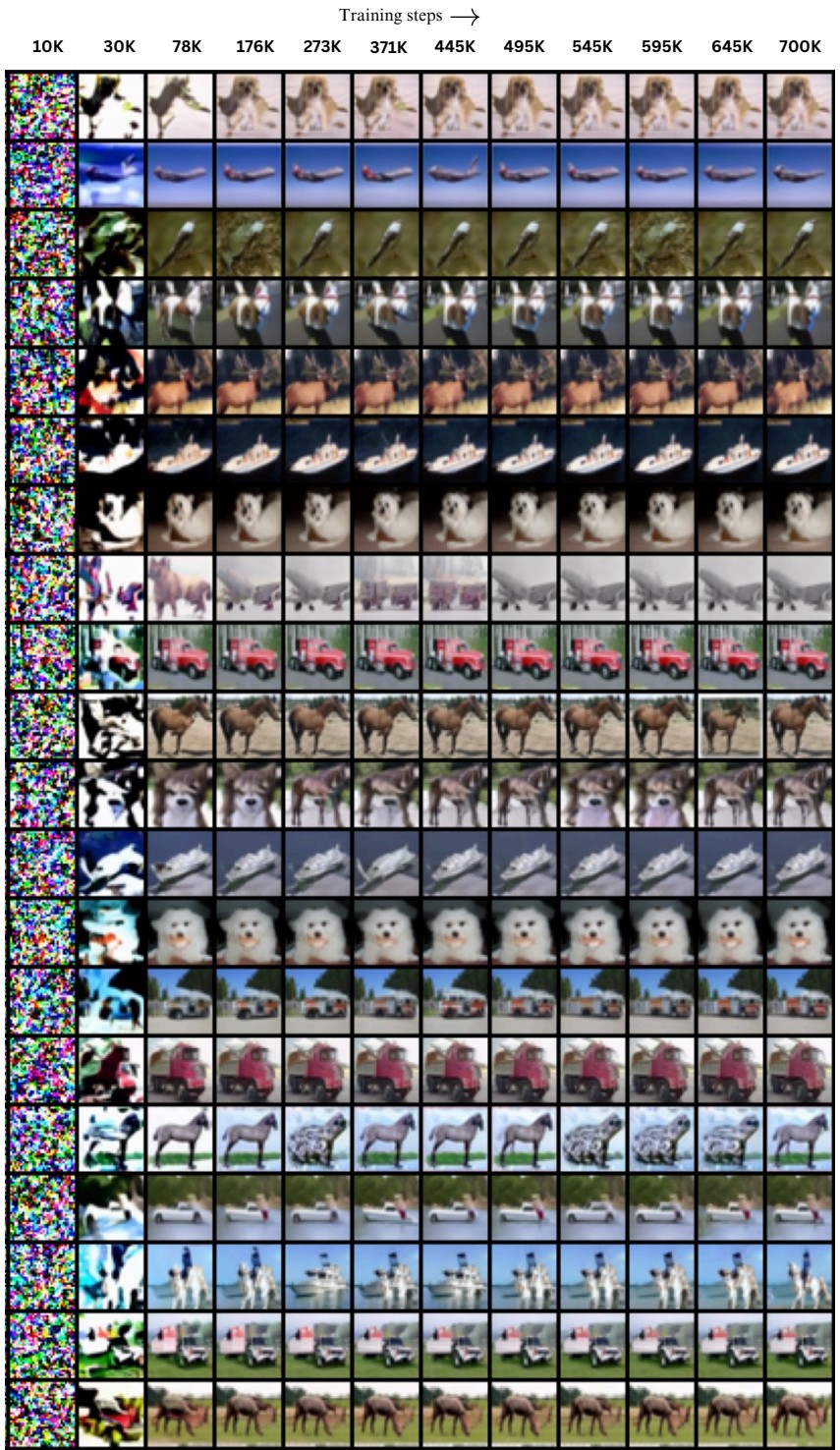

Figure 20: **Examples of images sampled using DDIM scheduler from the same noise during the training process for the ADM-32 model trained on the CIFAR-10 dataset.**

## N  PARAMETER IMPACT ANALYSIS

### N.1  NUMBER OF INVERSION STEPS

In this work, for the inversion process, we leverage the DDIM sampler with either $T = 50$ or $T = 100$ sampling steps. This choice aligns with prior works (Hong et al., 2024; Garibi et al., 2024; Kim et al., 2022b) in image edition domain, where the authors used from 50 up to 200 inversion steps as a proper balance between reconstruction quality and short algorithm runtime.

However, as described in Hong et al. (2024), the naïve DDIM inversion procedure (Song et al., 2021) can be reinterpreted as solving the forward diffusion ordinary differential equation (ODE) in reverse order (along the time axis) with Euler method. With this reformulation, the inversion is correct under the assumption that, with $dt$ being step size, noise predictions in $t$ and $t + dt$ steps are almost exact, thus works only when performing with many iterations.

Since the presence of image structures in DDIM latents can depend on the number of steps with which the inversion is performed, in Table 13 we show for ADM-64, DiT, and IF models that the observations we presented in this work generalize to cases where the number of inversion steps is several times greater (i.e., $T = 1000$, as performed during the training). Additionally, in Fig. 21 we show qualitatively by plotting the absolute error between starting Gaussian noise and DDIM latents, that also for a large number of steps, the uniform areas on the image contribute more significantly to overall inversion error.

For a more thorough analysis, in Table 14 we evaluate how our fix decorraltes latents in situation where we use $T = 1000$ inversion steps. As visible, replacing just 1 step of DDIM Inversion with forward diffusion significantly reduces correlation at minimal loss in image reconstruction.

| Object | Model | | |
|---|---|---|---|
| | **ADM-64** | **DiT** | **IF** |
| Noise $\mathbf{x_T}$ (baseline) | $0.039_{\pm.00}$ | $0.039_{\pm.00}$ | $0.039_{\pm.00}$ |
| DDIM Latent $\hat{\mathbf{x}}_{T=10}$ | $0.416_{\pm.03}$ | $0.297_{\pm.01}$ | $0.783_{\pm.01}$ |
| DDIM Latent $\hat{\mathbf{x}}_{T=25}$ | $0.242_{\pm.02}$ | $0.203_{\pm.02}$ | $0.698_{\pm.02}$ |
| DDIM Latent $\hat{\mathbf{x}}_{T=50}$ | $0.177_{\pm.02}$ | $0.144_{\pm.02}$ | $0.608_{\pm.02}$ |
| DDIM Latent $\hat{\mathbf{x}}_{T=100}$ | $0.133_{\pm.01}$ | $0.106_{\pm.02}$ | $0.500_{\pm.02}$ |
| DDIM Latent $\hat{\mathbf{x}}_{T=250}$ | $0.108_{\pm.01}$ | $0.078_{\pm.01}$ | $0.366_{\pm.02}$ |
| DDIM Latent $\hat{\mathbf{x}}_{T=500}$ | $0.100_{\pm.01}$ | $0.069_{\pm.01}$ | $0.294_{\pm.02}$ |
| DDIM Latent $\hat{\mathbf{x}}_{T=1000}$ | $0.095_{\pm.01}$ | $0.065_{\pm.01}$ | $0.249_{\pm.02}$ |

Table 13: **Latent encodings resulting from the DDIM Inversion exhibit correlations, even when the procedure is performed with a lot of steps.** By dividing latent encodings into $8 \times 8$ patches and calculating the mean of top-20 Pearson coefficients, we show that DDIM latents are correlated substantially higher than Gaussian noise, even when using $T = 1000$ inversion steps.

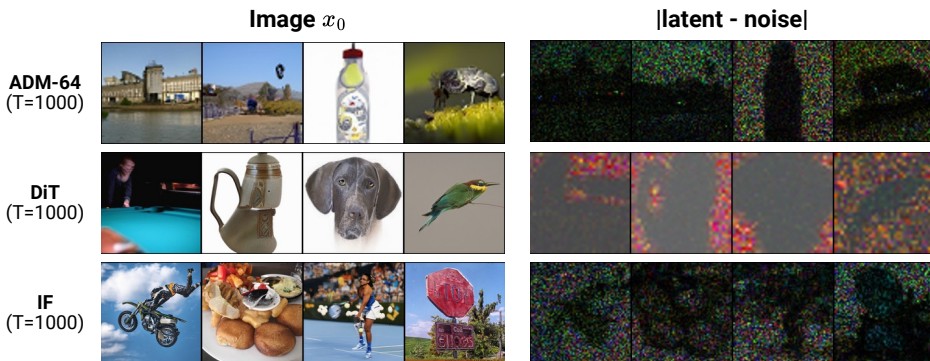

Figure 21: **Approximation errors in DDIM inversion are significantly higher for plain image surfaces than for the rest of the image.** Even when using $T = 1000$ steps, we observe image structures in DDIM latents, notably for uniform image regions.

| Model | Prior | Corr. ↓ | KL $\times 10^{-4}$ ↓ | Image Recon. ↓ |
|---|---|---|---|---|
| ADM-64 (T=1000) | Noise (upper bound) | $0.039_{\pm.00}$ | 4.832 | 0.000 |
| | DDIM Latent | $0.095_{\pm.01}$ | 47.015 | 0.014 |
| | **w/ our fix (4%)** | $0.055_{\pm.00}$ | 5.125 | 0.033 |
| | **w/ our fix (2%)** | $0.055_{\pm.00}$ | 5.363 | 0.024 |
| | **w/ our fix (1%)** | $0.055_{\pm.00}$ | 5.679 | 0.019 |
| | **w/ our fix (0.5%)** | $0.055_{\pm.00}$ | 6.057 | 0.017 |
| | **w/ our fix (0.2%)** | $0.055_{\pm.00}$ | 6.662 | 0.015 |
| | **w/ our fix (0.1%)** | $0.055_{\pm.00}$ | 7.228 | 0.014 |
| DiT (T=1000) | Noise (upper bound) | $0.039_{\pm.00}$ | 4.832 | 0.000 |
| | DDIM Latent | $0.065_{\pm.01}$ | 17.931 | 0.009 |
| | **w/ our fix (4%)** | $0.057_{\pm.00}$ | 5.233 | 0.060 |
| | **w/ our fix (2%)** | $0.057_{\pm.00}$ | 5.071 | 0.041 |
| | **w/ our fix (1%)** | $0.057_{\pm.00}$ | 5.850 | 0.029 |
| | **w/ our fix (0.5%)** | $0.058_{\pm.00}$ | 7.029 | 0.021 |
| | **w/ our fix (0.2%)** | $0.058_{\pm.00}$ | 8.045 | 0.016 |
| | **w/ our fix (0.1%)** | $0.058_{\pm.00}$ | 8.409 | 0.014 |
| IF (T=1000) | Noise (upper bound) | $0.039_{\pm.00}$ | 4.832 | 0.000 |
| | DDIM Latent | $0.249_{\pm.02}$ | 63.962 | 0.044 |
| | **w/ our fix (4%)** | $0.055_{\pm.00}$ | 5.024 | 0.037 |
| | **w/ our fix (2%)** | $0.055_{\pm.00}$ | 5.215 | 0.031 |
| | **w/ our fix (1%)** | $0.055_{\pm.00}$ | 5.706 | 0.027 |
| | **w/ our fix (0.5%)** | $0.055_{\pm.00}$ | 6.117 | 0.026 |
| | **w/ our fix (0.2%)** | $0.055_{\pm.00}$ | 5.562 | 0.025 |
| | **w/ our fix (0.1%)** | $0.056_{\pm.00}$ | 5.349 | 0.024 |

Table 14: **Latent correlations, KL divergence to random Gaussian noise, and image reconstruction error across models (ADM-64, DiT, IF) when using $T = 1000$ inversion steps.** We show that using our simple fix in just one step of inversion process significantly decorellates DDIM latents with minimal loss in image reconstruction performance.

### N.2 PERCENTAGE OF INVERSION STEPS REPLACED

The fix to DDIM Inversion algorithm, proposed in this work, namely replacing neural network predictions with random Gaussian Noise, implies the trade-off between preserving the original image information and improving the latents' editability. In this section, we present how the number of inversion steps substituted with forward step, impacts the image reconstruction error (MAE, LPIPS, and SSIM metrics) and latent editability (correlations and KL Divergence from $\mathcal{N}(0; \mathcal{I})$) for: IF (Table 15), DiT (Table 16), and SDXL (Table 17) models. We observe that replacing only the first $4\%$ of inversion steps with forward diffusion improves latent normality to the level of Gaussian noise ($100\%$ of steps), while increases reconstruction error only slightly. In Fig. 22, we show some failure cases when replacing $10\%$ or $20\%$ of the first steps can introduce significant changes to images.

| # Steps Replaced | Image Reconstruction | | | Latent Normality | |
|---|---|---|---|---|---|
| (Percentage) | MAE ↓ | LPIPS ↓ | SSIM ↑ | Correlation ↓ | KL Div. $\times 10^2$ ↓ |
| 0 (0%) | 0.073 | 0.030 | 0.878 | 0.643 | 60.449 |
| 1 (2%) | 0.069 | 0.037 | 0.854 | 0.057 | 0.934 |
| 2 (4%) | 0.071 | 0.038 | 0.845 | 0.050 | 0.352 |
| 3 (6%) | 0.074 | 0.040 | 0.830 | 0.050 | 0.346 |
| 4 (8%) | 0.078 | 0.043 | 0.813 | 0.049 | 0.341 |
| 5 (10%) | 0.082 | 0.047 | 0.796 | 0.049 | 0.338 |
| 10 (20%) | 0.099 | 0.066 | 0.713 | 0.049 | 0.344 |
| 20 (40%) | 0.131 | 0.113 | 0.556 | 0.049 | 0.360 |
| 30 (60%) | 0.169 | 0.179 | 0.394 | 0.049 | 0.367 |
| 40 (80%) | 0.233 | 0.279 | 0.204 | 0.049 | 0.370 |
| 50 (100%) | 0.487 | 0.437 | 0.009 | 0.049 | 0.374 |

Table 15: **Impact of percentage of inversion steps replaced with forward diffusion on reconstruction quality and latent normality for DeepFloyd IF.**

| # Steps Replaced | Image Reconstruction | | | Latent Normality | |
|---|---|---|---|---|---|
| (Percentage) | MAE ↓ | LPIPS ↓ | SSIM ↑ | Correlation ↓ | KL Div. $\times 10^2$ ↓ |
| 0 (0%) | 0.052 | 0.063 | 0.839 | 0.159 | 1.118 |
| 1 (2%) | 0.070 | 0.097 | 0.741 | 0.038 | 0.011 |
| 2 (4%) | 0.085 | 0.125 | 0.658 | 0.036 | 0.010 |
| 3 (6%) | 0.097 | 0.151 | 0.594 | 0.036 | 0.023 |
| 4 (8%) | 0.107 | 0.173 | 0.544 | 0.037 | 0.036 |
| 5 (10%) | 0.116 | 0.195 | 0.505 | 0.037 | 0.041 |
| 10 (20%) | 0.154 | 0.280 | 0.375 | 0.038 | 0.040 |
| 20 (40%) | 0.231 | 0.437 | 0.235 | 0.037 | 0.020 |
| 30 (60%) | 0.353 | 0.595 | 0.145 | 0.037 | 0.017 |
| 40 (80%) | 0.521 | 0.710 | 0.071 | 0.037 | 0.017 |
| 50 (100%) | 0.628 | 0.750 | 0.029 | 0.037 | 0.018 |

Table 16: **Impact of percentage of inversion steps replaced with forward diffusion on reconstruction quality and latent normality for Diffusion Transformer (DiT).**

| # Steps Replaced | Image Reconstruction | | | Latent Normality | |
|---|---|---|---|---|---|
| (Percentage) | MAE ↓ | LPIPS ↓ | SSIM ↑ | Correlation ↓ | KL Div. $\times 10^2$ ↓ |
| 0 (0%) | 0.027 | 0.099 | 0.814 | 0.166 | 0.800 |
| 1 (2%) | 0.029 | 0.106 | 0.790 | 0.151 | 0.600 |
| 2 (4%) | 0.035 | 0.137 | 0.716 | 0.120 | 0.300 |
| 3 (6%) | 0.038 | 0.155 | 0.685 | 0.117 | 0.300 |
| 4 (8%) | 0.041 | 0.171 | 0.663 | 0.116 | 0.300 |
| 5 (10%) | 0.043 | 0.183 | 0.646 | 0.116 | 0.200 |
| 10 (20%) | 0.051 | 0.230 | 0.588 | 0.115 | 0.200 |
| 20 (40%) | 0.065 | 0.313 | 0.515 | 0.115 | 0.200 |
| 30 (60%) | 0.082 | 0.389 | 0.460 | 0.115 | 0.200 |
| 40 (80%) | 0.112 | 0.464 | 0.464 | 0.115 | 0.200 |
| 50 (100%) | 0.159 | 0.541 | 0.541 | 0.115 | 0.200 |

Table 17: **Impact of percentage of inversion steps replaced with forward diffusion on reconstruction quality and latent normality for Stable Diffusion XL.**

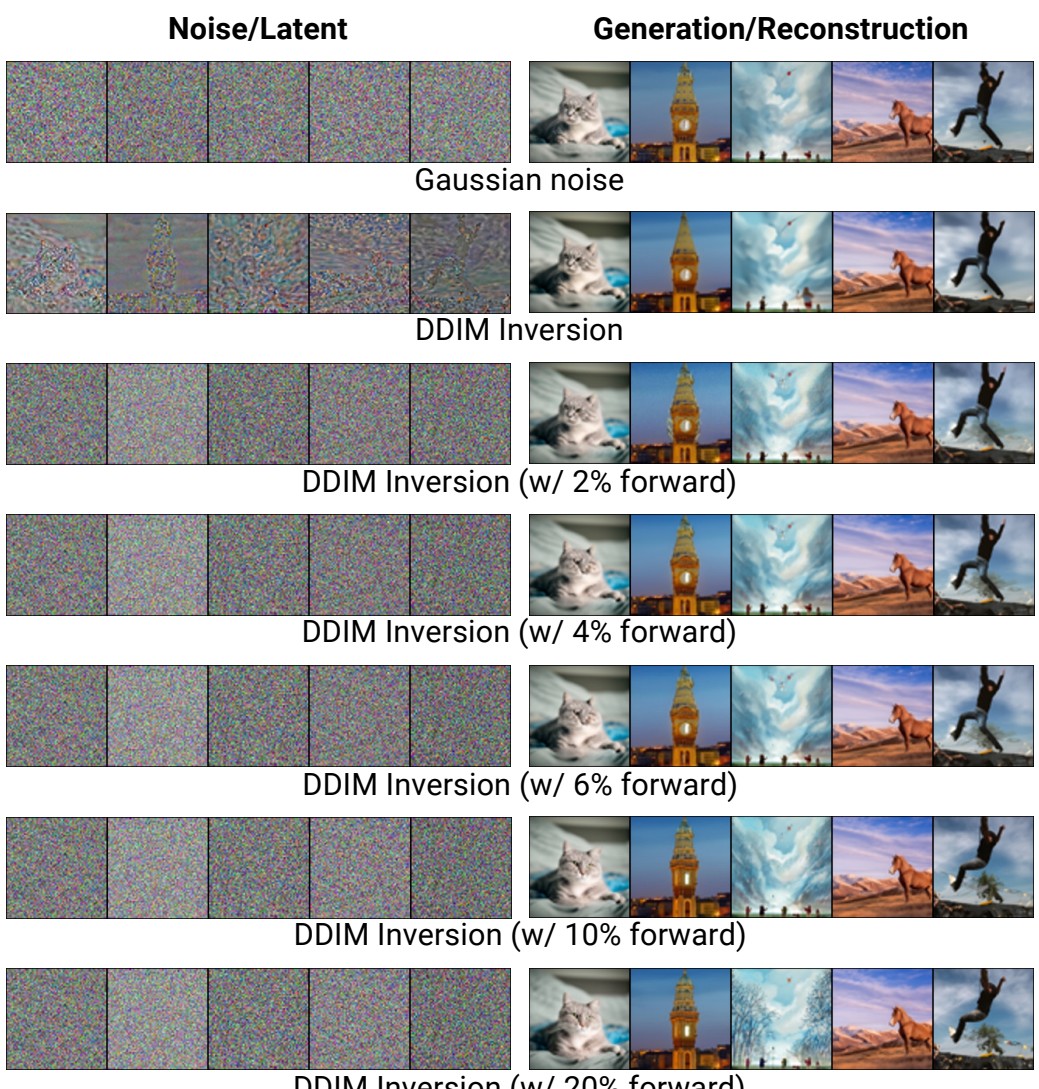

Figure 22: **Replacing first DDIM inversion steps with forward diffusion increases latents editability at the cost of a higher reconstruction error.** By replacing from 2% up to 4% of steps, we obtain reasonable image reconstructions while removing correlations from latents.

### N.3 GUIDANCE SCALE

During experiments, we fix the guidance scale to $w = 1$ to ensure that our analysis focuses solely on the DDIM approximation error (Eq. (3)). As described in Ju et al. (2024), prior works typically employ guidance scale between $1.0$ (the most common choice) and $3.0$ during inversion, as using $w > 3$ often results in drastically worse image reconstructions.

In this section, we evaluate how the proposed fix improves upon Naïve DDIM Inversion when a higher guidance scale $w \in \{1, 2, 3, 4, 5\}$ is applied both during inversion and reconstruction. In Table 18, we present result of this experiment with Stable Diffusion XL (Podell et al., 2024). For scenarios with a higher guidance scale, our simple fix, similarly to $w = 1$, reduces correlations and improves editability when comparing with Naive DDIM Inversion. Additionally, we observe that, when guidance is applied, our fix improves image reconstruction error (measured with LPIPS). We hypothesize that amplifying the inversion error in Naïve DDIM with a higher guidance scale leads to latents that are useless for image reconstruction. In such a case, replacing the first steps with random Noise may lead to more preferable reconstructions. In Fig. 23 and Fig. 24, we present qualitative comparison in image reconstruction between Naïve DDIM Inversion and our approach.

| Guidance Scale $w$ | Method | LPIPS ↓ | Latent Corr. ↓ | CLIP Alignment (Edit Prompt) ↑ |
|---|---|---|---|---|
| 1.0 | DDIM Inv. | 0.100 | 0.166 | 0.695 |
| | w/ ours (4%) | 0.137 | 0.120 | 0.722 |
| | Δ | **+0.037** | **-0.046** | **+0.027** |
| 2.0 | DDIM Inv. | 0.199 | 0.170 | 0.779 |
| | w/ ours (4%) | 0.179 | 0.120 | 0.807 |
| | Δ | **-0.020** | **-0.050** | **+0.028** |
| 3.0 | DDIM Inv. | 0.390 | 0.171 | 0.764 |
| | w/ ours (4%) | 0.267 | 0.121 | 0.815 |
| | Δ | **-0.123** | **-0.050** | **+0.051** |
| 4.0 | DDIM Inv. | 0.525 | 0.172 | 0.725 |
| | w/ ours (4%) | 0.372 | 0.121 | 0.800 |
| | Δ | **-0.153** | **-0.051** | **+0.075** |
| 5.0 | DDIM Inv. | 0.582 | 0.174 | 0.687 |
| | w/ ours (4%) | 0.452 | 0.121 | 0.770 |
| | Δ | **-0.130** | **-0.053** | **+0.083** |

Table 18: **Performance in image reconstruction (LPIPS), inverted latent normality (correlations), and text alignment to edit prompt for different values of guidance scale**. We show that our forward step replacement (4%) improves DDIM Inversion algorithm.

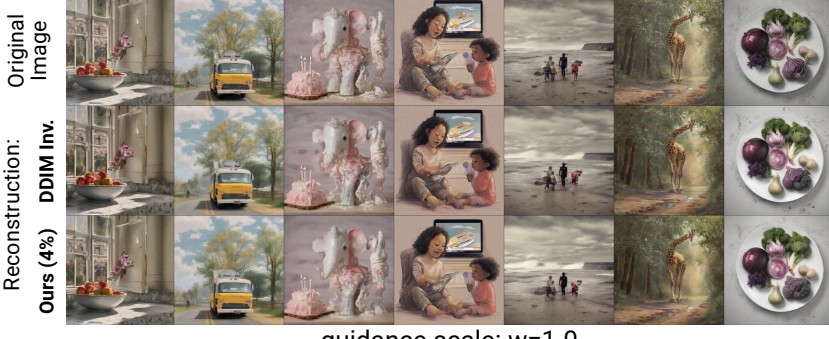

guidance scale: w=1.0

Figure 23: **Examples of image reconstruction with Naïve DDIM Inversion and DDIM Inversion incorporating our fix (forward $4\%$) for guidance scale $w = 1$.** Examples generated with Stable Diffusion XL.

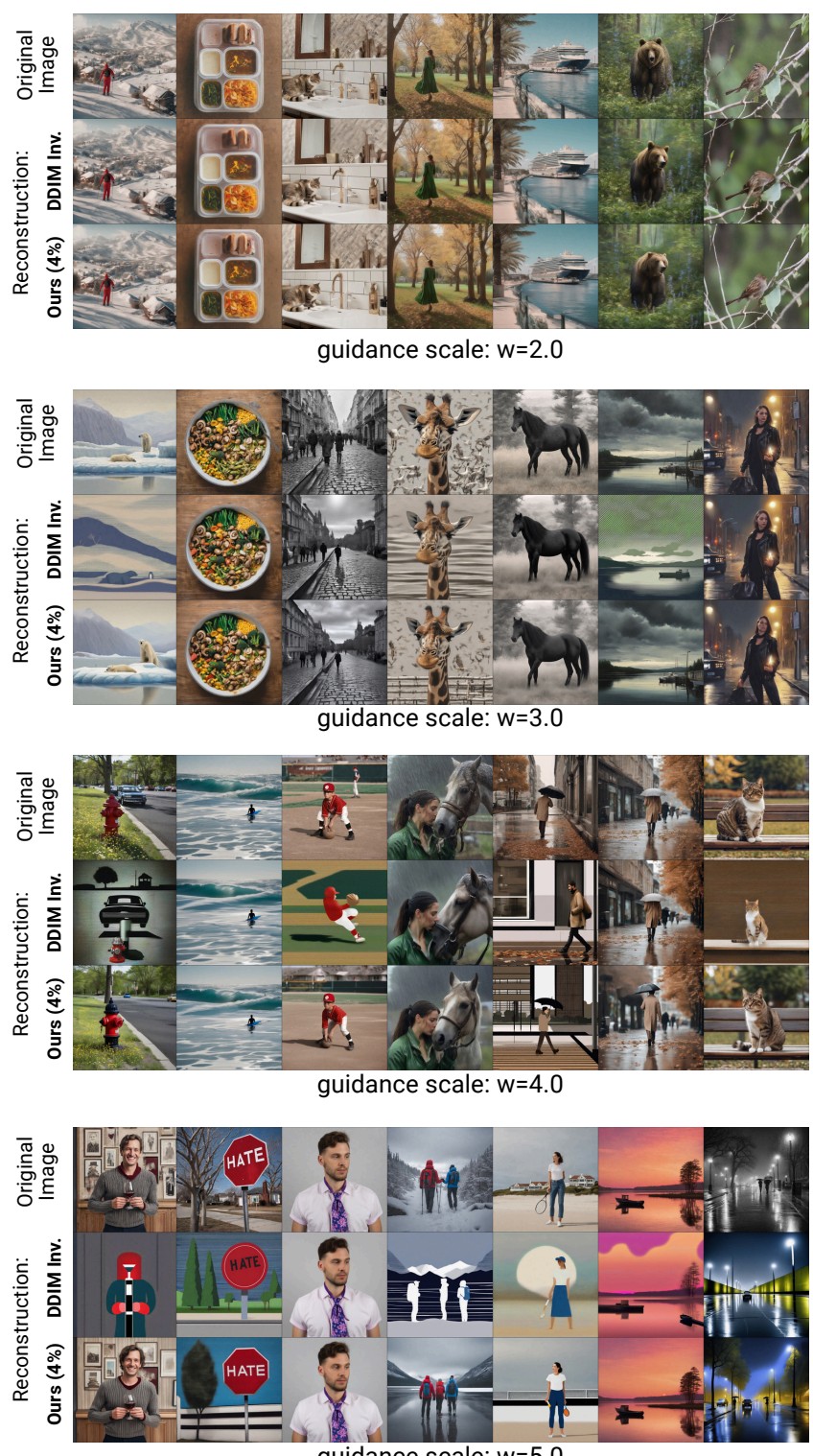

Figure 24: **Comparison of image reconstruction examples with Naïve DDIM Inversion and DDIM Inversion incorporating our fix (forward** $4\%$**) across different values of guidance scale** $w \in \{2, 3, 4, 5\}$**.** Examples generated with Stable Diffusion XL.

# O  QUALITATIVE EXAMPLES

## O.1  IMAGE INTERPOLATION

In Section 4, we presented that interpolating DDIM latents with SLERP (Shoemake, 1985) leads to a decrease in image quality and diversity when compared to Gaussian noise. In Fig. 25, we qualitatively compare our fix for removing correlations in latent encodings with naïve DDIM inversion in the task of image interpolation.

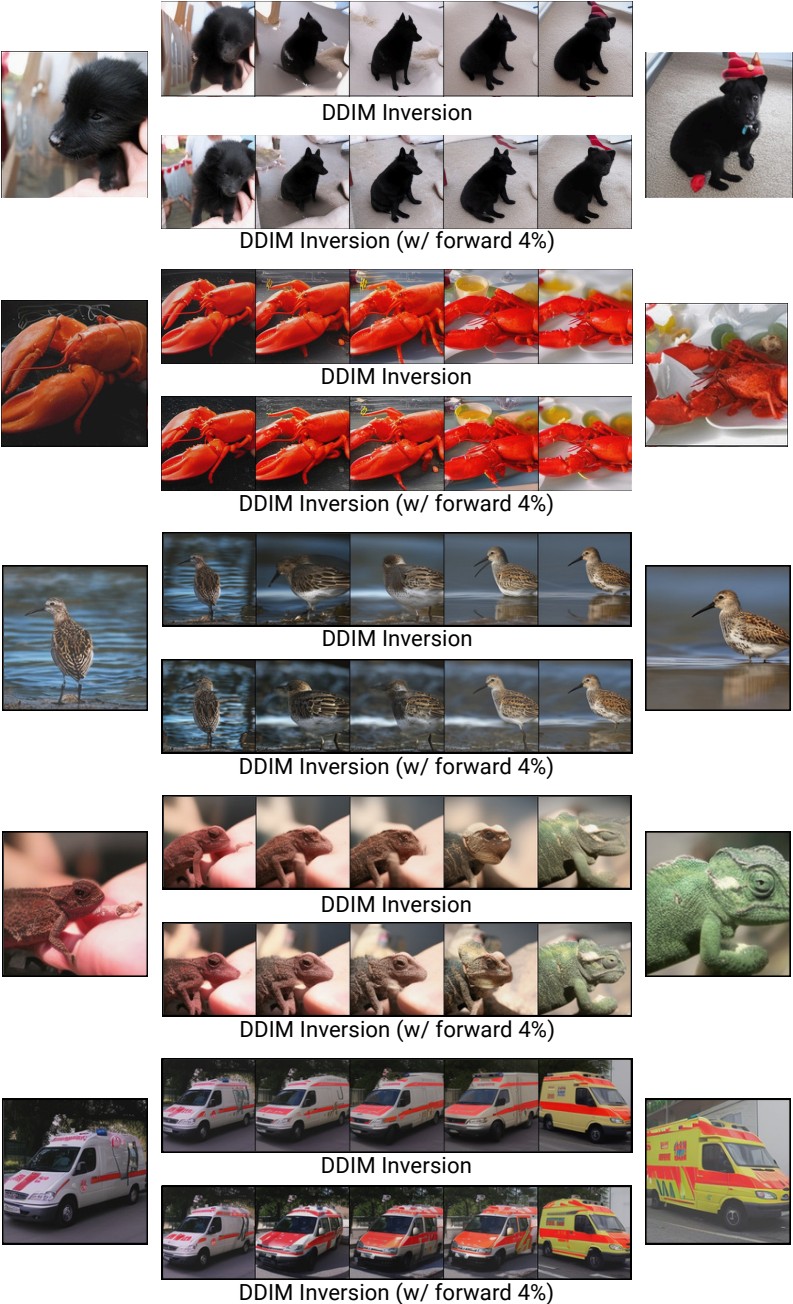

Figure 25: **Qualitative comparison of images generated from interpolated latents produced with DDIM Inversion and our fix.** Contrary to naïve DDIM inversion, the proposed solution enables generating high-quality objects with pixel-diverse backgrounds.

## O.2 RECONSTRUCTIONS OF REAL IMAGES

In Fig. 26, we present a qualitative comparison for reconstructions of real images from the Style-Drop (Sohn et al., 2023) dataset. We observe that DDIM Inversion with our fix sometimes provides imperfect image reconstructions. However, those failures are also observable with vanilla DDIM Inversion, indicating that they stem from DDIM approximation error itself, not from our replacement.

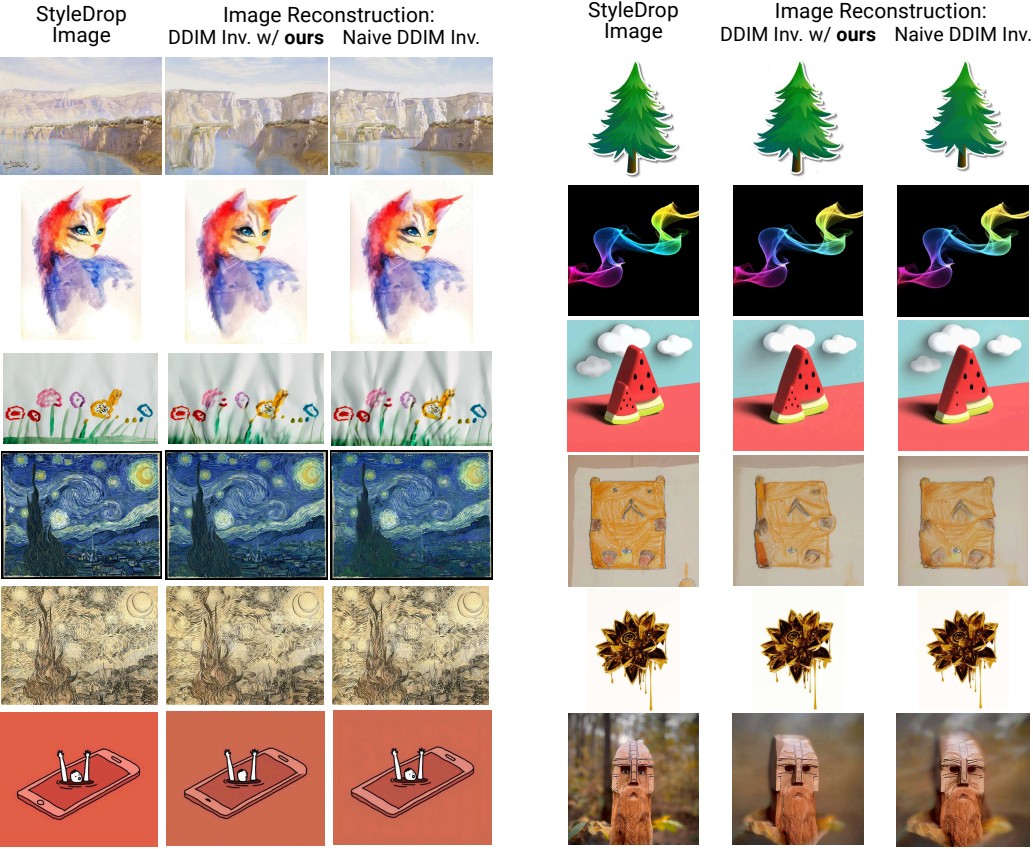

Figure 26: **Examples of reconstructions of real images from the StyleDrop (Sohn et al., 2023) dataset with Naive DDIM Inversion and DDIM with our fix (forward diffusion in $4\%$ of steps).** Inversion process is run with $T = 50$ steps and guidance scale $w = 1.0$. Reconstructions generated with Stable Diffusion XL.

## O.3 STOCHASTIC IMAGE EDITING

In this work, we propose a solution for decorrelating latent encodings resulting from DDIM inversion by replacing its first steps with the forward diffusion. As presented in Algorithm 1, the forward diffusion process involves sampling random Gaussian noise $\tilde{\epsilon} \sim \mathcal{N}(0, \mathcal{I})$ and interpolating it with the input image. Due to the fact that, when we replace a small fraction of steps ($2 - 4\%$), the change in image reconstruction error is insignificant, the use of different noises $\tilde{\epsilon}$ (in practice, sampled with different seeds) allows stochastic image editing, i.e. generating different manipulations of the input image, a feature not naturally available with DDIM inversion. In Fig. 27, we present examples of editing **real images** from the **ImageNet-R-TI2I** dataset (which we annotate using GPT-4o) with the IF model, showing various semantically correct modifications of the same image.

As preserving original image structure during editing is stated as a more difficult task for real images than the one naturally generated by the diffusion model, we follow Hertz et al. (2022) by, first, denoising latent encodings with source prompt (the one used during inversion), and, after $6\%$ of the steps, using target prompt as conditioning. The examples presented in Fig. 27 indicate that our fix (1) enables stochastic editing of images and (2) enables image manipulations in plain image regions, contrary to editing with naïve DDIM latents.

**Input image**  **Our editing** (varying forward diffusion seed)  **DDIM Inv. editing**

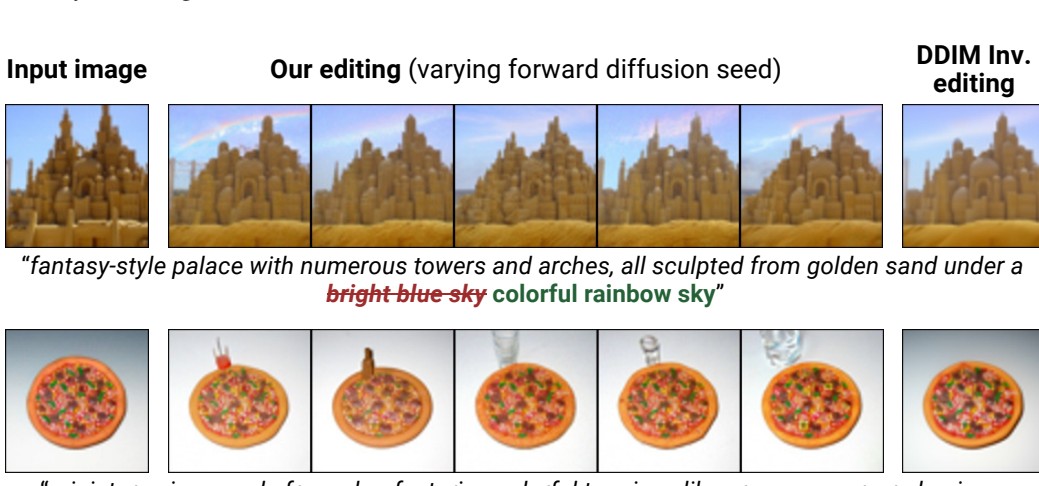

"*fantasy-style palace with numerous towers and arches, all sculpted from golden sand under a* ~~*bright blue sky*~~ **colorful rainbow sky**"

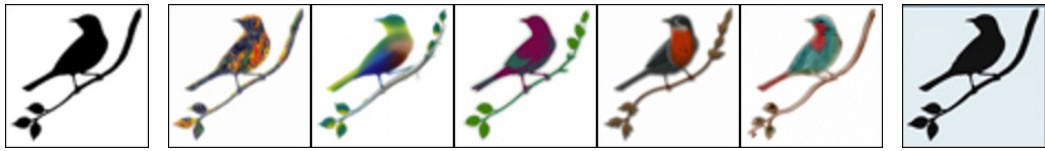

"*miniature pizza made from clay, featuring colorful toppings like green peppers, red onions, and brown sausage pieces on a bright red sauce base. the pizza is circular with a thick crust* ~~*on a white-gray gradient background*~~ **with a glass of drink**"

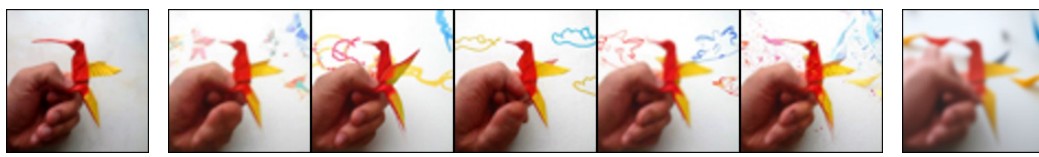

"*simple* ~~**black**~~ **colorful** *silhouette of a bird on a thin, curved branch with a few leaves, set against a plain white background, the bird faces to the right*"

"*a close-up photo of a hand holding a colorful origami hummingbird made from red and yellow paper,* ~~**set against a plain light**~~ **drawings in the** background"

Figure 27: **Replacing first DDIM inversion steps with forward diffusion enables stochastic image editing, resulting in multiple semantically correct manipulations of the same input image.** Contrary to DDIM Inversion, editing with latents produced by the solution introduced in this work enables image manipulations in uniform input image areas.

## O.4 REAL IMAGE EDITING WITH MASACTRL

In this section, we present examples for editing real images from the PIEBench dataset (Ju et al., 2024) when our inversion method is combined with the MasaCtrl (Cao et al., 2023) editing engine. In Figs. 28 to 30, we qualitatively compare with Naïve DDIM Inversion across several editing tasks: **object replacement**, **attribute editing**, and **object removal**. We present that replacing first 4% of inversion steps with forward diffusion leads to more successful edits in prompt adherence, while not observing degradation in consistency to input images.

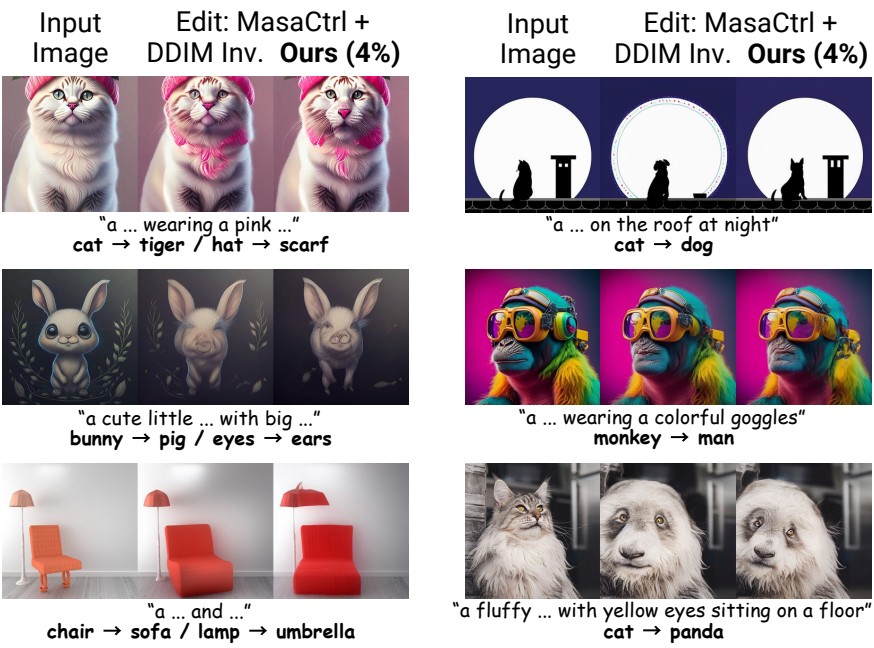

Figure 28: **Object replacement on real images with MasaCtrl.** Comparison for Naïve DDIM Inversion and DDIM with our fix (forward diffusion in 4% of steps). Model: Stable Diffusion 1.4.

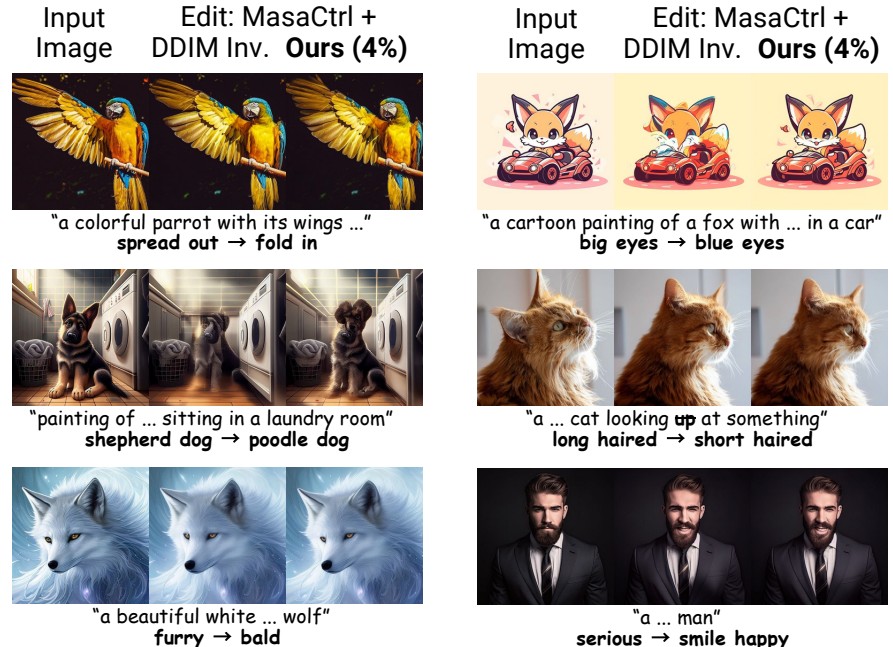

Figure 29: **Attribute editing on real images with MasaCtrl.** Comparison for Naïve DDIM Inversion and DDIM with our fix (forward diffusion in 4% of steps). Model: Stable Diffusion 1.4.

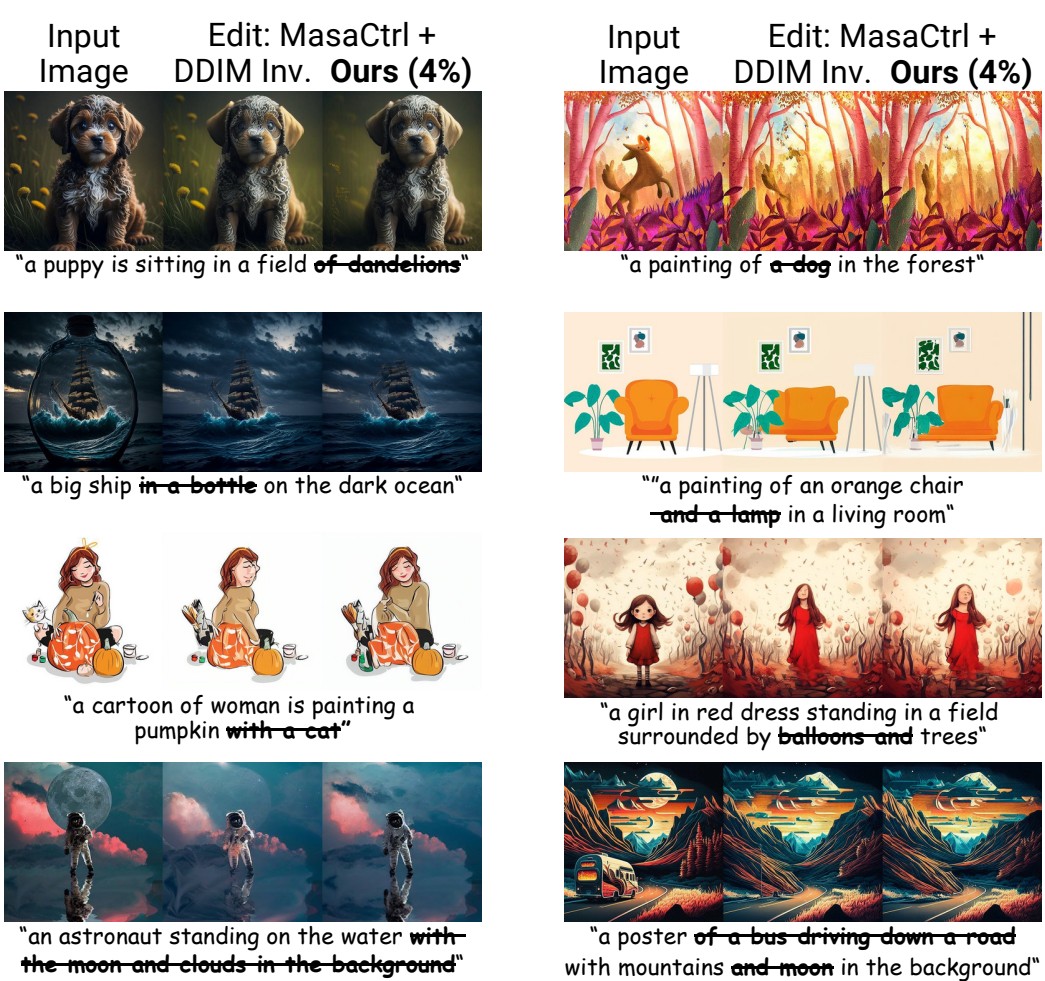

Figure 30: **Object removal on real images with MasaCtrl.** Qualitative comparison for Naïve DDIM Inversion and DDIM with our fix (forward diffusion in 4% of steps). Examples generated with Stable Diffusion 1.4.

## O.5 STYLE TRANSFER WITH STYLEALIGNED

In Figs. 31 and 32, we present a qualitative comparison of Naïve DDIM Inversion and our approach when combined with StyleAligned (Hertz et al., 2024) for the task of Style Transfer. Examples have been generated with Stable Diffusion XL using the same hyperparameters for both settings on the StyleDrop dataset (Sohn et al., 2023). We observe that replacing the first steps of DDIM Inversion with forward diffusion enables better prompt-adherence for generations.

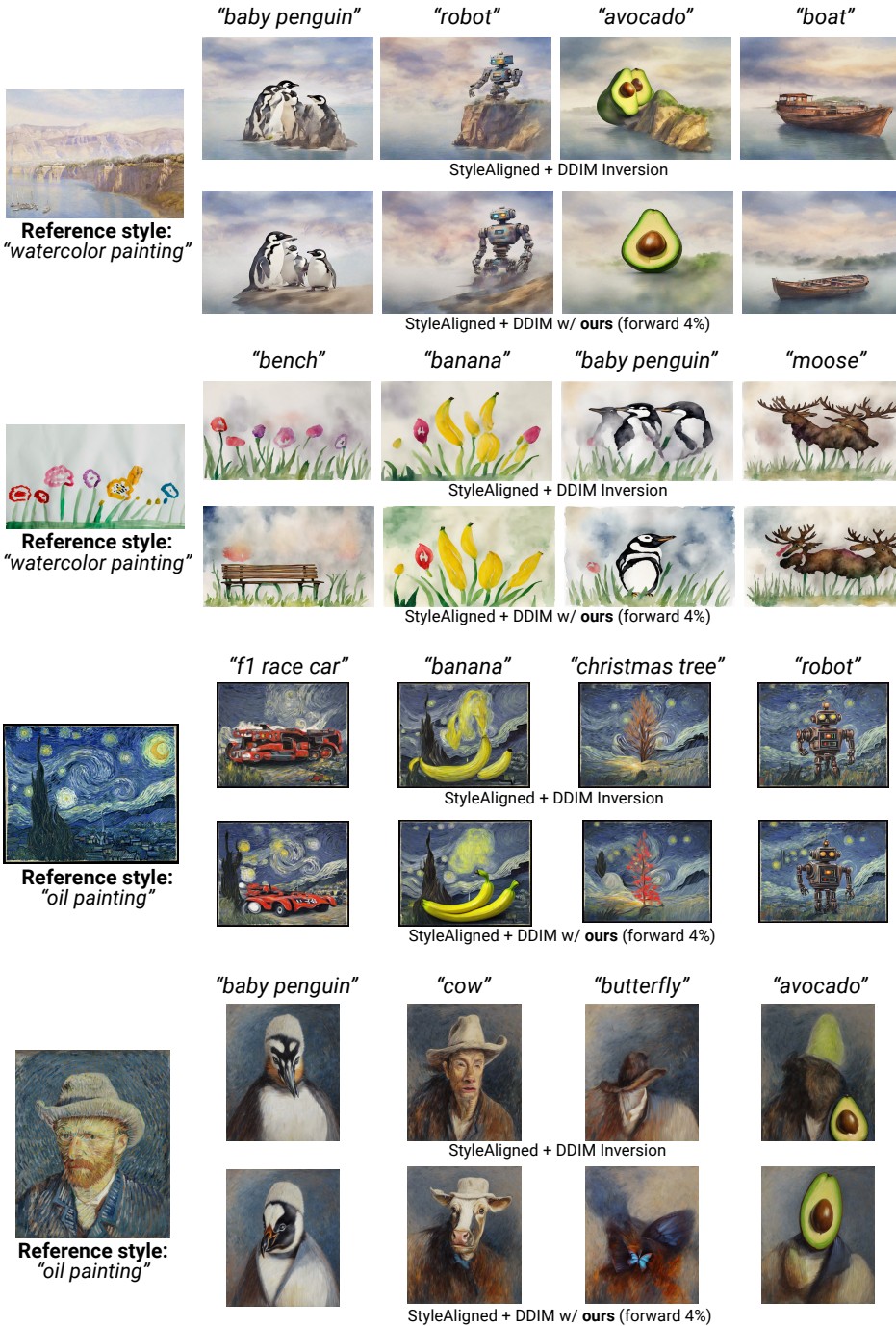

Figure 31: **Examples of style transfers from real images from the StyleDrop (Sohn et al., 2023) dataset.** Comparison for Naive DDIM Inversion and DDIM with our fix (4% of steps replaced).

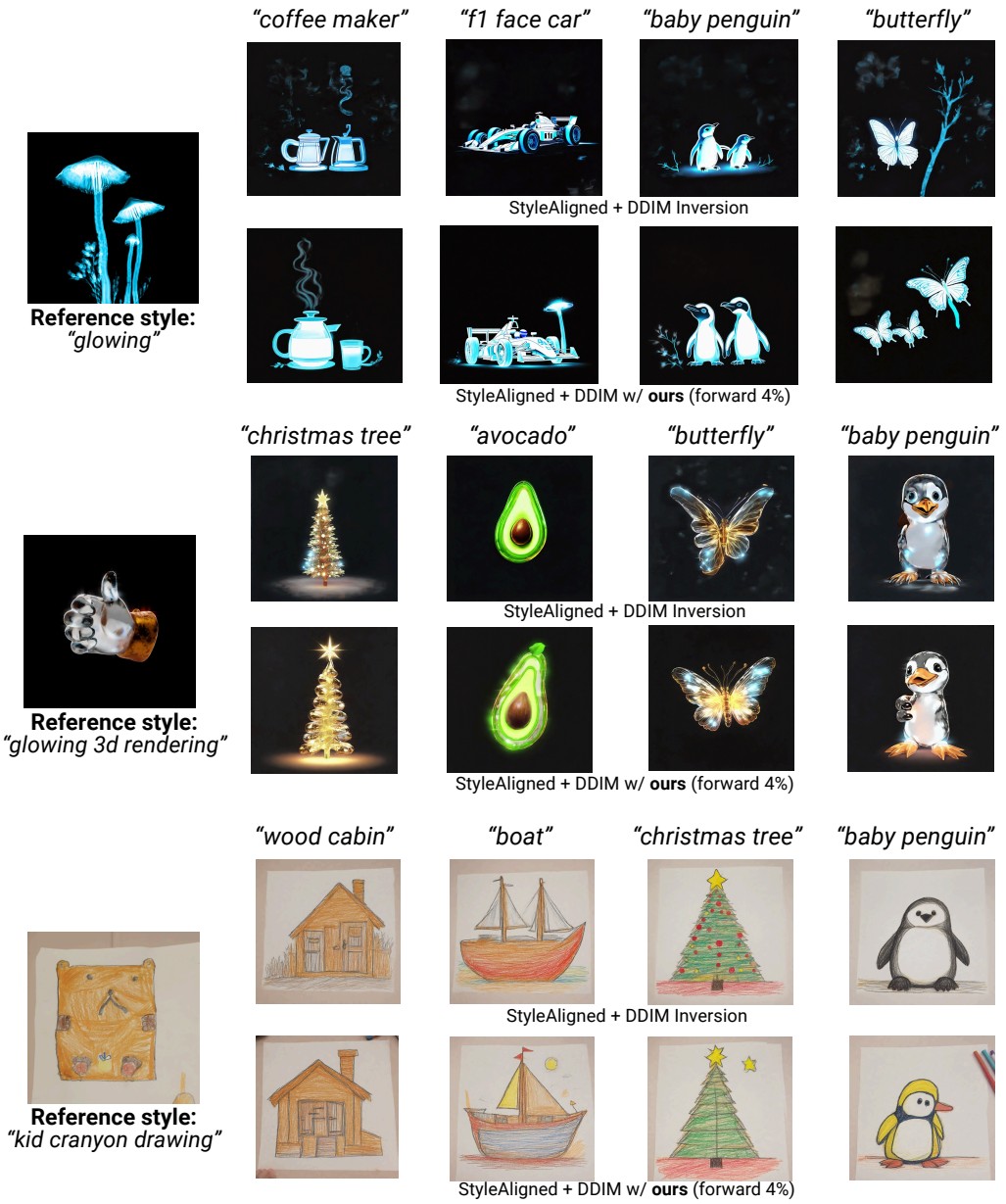

Figure 32: **Examples of style transfers from real images from the StyleDrop (Sohn et al., 2023) dataset.** Comparison for Naive DDIM Inversion and DDIM with our fix (forward diffusion in 4% of steps). Examples generated with Stable Diffusion XL.

# P   LATENT CORRELATIONS IN FLOW MATCHING MODELS

In this section, we analyze if, similarly to latents produced with DDIM inversion in Diffusion Models, inversion procedure with Flow Matching models leads to correlations.

The inversion procedure can be incorporated into Flow Matching (FM) models (Lipman et al., 2023; Liu et al., 2023), e.g., for image editing (Avrahami et al., 2025; Kulikov et al., 2025; Rout et al., 2025). The generative process of FMs is defined as an ordinary differential equation (ODE) over time $t \in [0, 1]$ with time-dependent velocity field $V$:

$$dz_t = V(z_t, t)dt. \tag{16}$$

Commonly, this ODE, given an initial condition $z_1 \sim \mathcal{N}(0; \mathcal{I})$, is solved numerically with Euler method, leading to iterative sampling process $t \in \{T, T-1, \ldots, 1\}$, defined as

$$z_{t-1} = z_t + (\sigma_{t-1} - \sigma_t) \cdot \nu_\theta(z_t, t), \tag{17}$$

with $\nu_\theta$ being a neural network parametrizing the continuous velocity field leading to clean images $z_0$ and $\sigma_t$ being a noise schedule.

The inverse step, as described in Avrahami et al. (2025), can be expressed as

$$z_t = z_{t-1} + (\sigma_t - \sigma_{t-1}) \cdot \nu_\theta(\mathbf{z_{t-1}}, t), \tag{18}$$

with an assumption that locally $\nu_\theta(z_t, t) \approx \nu_\theta(z_{t-1}, t)$. We refer to this formulation as ODE Inversion.

As the approximation relies on a similar assumption as in the case of DDIM (Eq. (3)), we analyze if the ODE Inversion, similarly, induces correlation patterns in output latents. In Table 19, we report image reconstruction error, editing textual alignment (CLIP Similarity to edit prompt and Directional Similarity (Gal et al., 2022)), and metrics validating the latents' normality. We employ FLUX.1 (Labs, 2024) model with $T = 50$ inversion and sampling steps. We present that the latents resulting from the ODE Inversion algorithm, similarly to the case of DDIM latents, exhibit correlations and visible deviation from the Gaussian distribution. Importantly, these deviations, when compared to using original noise, lead to a significant decrease in prompt alignment when starting the generation process with an editing prompt. Additionally, in Table 19, we compare original noise and ODE Inversion latent diversity for plain and non-plain input image pixel regions. Although not as significantly visible as in DDIM latents, ODE Inversion outputs as well tend to be more erroneous for plain image pixels and less diverse in those areas.

Finally, in Fig. 33, we present qualitative examples for image reconstructions and latent correlation when ODE Inversion is performed. As visible, after decoding with FLUX's decoder, ODE Inversion latents exhibit correlations in locations that represent smooth pixel areas of images. Additionally, we plot the absolute error between original Gaussian Noise and ODE Inversion latents after applying PCA for dimensional reduction (as FLUX operates in 16-channel latent space).

| Metric | | Gaussian Noise | ODE Inv. Latent |
|---|---|---|---|
| **Image Reconstr.** | MAE ↓ | 0.00 | 0.05 |
| | LPIPS ↓ | 0.00 | 0.16 |
| **CLIP Text Alignment** | Edit prompt ↑ | 81.49 | 56.40 |
| | Directional Sim. ↑ | 87.94 | 55.32 |
| **Normality** | Correlation ↓ | 0.14 | 0.27 |
| | KL Div. $\times 10^{-2}$ ↓ | 0.20 | 3.80 |
| **Noise Error** | Plain pixels | 0.00 | 0.31 |
| | Non-plain pixels | 0.00 | 0.26 |
| **Variance** | Plain pixels | 0.98 | 0.94 |
| | Non-plain pixels | 1.01 | 1.03 |

Table 19: **Comparison between original Gaussian Noise and latents resulting from ODE Inversion process with FLUX.1 model.** We show that ODE inversions are more correlated than Gaussian and significantly deviate from normal distribution. This leads to worse text alignment during editing.

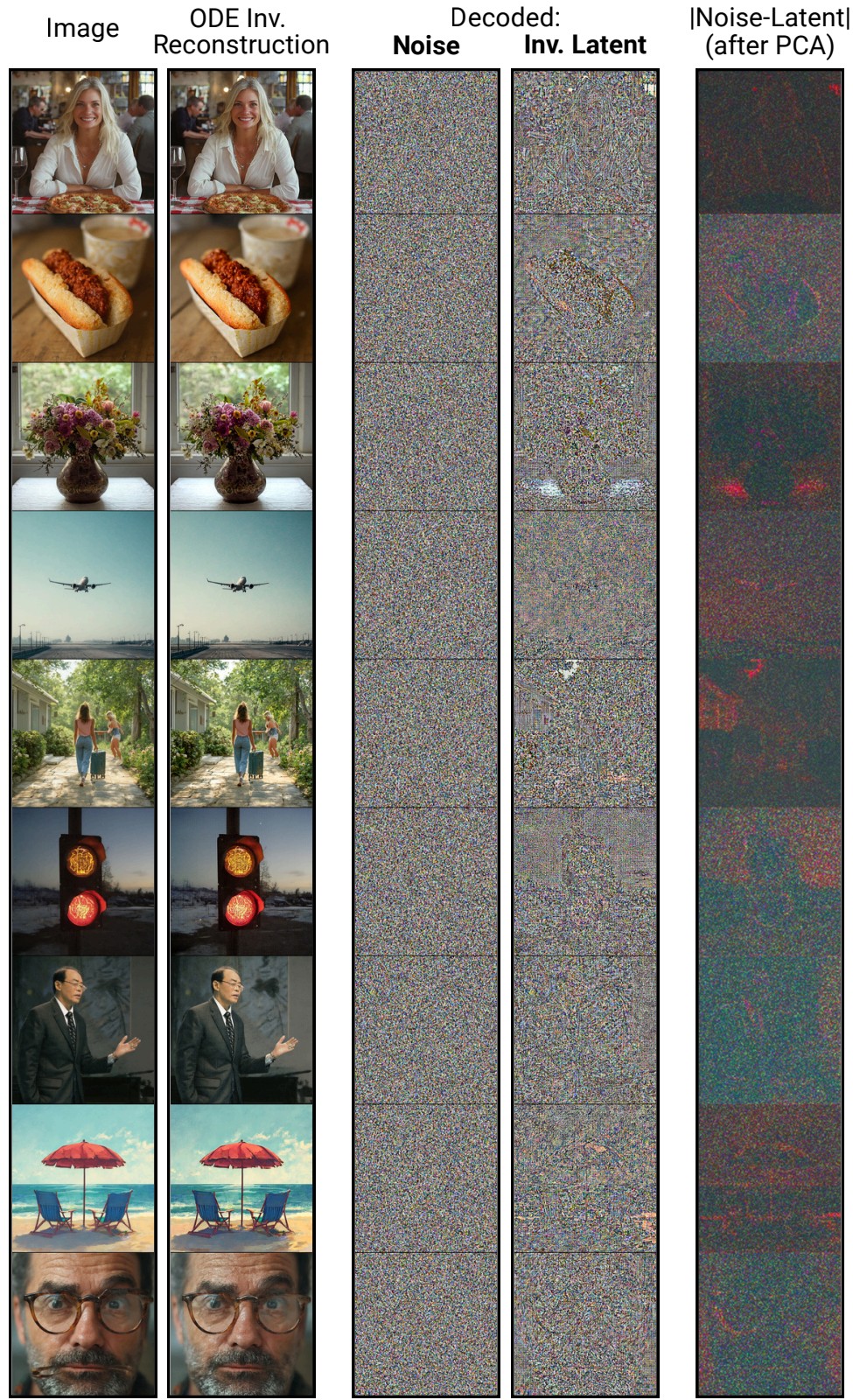

Figure 33: **ODE Inversion in Flow-Matching models, similarly as DDIM Inversion in Diffusion models, produces latent encodings with correlations.** Reconstructions performed with FLUX.

