# OpenReview forum: "There and Back Again: On the relation between Noise and Image Inversions in Diffusion Models"
_ICLR.cc/2026/Conference — ICLR 2026 Poster_

### Official Review · Reviewer_H3z8 · 2025-10-25

**Soundness:** 3
**Presentation:** 3
**Contribution:** 3
**Rating:** 6
**Confidence:** 4

**Summary:**

This paper analyzes DDIM inversion, specifically the non-gaussian properties of the final latent $\mathbf{\hat{x}}_T$ which has been mentioned multiple times in previous literature but not explored throughoutly like the authors have done in this work. The authors show that the inverted latents quantitatively exhibit spatial correlations with the original image, especially in smooth (plain) iage regions, attributing it to the approximation errors in the early inversion steps. They propose a simple fix of replacing the first few steps with the original forward diffusion process. Their fix restores the gaussian properties of the final inverted latents, improving editability and text alignment when using DDIM inversion.

Overall, although this work has limited applicability (see weaknesses), this work is technically solid with extensive experiments that dive deep into the emprical findings that have been brushed off in previous works. Limited applicability do not outweigh its contributions in the form of empirical analyses.

**Strengths:**

- The main strenghts of this paper is in its arrangement and analysis of previous work. The authors go through great lengths in analyzing the claims of previous work that have mentioned this non-gaussian property of the final latent in DDIM inversion, and structures it well with their own quantitative experiments, corroborating past claims with evidence.
- Their quantitative experiments bring a clear insight that the early inversion steps were the root cause of the issue, emphasized more in what the authors call smooth (plain) image regions. Their proposed fix is simple and easy to implement, with convincing quantitative/qualitative improvements compared to the vanilla method of simply changing the target prompt starting from $\mathbf{\hat{x}}_T$.
- The experiments are wide breadth-wise, with multiple ablations that quantitatively show improvements.

**Weaknesses:**

Weaknesses:
- The main weakness of this work that I've found is the limited theoretical grounding of their explanation on why the early inversion steps causes correlation with the original image. Their explanation for why the early steps cause correlation is mostly empirical, and the connection they make to the ODE curvature could be formalized further. In order to prove why the proposed simple fix works should be first followed why the early inversion steps are theoretically the root cause. I do not see the simple, heuristic fix the authors propose as a weakness, just that the cause should be diven into deeper.
- The scope of their fix, especially in terms of image editing, is narrow as they demonstrate it on solely DDIM inversion. To the extent of my knowledge, most image editing tasks with text prompt inversion methods do not rely on plain DDIM inversion, but rather "image-to-noise inversion techniques" as described in section 2. Some of these inversion techniques might not be directly applicable to the proposed method, but when they are, I believe the proposed method should be demonstrated on them.

Comments:
- Though not strictly a weakness, It would be better if how the plain background or surfaces are mathematically calculated was included in the main text (even just a breif 2-3 liner) instead of fully allocating it to appendix I.

**Questions:**

- in appendix G, the authors mention: "Classifier-Free Guidance introduces additional errors to the DDIM inversion, so to focus solely on the inversion approximation error, we disable CFG by setting the guidance scale to w = 1. To the extent of my knowledge, this is the main reason multiple "image-to-noise inversion techniques" have been explored in literature. Although setting w > 1 does drastically alter the given image, I wonder if applying the proposed fix also provide a meaningful improvement in terms of fidelity or text alignment.

---

> ### Author Response · Authors · 2025-11-26
> **Response to Reviewer H3z8 (1/N)**
>
> We thank the Reviewer for their comments that helped us improve our work. Below, we answer to the mentioned weaknesses and questions.
>
> > W1: A limited theoretical grounding of their explanation on why the early inversion steps causes correlation with the original image.
>
> We appreciate the reviewer's insight regarding the theoretical underpinning of these correlations. While obtaining a closed-form proof for the specific error dynamics of DDIM inversion is analytically challenging, we can ground our empirical observations in established statistical theory. We propose the following hypothesis, which links the noise sensitivity of plain regions to their statistical properties.
>
> The core of our hypothesis is that plain regions become "out-of-distribution" for the denoising network much faster than textured regions because of the disparity in signal variance between homogenous and heterogeneous regions. In plain regions, the signal energy is so low that it is comparable to the magnitude of the inversion approximation error, making it sensitive to perturbations. This sensitivity causes plain regions to diverge significantly from the true distribution during early steps, which leads to even more severe reconstruction errors. Our method is effective because it avoids inverting these low-signal steps altogether. Instead, we inject exact Gaussian noise via forward diffusion, artificially maintaining the correct signal statistics, and skipping the problematic part of the inversion procedure.
>
> > W2: Combination with editing engines
>
> Thank you for this suggestion. During the rebuttal, we conducted additional experiments, combining our inversion method with StyleAligned [1] for the task of style transfer from reference images and with MasaCtrl [2] for the task of text-based real image editing.
>
> We evaluate style transfer by measuring generations' text alignment (with CLIP score), set consistency (pairwise cosine similarities in the dataset with DINO and CSD models), and style consistency between generations and reference image (cosine similarity with DINO and CSD). The table below presents the results of a comparison of the StyleDrop dataset. Additionally, in Appendix O.5, we qualitatively compare both inversion methods (Naive DDIM and the version with our fix) in this task.
>
> | Inversion Method | CLIP Prompt Alignment $\uparrow$ | Set Consistency (DINO) $\uparrow$ | Set Consistency (CSD) $\uparrow$ | Style Similarity (DINO) $\uparrow$ | Style Similiarity (CSD) $\uparrow$ |
> | --- | --- | --- | --- | --- | --- |
> | Naive DDIM | **0.795** | **0.476** | 0.552 | 0.505 | 0.690 |
> | Ours (forward 4%) | **0.795** | 0.471 | **0.554** | **0.510** | **0.697** |
>
> In Appendix O.4, we qualitatively present how incorporating our simple fix to the inversion process improves text-based image editing when combined with MasaCtrl across several tasks: object replacement, attribute editing, and object removal.
>
>
> [1] Hertz, Amir, et al. "Style aligned image generation via shared attention." CVPR. 2024.
>
> [2] Cao, Mingdeng, et al. "Masactrl: Tuning-free mutual self-attention control for consistent image synthesis and editing." ICCV. 2023.
>
> > Though not strictly a weakness, It would be better if how the plain background or surfaces are mathematically calculated was included in the main text (even just a breif 2-3 liner) instead of fully allocating it to appendix I.
>
> Thank you for this suggestion. We have already updated our work to include more details on this procedure in the main part.

---

> ### Author Response · Authors · 2025-11-26
> **Response to Reviewer H3z8 (2/N)**
>
> > Q1: in appendix G, the authors mention: "Classifier-Free Guidance introduces additional errors to the DDIM inversion, so to focus solely on the inversion approximation error, we disable CFG by setting the guidance scale to w = 1. To the extent of my knowledge, this is the main reason multiple "image-to-noise inversion techniques" have been explored in literature. Although setting w > 1 does drastically alter the given image, I wonder if applying the proposed fix also provide a meaningful improvement in terms of fidelity or text alignment.
>
> Thank you for this interesting question. Indeed, in our experiments, we fix the guidance scale to $w=1$ to ensure that our analysis focuses solely on the DDIM approximation error (Equation 3). We would like to highlight that previous studies have typically employed an inversion guidance scale between $1.0$ (which is the most common choice) and $3.0$, and that using $w>3$ often results in drastically worse image reconstructions (see Table 10 from [1] for reference).
>
> However, we share the Reviewer's perspective that testing our method with a higher guidance scale could potentially enhance our contribution and validate our method in the practical scenarios of text-based image editing.
>
> To this end, we tested if our simple fix improves Naive DDIM Inversion for guidance scales $w\in{1,2,3,4,5}$ applied both during inversion and reconstruction processes. The experiment was conducted using the Stable Diffusion XL text-to-image model. As presented in the Table below, for scenarios with a higher guidance scale, our simple fix, similarly to $w=1$, reduces correlations and improves editability when comparing with Naive DDIM Inversion.
>
> Additionally, we observe that our fix improves image reconstruction error (measured with LPIPS). We hypothesize that amplifying the inversion error in Naive DDIM with a higher guidance scale (i.e., $w>3$) leads to latents that are useless for image reconstruction. In such a case, replacing the first steps with random Noise may lead to more preferable reconstructions. For better comparison, in Appendix N.3 of the updated manuscript, we present examples of image reconstructions with varying guidance scale for SDXL.
>
> ### Stable Diffusion XL:
> | guidance scale $w$ | method | $LPIPS(x_0, \hat{x}_0)$ $\downarrow$ | Correlation in $\hat{x}_T$ $\downarrow$ | CLIP(edit prompt) $\uparrow$ |
> | --- | --- | --- | --- | --- |
> | 1.0 | DDIM Inv. | 0.100 | 0.166 | 0.695 |
> | 1.0 | w/ forward=4%  | 0.137 | 0.120 | 0.722 |
> | 1.0 | $\Delta$ | +0.037 | -0.046 | +0.027 |
> | --- | --- | --- | --- | --- |
> | 2.0 | DDIM Inv. | 0.199 | 0.170 | 0.779 |
> | 2.0 | w/ forward=4%  | 0.179 | 0.120 | 0.807 |
> | 2.0 | $\Delta$ | -0.020 | -0.050 |+0.028|
> | --- | --- | --- | --- | --- |
> | 3.0 | DDIM Inv. | 0.390 | 0.171 | 0.764 |
> | 3.0 | w/ forward=4%  | 0.267 | 0.121 | 0.815 |
> | 3.0 | $\Delta$ | -0.123 | -0.050 | +0.051 |
> | --- | --- | --- | --- | --- |
> | 4.0 | DDIM Inv. | 0.525 | 0.172 | 0.725 |
> | 4.0 | w/ forward=4%  | 0.372 | 0.121 | 0.800 |
> | 4.0 | $\Delta$ | -0.153 |-0.051 |+0.075 |
> | --- | --- | --- | --- | --- |
> | 5.0 | DDIM Inv. | 0.582 | 0.174 | 0.687 |
> | 5.0 | w/ forward=4%  | 0.452 | 0.121 | 0.770 |
> | 5.0 | $\Delta$ | -0.130 | -0.053 | +0.083 |
>
> [1] Ju, Xuan, et al. "Direct inversion: Boosting diffusion-based editing with 3 lines of code." ICLR. 2024.

---

### Official Review · Reviewer_Mq2S · 2025-10-27

**Soundness:** 2
**Presentation:** 3
**Contribution:** 2
**Rating:** 4
**Confidence:** 3

**Summary:**

This paper sheds light on a known phenomena—the discrepancy between a real Gaussian noise and the inversion noise obtained through DDIM inversion—and provides a simple fix to mitigate it. When performing DDIM inversion on an image, the resulting noise is generally not Gaussian. The authors point out that this is particularly true in regions of the image that are flat (i.e. textureless). Through a set of experiments, they pinpoint the reason of this discrepancy to lie in the very first few steps of inversion. To solve this, the paper proposes to replace the first few steps of inversion by a forward noising step. This is generally done for only ~4% of the denoising path.

Experiments ablate the threshold to which one should do forward noising, and validate that this fix indeed improves inversion quality for editing tasks.

**Strengths:**

The paper provides a structured analysis of the problem, which can be valuable to the community. While many of its conclusions are already well-established within the field (DDIM inversion is not a Gaussian noise etc.), the experiments presented in the methods section offer tangible evidence that reaffirms these claims. Furthermore, the paper proposes a simple solution that appears to effectively mitigate the identified issues. Overall, the presentation is clear and easy to follow.

**Weaknesses:**

My main concern with the paper lies in its contributions. The method is organized into three subsections, in which the authors address the following questions:

- Sec 3.1: Are there any difference between the original noise and the DDIM inversion noise? (Answer: Yes)
- Sec 3.2: How does it differ? (Answer: Loss of variance mostly in plain regions)
- Sec 3.3: Why? (Answer: Because it happens in the first few steps)

In my opinion, the first two sections primarily reiterate observations and facts that are already well known to the community, albeit supplemented with additional visualizations and plots that support these claims. The authors themselves mention prior work studying the same problem at the beginning of Sec 3.1. The third section has the potential to be more interesting; however, its conclusion appears to be based mainly on empirical observation, and the authors do not provide a convincing explanation for the observed phenomenon, which would have been the most compelling aspect of the analysis. Lastly, the proposed fix, while somewhat effective, appears ad hoc and lacks a principled justification.

**Questions:**

**DDIM Inversion Formula**

There appear to be different formulations of DDIM inversion in the literature. On one hand, Null-text inversion (https://arxiv.org/pdf/2211.09794, Sec 3.1) and Dhariwal & Nichol (https://arxiv.org/abs/2102.09672) both call the denoising network on $x_{t-1}$ with timestep $t-1$, i.e. $\epsilon^{(t)}\_{\theta}(x_{t}, c) \simeq \epsilon^{(t-1)}\_{\theta}(x_{t-1}, c)$. On the other hand, the authors in Eq. (3) seem to follow ReNoise (https://arxiv.org/pdf/2403.14602, Sec 3.1) and keep the same timestep but change $x_{t}$ to $x_{t-1}$, i.e. $\epsilon^{(t)}\_{\theta}(x_{t}, c) \simeq \epsilon^{(t)}\_{\theta}(x_{t-1}, c)$. In my understanding, the former formulation (calling the network on $x_{t-1}$ with its corresponding timestep $t-1$) represents the correct reverse ODE. This leads to the following questions:

- Which version exactly did the authors implement in the experiments?
- Have the authors noted any difference in the conclusions using the other version?
- In particular, this choice changes the definition of the approximation error $\xi(t)$ in Eq. (4). How does that change the results and conclusions?

Note that in Appendix E, the authors use the latter formulation (non-matching timesteps) in the text, but the illustration in Fig. 8 seems to show the former formulation (matching timesteps). I believe this could be further clarified to improve the reproducibility of the experiments.

**Image Editing and Reconstruction**

Overall, I am not very convinced by the validation provided to measure the quality of image editing applications.

- In Fig. 7, the explanation suggests the method is superior to standard DDIM inversion because it performs similarly to original Gaussian noise in both image diversity and prompt alignment. However, this comparison may not fully validate the editing task. While using real Gaussian noise $x_T$ with a target prompt will expectedly yield a high-quality, prompt-aligned image, it offers no guarantee of fidelity to the source image. Successful editing requires preserving significant portions of the original image while making targeted modifications. Therefore, performance closer to real Gaussian noise does not, by itself, validate superior editing performance.
- A similar remark applies to Table 7 and Sec 4.3: CLIP text alignment alone is insufficient to assess image editing quality, as it does not measure the preservation of unedited regions. I would suggest considering metrics specifically designed for editing, such as **directional CLIP similarity** (StyleGAN-NADA, https://arxiv.org/abs/2108.00946) or **AugCLIP** (https://arxiv.org/abs/2410.11374), for a more comprehensive assessment.
- Image editing performance often differs between generated and real images. I may have missed it, but are there any editing results with real images? If not, adding a few such examples would significantly strengthen the assessment of the method's practical utility.

**Result with Large Latent Diffusion Models**

Large text-to-image diffusion models like Stable Diffusion are very common in the field, yet not fully assessed in this paper.

- Considering the widespread use of Stable Diffusion models (SD2.1, SDXL, SD3), I was wondering why the authors did not include analysis with these models in the first part of the paper (beyond the results with SDXL in Sec. 4.3)?
- Did the authors observe any correlation between the quality of standard DDIM inversion and the size of the model's training dataset? One might expect that models trained on larger datasets are less prone to inversion errors, and that the proposed fix would have less impact in these cases. I would be happy to get the authors' perspective on this.

**Other questions and notes**

- What model was used in Figure 1, Figure 6 and Figure 13?
- The CFG scale can significantly impact experiment outcomes. The authors mention in Appendix G that they use a CFG scale $w=1$ for conditional diffusion models. Does this apply only to the image editing part, or to all experiments in the first part of the paper as well?
- In Sec 4.3, I believe it could be more explicitly mentioned that the results in Table 7 are based on a comparison with an equal number of function evaluations (NFEs).
- Appendix H shows interesting differences in the shape of the most probable triangles between models. Do the authors have an explanation for what might be causing these vastly different triangle shapes?
- In Fig. 3, perhaps a different color scheme or the addition of level set lines could help better visualize the intended point. If I understand correctly, the bottom-left corner (distance between $x_T$ and $x_T$) should be 0. However, its color appears similar to the top-right corner, which would suggest that the image latent $x_0$ is almost the same as the inverted noise $\hat{x}_T$. This seems counter-intuitive, and a clarification would be helpful.
- Appendix N shows an interesting experiment using the $L_2$ distance. In general, it is understood that the noise is not necessarily close to the image in an $L_2$ sense, but rather that they are correlated. Out of simple curiosity, does the mapping become feasible if a Pearson correlation metric is used instead?
- Lastly, I noted a minor citation issue: the reference to Samuel et al. appears to be missing the year, showing up as "Samuel et al." in the text instead of "Samuel et al. 2025".

I thank the authors for their time and look forward to any clarifications they can provide.

---

> ### Author Response · Authors · 2025-11-26
> **Response to Reviewer Mq2S (1/N)**
>
> We thank the Reviewer for their extensive remarks. Below, please find our thorough response.
>
> > Weakness: limited contribution, and ad-hoc solution
>
> Thank you for your critical assessment. We appreciate the opportunity to clarify our contributions and the reasoning behind our proposed fix.
>
> 1. Relation to the prior works:
> We clarify that while previous works have noted a general divergence between DDIM latents and Gaussian noise, prior analyses have not gone beyond stating this fact. In Section 3.1, we quantitatively confirm these observations to establish a baseline and facilitate the flow of the paper—a structural choice appreciated by Reviewer H3z8. However, Section 3.2 moves beyond existing literature. To the best of our knowledge, no prior work has spatially localized this divergence or identified that the loss of variance is concentrated specifically in plain image regions. This is a novel finding that distinguishes our contribution from a simple reiteration of known facts.
>
> 2. Regarding the proposed fix, we agree that the method is technically straightforward and not algorithmically complex. However, we view this simplicity as a feature rather than a limitation. Our primary contribution in this work is the structural analysis of the inversion error—specifically diagnosing that it stems from variance collapse in plain regions during the early steps. The proposed solution serves primarily as an experimental validation of this diagnosis: the fact that such a simple, targeted intervention successfully resolves the issue confirms that our analytical findings are correct.
>
> > Question 1: Different formulations of DDIM inversion in the literature.
>
> Thank you for this interesting question. After revisiting the literature, we see that this aspect involves some notational ambiguity in the community, given that there are works:
> - e.g., [1] (Section 3.1), [2] (Equation 4), formulating DDIM Inversion approximation as $\epsilon_\theta(z_t,t)\approx\epsilon_\theta(z_{t-1},t-1)$, and
> - e.g., [3] (Equation 3), [4] (Equation 5), [5] (Equation 5), formulating the approximation as $\epsilon_\theta(z_t,t)\approx\epsilon_\theta(z_{t-1},t)$,
> Even though all of them refer to the original DDIM/inversion works [6,7], which also follow the first version of the Equation.
>
> To resolve these discrepancies, we conducted a thorough analysis of the official codebase repositories corresponding to the cited works -- including the original ADM[6], Null-Text Inversion [1], and the standard implementation in the *diffusers* library (used in, e.g., ReNoise [3]). Our investigation reveals that while some manuscripts textually denote the conditioning timestep as $t-1$ (suggesting the first Equation), the actual implementations consistently pass the target timestep $t$ into the model (following the second Equation).
>
> Accordingly, our experiments strictly follow the "non-matching timestep" formulation described in Equation (3), as this aligns with the practical implementation found in diffusers and the underlying logic of the original ADM code, which we also believe represents the correct and reproducible approach. This choice ensures our definition of the approximation error in Equation (4)—measuring the drift between the true noise at step $t$ and the approximated noise derived from $z_{t-1}$—remains methodologically sound. We have revised Figure 8 and the corresponding text in Appendix E to explicitly match this formulation, ensuring the manuscript is free of the ambiguity found in previous literature.
>
> Additionally, we note that for discrete diffusion models, applying the first formulation ($\epsilon_\theta(z_t,t)\approx\epsilon_\theta(z_{t-1},t-1)$) presents a practical difficulty at the boundary of the process. Specifically, the first inversion step moves from the clean image $x_0$ to $x_1$. Adhering strictly to $t-1$ conditioning would require querying the model with timestep conditioning $0$ (an embedding that does not exist during training), whereas discrete models are typically trained to predict residuals starting at $t=1$. We note that continuous-time models, e.g., Flow-Matching (FMs) models, might theoretically accommodate this limitation. However, as described in [8], the second formulation ($\epsilon_\theta(z_t,t)\approx\epsilon_\theta(z_{t-1},t)$) is also used in practice for FMs, such as Flux, in ODE Inversion.

---

> > ### Author Response · Authors · 2025-11-26
> > **Response to Reviewer Mq2S (2/N)**
> >
> > > Which version exactly did the authors implement in the experiments? In particular, this choice changes the definition of the approximation error  in Eq. (4). How does that change the results and conclusions?
> >
> > We leverage implementations from the referenced papers. More precisely, in the case of ADM models (ADM-32, ADM-64, ADM-256), we use the original implementation from [6]. In our experiments with LDM, DIT, Deepfloyd IF, and SDXL models, we use the implementation of DDIM Inversion from the Diffusers library, which is used, e.g., in ReNoise [3] and Null-text inversion [1]. In our experiments, we did not observe any significant differences in results. Methods from both implementations produce highly correlated latents (see Table 2 and Table 3).
> >
> > [1] Mokady, Ron, et al. "Null-text inversion for editing real images using guided diffusion models." CVPR. 2023.
> >
> > [2] Ju, Xuan, et al. "PnP Inversion: Boosting Diffusion-based Editing with 3 Lines of Code." ICLR. 2024.
> >
> > [3] Garibi, Daniel, et al. "Renoise: Real image inversion through iterative noising." ECCV. 2024.
> >
> > [4] Wallace, Bram, Akash Gokul, and Nikhil Naik. "Edict: Exact diffusion inversion via coupled transformations." CVPR. 2023.
> >
> > [5] Samuel, Dvir, et al. "Lightning-fast image inversion and editing for text-to-image diffusion models." ICLR. 2025.
> >
> > [6] Dhariwal, Prafulla, and Alexander Nichol. "Diffusion models beat gans on image synthesis." NeurIPS 2021.
> >
> > [7] Song, Jiaming, Chenlin Meng, and Stefano Ermon. "Denoising diffusion implicit models." ICLR 2021.
> >
> > [8] Avrahami, Omri, et al. "Stable flow: Vital layers for training-free image editing." CVPR. 2025.
> >
> > > In Fig. 7, the explanation suggests the method is superior to standard DDIM inversion because it performs similarly to original Gaussian noise in both image diversity and prompt alignment. However, this comparison may not fully validate the editing task. While using real Gaussian noise
> >  with a target prompt will expectedly yield a high-quality, prompt-aligned image, it offers no guarantee of fidelity to the source image. Successful editing requires preserving significant portions of the original image while making targeted modifications. Therefore, performance closer to real Gaussian noise does not, by itself, validate superior editing performance.
> >
> > We appreciate this insightful observation. We agree with the Reviewer that successful editing requires a balance between responding to the target prompt and preserving source fidelity. However, in our submission, we adhere to the definition of the inversion process goal, which is "the initial noise vector that produces the input image when passed through the diffusion process" [1]. In this context, the original Gaussian noise serves as the theoretical ground truth; it creates the input image with zero reconstruction error (when conditioned on the source prompt) and potentially a perfectly aligned upper bound of the edition (given the target prompt), bounded only by the capabilities of the diffusion model itself.
> >
> > While we agree that successful editing requires preserving source fidelity, we argue that relying on the artifacts of DDIM inversion approximation errors to enforce that fidelity is methodologically unsound. Standard DDIM inversion often "locks" the image structure, not because it correctly captures the semantics, but because the latent space contains artificial correlations and biases. This "locking" prevents the model from performing valid modifications requested by the target prompt, as demonstrated in our experiments.
> >
> > Instead, a robust editing pipeline should start with a high-quality, decorrelated latent (being a Ground Truth noise in the ideal scenario) to ensure editability ("plasticity"), while structural preservation can be additionally handled by explicit, controllable mechanisms. The literature offers a plethora of such methods, including Prompt-to-Prompt [2], Null-text inversion [3], MasaCTRL [4], and others, which utilize techniques like attention control to preserve structure. Additionally, structural preservation of the images can be achieved by starting the generation process with a source prompt and switching to a target prompt after some steps [2].
> >
> > [1] Wallace, Bram, Akash Gokul, and Nikhil Naik. "Edict: Exact diffusion inversion via coupled transformations." CVPR. 2023.
> >
> > [2] Hertz, Amir, et al. "Prompt-to-prompt image editing with cross attention control." ICLR. 2022.
> >
> > [3] Mokady, Ron, et al. "Null-text inversion for editing real images using guided diffusion models." CVPR. 2023.
> >
> > [4] Cao, Mingdeng, et al. "Masactrl: Tuning-free mutual self-attention control for consistent image synthesis and editing." ICCV. 2023.

---

> ### Author Response · Authors · 2025-11-26
> **Response to Reviewer Mq2S (3/N)**
>
> > I would suggest considering metrics specifically designed for editing, such as directional CLIP similarity or AugCLIP, for a more comprehensive assessment.
>
> Thank you for this suggestion. The proposed metrics are much better suited for the assessment of image similarities in an editing scenario. Below, please find the results from Table 7 and Figure 7 updated with the Directional CLIP similarity metric from [1]. As visible, our fix leads to latents that enable better directional CLIP similarity than other evaluated methods.
>
> ### Evaluation of inversion methods (Table 7)
>
> | Prior                     | CLIP-T (Source)        | CLIP-T (Target)    | Directional CLIP    |
> |---------------------------|-------------------------|-------------------------|-|
> | Gaussian Noise            | 31.88 ± 11.66           | 73.34 ± 9.65            | 80.62 ± 16.96 |
> |-|-|-|-|
> | DDIM Inv.                 | 34.99 ± 11.36           | 69.58 ± 10.17           | 75.59 ± 17.86 |
> | Pix2Pix-Zero              | 34.86 ± 11.39           | 69.73 ± 10.12           | 75.83 ± 17.87 |
> | ReNoise (T=50, K=1)       | 34.89 ± 11.46           | 69.87 ± 10.07           | 76.47 +- 0.18 |
> | ReNoise (T=25, K=2)       | 35.21 ± 11.61           | 69.68 ± 10.09           | 76.17 ± 18.15 |
> | ReNoise (T=17, K=3)       | 35.79 ± 11.65           | 69.04 ± 9.98            | 75.20 ± 17.96 |
> | DPMSolver-1 (T=50)        | 34.81 ± 11.40           | 70.26 ± 10.42           | 74.76 ± 18.02 |
> | DPMSolver-2 (T=25)        | 34.69 ± 11.55           | 71.24 ± 9.91            | 76.17 ± 18.10 |
> | **Ours (forward 2%)**     | 34.32 ± 11.49           | 70.21 ± 10.13           | 76.76 ± 17.79 |
> | **Ours (forward 4%)**     | **33.62** ± 11.63       | **72.17** ± 9.94        | **78.91 ± 17.49** |
>
> ### Prompt-alignment -- Diffusion Transformer (Figure 7)
> | Prior                     | CLIP-T (Source)   $\downarrow$     | CLIP-T (Target) $\uparrow$    | Directional CLIP $\uparrow$   |
> |---------------------------|-------------------------|-------------------------|-|
> | Gaussian Noise            | $46.50_{±6.97}$          | $68.07_{±8.74}$            | $57.02_{±19.20}$ |
> |-|-|-|-|
> | DDIM Inv.                 | $48.05_{±7.15}$ | $66.20_{±8.86}$           | $54.05_{±19.71}$ |
> | **Ours (forward 4%)**     | **46.88**$_{±7.02}$| **67.72**$_{±8.70}$ | **56.45**$_{±19.18}$ |
>
> ### Prompt-alignment -- Deepfloyd IF (Figure 7)
> | Prior                     | CLIP-T (Source)   $\downarrow$     | CLIP-T (Target) $\uparrow$    | Directional CLIP $\uparrow$   |
> |---------------------------|-------------------------|-------------------------|-|
> | Gaussian Noise            | $27.30_{±10.64}$          | $64.90_{±9.30}$            | $77.60_{±17.10}$ |
> |-|-|-|-|
> | DDIM Inv.                 | $35.28_{±11.14}$ | $61.43_{±10.47}$           | $67.61_{±19.36}$ |
> | **Ours (forward 4%)**     | **28.70**$_{±10.66}$| **64.94**$_{±9.19}$ | **76.65**$_{±17.36}$ |
>
> [1] Gal, Rinon, et al. "Stylegan-nada: Clip-guided domain adaptation of image generators." ACM Transactions on Graphics (TOG) 41.4 (2022): 1-13.
>
> > Examples on real images.
>
> We thank the reviewer for this important observation regarding practical utility. Our primary experimental setup relied on generated images because evaluating the inversion mechanism requires access to the ground truth starting noise ($x_T$) to accurately quantify latent correlations and approximation errors (as analyzed in Section 3). However, we fully agree that performance on real images is essential from a practical point of view. In the first version of our submission, we have included some examples for editing real images from the ImageNet-R-TI2I dataset in Appendix O.3 (Figure 27), demonstrating that our method successfully edits real photos and enables stochastic variations, a feature not available with standard DDIM inversion. Those edits were performed using the Deepfloyd IF model.
>
> During the rebuttal, we added more examples for real images:
> - **Appendix O.2.** In Fig. 26, we present that our method successfully reconstructs real images from the StyleDrop [1] dataset.
> - **Appendix O.5.** In Figs. 27 and 28, we show that our method, when combined with StyleAligned [2], can be used to transfer style from a reference image to new generations. In case you are interested in the setup of this experiment, see our response to Reviewer W4R9 (W2).
> - **Appendix O.4.** In Figs. 29, 30, and 31, we show how our method, combined with MasaCtrl [3] editing engine, improves text-based real image editing. We present examples on three tasks from the PIE-Bench benchmark: object replacement, attribute editing, and object removal.
>
> [1] Sohn, Kihyuk, et al. "Styledrop: Text-to-image generation in any style." NeurIPS. 2023.
>
> [2] Hertz, Amir, et al. "Style aligned image generation via shared attention." CVPR. 2024.
>
> [3] Cao, Mingdeng, et al. "Masactrl: Tuning-free mutual self-attention control for consistent image synthesis and editing." ICCV. 2023.

---

> ### Author Response · Authors · 2025-11-26
> **Response to Reviewer Mq2S (4/N)**
>
> > Considering the widespread use of Stable Diffusion models (SD2.1, SDXL, SD3), I was wondering why the authors did not include analysis with these models in the first part of the paper (beyond the results with SDXL in Sec. 4.3)?
>
> We would like to highlight that we conduct our analytical experiments on a diverse set of Diffusion architectures, including pixel-space (ADMs, IF) and latent-space (LDM, DIT) models, conditional (DIT, IF) and unconditional (ADMs, LDM), trained on small (ADMs) and big (IF) datasets. We believe that our choice of models supports the generality of the findings, as appreciated by Reviewer W4R9.
>
> However, to show that our findings generalize to SOTA architectures, we report results for our correlation analysis with the SDXL model below. Additionally, given the recent popularity of the FLUX model, during rebuttal, we also conducted a similar analysis using the ODE Inversion technique (implementation from [1]). Additionally, in Appendix P, we qualitatively show that ODE Inversion latents, similar to DDIM Inversion latents, exhibit correlations.
>
> ### Models
> |                | ADM-32 | ADM-64 | ADM-256 | LDM            | DiT            | DeepFloyd-IF | SDXL | FLUX |
> |----------------|-----------|-----------|------------|----------------|----------------|--------------|-|-|
> | Diffusion      | Pixel     | Pixel     | Pixel      | Latent         | Latent         | Pixel        | Latent | Latent|
> | Image res.     | 32×32     | 64×64     | 256×256    | 256×256        | 256×256        | 64×64        | 1024×1024 | 1024×1024 |
> | Latent res.    | –         | –         | –          | 3×64×64        | 4×32×32        | –            | 4×128×128 | 16×128×128 |
> | Trainset       | CIFAR-10  | ImageNet  | ImageNet   | CelebA         | ImageNet       | LAION-A      | *Unknown* | *Unknown* |
> | Cond?          | ✗         | ✗         | ✗          | ✗              | ✓              | ✓            | ✓ | ✓ |
> | Backbone       | U-Net     | U-Net     | U-Net      | U-Net          | DiT            | U-Net        | U-Net | MM-DiT |
>
> ### Correlations in Noise, Latents and Samples
> | Model     | Noise ($x_T$)         | Latent ($\hat{x}_T$)         | Sample ($x_0$)         |
> |-----------|--------------------|--------------------|--------------------|
> | ADM-32    | 0.039 ± .003       | 0.382 ± .010       | 0.964 ± .022       |
> | ADM-64    | 0.039 ± .003       | 0.126 ± .008       | 0.925 ± .021       |
> | ADM-256   | 0.039 ± .003       | 0.161 ± .013       | 0.960 ± .008       |
> | IF        | 0.039 ± .003       | 0.498 ± .025       | 0.936 ± .019       |
> | LDM       | 0.039 ± .003       | 0.045 ± .014       | 0.645 ± .099       |
> | DiT       | 0.041 ± .003       | 0.103 ± .021       | 0.748 ± .064       |
> | SDXL       | 0.036 ± .002       | 0.155 ± .044       | 0.637 ± .064       |
> | FLUX       | 0.043 ± .002       | 0.269 ± .022       | 0.877 ± .021       |
>
> ### Inversion Error and Diversity of Latents in Plain and Non-plain Input Image Areas
> | Model      | Abs. Error (Plain) | Abs. Error (Non-plain) | Std. Dev (Plain) | Std. Dev (Non-plain) |
> |------------|---------------------|--------------------------|-------------------|------------------------|
> | Noise (def.) | --                | --                       | 1.0               | 1.0                    |
> | ADM-32     | 0.49                | 0.43                     | 0.34              | 0.46                   |
> | ADM-64     | 0.22                | 0.15                     | 0.49              | 0.64                   |
> | ADM-256    | 0.39                | 0.29                     | 0.53              | 0.66                   |
> | IF         | 0.56                | 0.40                     | 0.46              | 0.72                   |
> | LDM        | 0.13                | 0.03                     | 0.45              | 0.59                   |
> | DiT        | 0.12                | 0.06                     | 0.43              | 0.54                   |
> | SDXL        | 0.30                | 0.26                     | 0.87              | 0.96                   |
> | FLUX        | 0.31                | 0.26                     | 0.94              | 1.03                   |
>
> [1] Avrahami, Omri, et al. "Stable flow: Vital layers for training-free image editing." CVPR. 2025.

---

> ### Author Response · Authors · 2025-11-26
> **Response to Reviewer Mq2S (5/N)**
>
> > What model was used in Figure 1, Figure 6 and Figure 13?
>
> In all of those visualisations (Figure 1,6 and 13) we have used the Deepfloyd IF model. However, in the revised version of the manuscript we have added more examples from the SDXL model. See section O of Appendix.
>
> > The CFG scale can significantly impact experiment outcomes. The authors mention in Appendix G that they use a CFG scale w=1 for conditional diffusion models. Does this apply only to the image editing part, or to all experiments in the first part of the paper as well?
>
> Yes, all of the experiments were run using the classifier-guidance $w=1$. It was already noticed by prior works [1,2,3] that CFG, when applied to the inversion process, significantly reduces the reconstruction accuracy (see Table 10 and Appendix E.4 from [3] for reference). Since our goal was to focus on the DDIM approximation in our analysis solely, we disabled CFG during the inversion process.
>
> However, to present the whole picture better, we run additional experiments with higher CFG scales for the SDXL model. More precisely, we tested if our simple fix improves Naive DDIM Inversion for guidance scales $w\in{1,2,3,4,5}$ applied both during inversion and reconstruction processes. As presented in the Table below, for scenarios with a higher guidance scale, our simple fix, similarly to $w=1$, reduces correlations and improves editability when comparing with Naive DDIM Inversion.
>
> Additionally, we observe that our fix improves image reconstruction error (measured with LPIPS). We hypothesize that amplifying the inversion error in Naive DDIM with a higher guidance scale (i.e., $w>3$) leads to latents that are useless for image reconstruction. In such a case, replacing the first steps with random Noise may lead to more preferable reconstructions. For better comparison, in Appendix N.3 of the updated manuscript, we also present qualitative results of image reconstructions across varying guidance scales.
>
> | guidance scale $w$ | method | $LPIPS(x_0, \hat{x}_0)$ $\downarrow$ | Correlation in $\hat{x}_T$ $\downarrow$ | CLIP(edit prompt) $\uparrow$ |
> | --- | --- | --- | --- | --- |
> | 1.0 | DDIM Inv. | 0.100 | 0.166 | 0.695 |
> | 1.0 | w/ forward=4%  | 0.137 | 0.120 | 0.722 |
> | 1.0 | $\Delta$ | +0.037 | -0.046 |+0.027 |
> | --- | --- | --- | --- | --- |
> | 2.0 | DDIM Inv. | 0.199 | 0.170 | 0.779 |
> | 2.0 | w/ forward=4%  | 0.179 | 0.120 | 0.807 |
> | 2.0 | $\Delta$ | -0.020 | -0.050 | +0.028 |
> | --- | --- | --- | --- | --- |
> | 3.0 | DDIM Inv. | 0.390 | 0.171 | 0.764 |
> | 3.0 | w/ forward=4%  | 0.267 | 0.121 | 0.815 |
> | 3.0 | $\Delta$ | -0.123 | -0.050 | +0.051 |
> | --- | --- | --- | --- | --- |
> | 4.0 | DDIM Inv. | 0.525 | 0.172 | 0.725 |
> | 4.0 | w/ forward=4%  | 0.372 | 0.121 | 0.800 |
> | 4.0 | $\Delta$ | -0.153 | -0.051 | +0.075 |
> | --- | --- | --- | --- | --- |
> | 5.0 | DDIM Inv. | 0.582 | 0.174 | 0.687 |
> | 5.0 | w/ forward=4%  | 0.452 | 0.121 | 0.770 |
> | 5.0 | $\Delta$ | -0.130 | -0.053 | +0.083 |
>
> [1] Hertz, Amir, et al. "Prompt-to-prompt image editing with cross attention control." ICLR. 2022.
>
> [2] Mokady, Ron, et al. "Null-text inversion for editing real images using guided diffusion models." CVPR. 2023.
>
> [3] Ju, Xuan, et al. "Direct inversion: Boosting diffusion-based editing with 3 lines of code." ICLR. 2024.
>
> > Appendix H shows interesting differences in the shape of the most probable triangles between models. Do the authors have an explanation for what might be causing these vastly different triangle shapes?
>
> We believe the plots in Appendix G (Appendix H before manuscript update) provide valuable intuition into the inversion process, directly reflecting the magnitude of approximation errors. The ADM-32 model, which exhibited the highest inversion errors and pixel correlations as shown in Tables 2 and 3, produces a nearly isosceles triangle where the latent encodings share roughly equal amounts of information with the original image and the noise. On the other hand, the LDM model yielded the lowest correlations, resulting in latents that are much closer to the noise-image side of the triangle. Nevertheless, the crucial observation is that the angle next to the noise vertex remains acute across all models, indicating that even in low-error cases, the latents still encode residual information from the inverted sample.
>
> >Smaller details:
> > - It could be more explicitly mentioned that the results in Table 7 are based on a comparison with an equal number of function evaluations (NFEs).
> > - Citation issue: the reference to Samuel et al. appears to be missing the year, showing up as "Samuel et al." in the text instead of "Samuel et al. 2025".
> > - Confusion with Fig. 3.
>
> Thank you for these suggestions. We have addressed them in the revised version of the manuscript.

---

> > ### Author Response · Authors · 2025-11-26
> > **Response to Reviewer Mq2S (6/N)**
> >
> > We thank the Reviewer for the constructive feedback and extremely detailed review! In response, we have improved our work, expanding our analysis to SDXL and FLUX, incorporating Directional CLIP metrics, and adding extensive real-image editing examples. We hope these additions effectively address your concerns regarding validation and practical utility, and that you might consider raising your score.

---

### Official Review · Reviewer_Q9MZ · 2025-10-28

**Soundness:** 2
**Presentation:** 3
**Contribution:** 3
**Rating:** 4
**Confidence:** 4

**Summary:**

The authors examine DDIM inversion in diffusion models, a process of recovering an approximate noise map that reconstructs a user provided image. First, they study the statistical relation between pure normal gaussian noise, the corresponding sampled image, and the noise obtained via inversion of that same image. They show that the inverted noise maps differ from the standard gaussian distribution, having especially high neighboring pixel correlations in areas where the input image is smooth. Secondly, they claim that these statistical deviations lead to higher inversion error, and worse performance when used for of text-based editing and image interpolation. Finally, the authors claim that this error stems from low accuracy in the initial inversion timesteps and propose a simple fix of using random added noise instead, which improves the inverted noise statistics and its applicability to interpolation and editing.

**Strengths:**

1.	The authors provide a nice observation regarding the statistics of inverted noise latents compared to the native noise space, and the repercussions regarding the usage of these latents for image manipulation and editing.
2.	The authors provide a simple remedy to fix the inverted latent statistics, and as such their downstream editing potential, with minimal cost of reconstruction quality.
3.	The authors provide thorough quantitative evaluation of the tasks presented (image interpolation and text-based image editing).

**Weaknesses:**

1.	Adding random gaussian noise instead of inversion in the first steps. The authors claim that the last steps are not important for reconstruction quality, but the diffusion process acts as a coarse-to-fine spectral regressor throughout the whole process [1]. This means that the last steps should correspond to fine-grained details, such as textures. This can be observed, for example, in the results in figure 6 (top right) where the tower in the background is generated with small windows,  which did not exist in the original image.  An ablation on the effect of the added noise for different amounts of timesteps is recommended.

2.	Hard to qualitatively evaluate the results for inversion reconstruction and text-based image editing. Only a few results are present in the main paper, and the results in the supplementary section are highly compressed, while no further image files were provided in the zip. I would recommend adding more results to the paper, especially for the editing experiments, since current metrics cannot fully replace human assessment.

[1] Rissanen et al. 2023. Generative Modelling With Inverse Heat Dissipation

**Questions:**

Please see weakness section. Addressing the raised concerns could affect my final rating for the paper.

---

> ### Author Response · Authors · 2025-11-26
> **Response to Reviewer Q9MZ (1/N)**
>
> Thank you for your remarks. Below, please find our response.
> > W1: An ablation on the effect of the added noise for different amounts of timesteps is recommended.
>
> We thank the reviewer for this insightful observation regarding the coarse-to-fine nature of the diffusion process. We agree that the final steps of the generative process (which correspond to the first steps of the inversion) are responsible for resolving high-frequency details and textures. However, our claim, supported by recent works [1], is that denoising of the fine-grained features often functions as data-agnostic image quality improvement rather than semantic generation, which is why replacing them with random noise does not destroy the semantic structure of the image.
>
> We agree that our simple, straightforward fix does come with a limitation, which is a trade-off between preserving the original image information and improving the editability. However, this trade-off is highly favorable. We discuss it in Section 4 and Table 4, where we show that replacing the initial 4% of inversion steps reduces the latent's KL Divergence from 11.57 to near-ideal 0.25, while the absolute reconstruction error increases only slightly: from 0.05 to 0.07. At the same time, for the Deepfloyd IF model, we observe a crucial improvement in latent quality (KL Div drop from 608.25 to 0.48) without increasing the reconstruction error at all.
>
> Additionally, in Appendix N.1 (Table 14), you can find our analysis on how an increasing number of replaced steps impacts reconstruction quality and latent editability when a larger number of inversion steps (i.e.,$T=1000$) is used. In such a case, replacing the first two inversion steps with forward diffusion ($0.2\%$ of steps) yields higher quality latents with on-par image reconstruction error. In Appendix N.2 (Figure 22), we also qualitatively present some failure cases when image reconstruction is worse when using our fix.
>
> Finally, we agree that a more thorough exploration of this trade-off can improve our work. In the tables below, we present how the number of inverstion steps substituted with forward step, impacts the image reconstruction error (MAE, LPIPS, and SSIM metrics) and latent editability (their correlations and KL Divergence from Gaussian Noise) for: Deepfloyd IF, DiT, and Stable Diffusion XL. We observe minimal loss in performance of image reconstruction at the cost of significantly more editable latents (to the level of Gaussian Noise).
>
> ### Deepfloyd IF
> | #steps replaced (%) | MAE$(x_0, \hat{x}_0)$ $\downarrow$ | LPIPS$(x_0, \hat{x}_0)$ $\downarrow$ | SSIM$(x_0, \hat{x}_0)$ $\uparrow$ | Correlation in $\hat{x}_T$ $\downarrow$ | $KL(x_T, \hat{x}_T)$ $\downarrow$ ($\times 10^2$) |
> | --- | --- | --- | --- | --- | --- |
> | 0 (0%) | 0.073 | 0.030 | 0.878 | 0.643 | 60.449|
> | 1 (2%) | 0.069 | 0.037 | 0.854 | 0.057 | 0.934 |
> | 2 (4%) | 0.071 | 0.038 | 0.845 | 0.050 | 0.352|
> | 3 (6%) | 0.074 | 0.040 | 0.830 | 0.050 |0.346 |
> | 4 (8%) | 0.078 |0.043 |0.813 | 0.049 |0.341 |
> | 5 (10%) | 0.082 | 0.047 | 0.796 | 0.049 |0.338 |
> | 10 (20%) | 0.099 | 0.066 | 0.713 | 0.049 |0.344 |
> | 20 (40%) | 0.131 |0.113 |0.556 | 0.049 | 0.360|
> | 30 (60%) | 0.169 | 0.179 |0.394 | 0.049 |0.367 |
> | 40 (80%) | 0.233 | 0.279 |0.204 | 0.049 |0.370 |
> | 50 (100%) | 0.487 | 0.437 | 0.009 | 0.049 | 0.374 |
>
> ### Diffusion Transformer (DiT)
> | #steps replaced (%) | $MAE(x_0, \hat{x}_0)$ $\downarrow$ | $LPIPS(x_0, \hat{x}_0)$ $\downarrow$ | $SSIM(x_0, \hat{x}_0)$ $\uparrow$ | Correlation in $\hat{x}_T$ $\downarrow$ | $KL(x_T, \hat{x}_T)$ $\downarrow$ ($\times 10^2$) |
> | --- | --- | --- | --- | --- | --- |
> | 0 (0%) | 0.052 | 0.063 | 0.839 | 0.159 | 1.118 |
> | 1 (2%) | 0.070 | 0.097 | 0.741 | 0.038 | 0.011 |
> | 2 (4%) | 0.085 | 0.125 | 0.658 | 0.036 |0.010 |
> | 3 (6%) | 0.097 | 0.151 | 0.594 | 0.036 | 0.023 |
> | 4 (8%) | 0.107 | 0.173 | 0.544 | 0.037 | 0.036 | 0.035
> | 5 (10%) | 0.116 | 0.195 | 0.505 | 0.037 | 0.041|
> | 10 (20%) | 0.154 | 0.280 | 0.375 | 0.038 | 0.040|
> | 20 (40%) | 0.231 | 0.437 | 0.235| 0.037 | 0.020 |
> | 30 (60%) | 0.353 | 0.595| 0.145| 0.037 | 0.017 |
> | 40 (80%) | 0.521 | 0.710 | 0.071| 0.037 | 0.017 |
> | 50 (100%) | 0.628 | 0.750 | 0.029 | 0.037 | 0.018|

---

> ### Author Response · Authors · 2025-11-26
> **Response to Reviewer Q9MZ (2/N)**
>
> ### Stable Diffusion XL
> | #steps replaced (%) | $MAE(x_0, \hat{x}_0)$ $\downarrow$ | $LPIPS(x_0, \hat{x}_0)$ $\downarrow$ | $SSIM(x_0, \hat{x}_0)$ $\uparrow$ | Correlation in $\hat{x}_T$ $\downarrow$ | $KL(x_T, \hat{x}_T)$ $\downarrow$ |
> | --- | --- | --- | --- | --- | --- |
> | 0 (0%) | 0.027 | 0.099 | 0.814 | 0.166 | 0.008 |
> | 1 (2%) | 0.029 | 0.106 | 0.790 | 0.151 | 0.006 |
> | 2 (4%) | 0.035 | 0.137 | 0.716 | 0.120 | 0.003 |
> | 3 (6%) | 0.038 | 0.155 | 0.685 | 0.117 | 0.003 |
> | 4 (8%) | 0.041 | 0.171 | 0.663 | 0.116 | 0.003 |
> | 5 (10%) | 0.043 | 0.183 | 0.646 | 0.116 | 0.002 |
> | 10 (20%) | 0.051 | 0.230 | 0.588 | 0.115 | 0.002 |
> | 20 (40%) | 0.065 | 0.313 | 0.515 | 0.115 | 0.002 |
> | 30 (60%) | 0.082 | 0.389 | 0.460 | 0.115 | 0.002 |
> | 40 (80%) | 0.112 | 0.464 | 0.464 | 0.115 | 0.002 |
> | 50 (100%) | 0.159 | 0.541 | 0.541 | 0.115 | 0.002 |
>
> Regarding the particular example from Figure 6, please notice that the edition from the original starting Gaussian noise (which we consider a ground truth for any inversion method) also led to significant changes in the tower texture that seem to better align with the new, target prompt. In fact, by intentionally relaxing the reconstruction constraint in the first 4% of steps, we trade a negligible amount of pixel precision (ablated in Table 4) for the ability to generate necessary details that allow for better alignment with the target prompt. Furthermore, as detailed in Appendix O.3, this unlocks stochastic editing, allowing the user to explore multiple valid interpretations of a prompt - a feature impossible with deterministic inversion. We believe that this flexibility creates a more useful tool for users than a method that strictly reconstructs the original texture at the expense of the alignment with the target prompt.
>
> [1] Deja, Kamil, et al. "On analyzing generative and denoising capabilities of diffusion-based deep generative models." Advances in Neural Information Processing Systems 35 (2022): 26218-26229.
>
> > W2: I would recommend adding more results to the paper, especially for the editing experiments, since current metrics cannot fully replace human assessment.
>
> Thank you for this suggestion. In the updated manuscript, you can find more qualitative results. Precisely:
> + Appendix N.3: image reconstructions for varying value of guidance scale $w\in\{1,2,3,4,5\}$;
> + Appendix O.2: reconstructions of real images from the StyleDrop [1] dataset;
> + Appendix O.5: style transfer results when incorporating our fix for Naive DDIM Inversion with StyleAligned [2] method on StyleDrop dataset [1] (in case you are interested in the setup of this experiment, see our response to Reviewer W4R9 -- W2);
> + Appendix O.4: text-based real image editing when combining our method with MasaCtrl [3]. We present examples on three tasks from the PIE-Bench benchmark: object replacement, attribute editing, and object removal.
>
> [1] Sohn, Kihyuk, et al. "Styledrop: Text-to-image generation in any style." NeurIPS. 2023.
>
> [2] Hertz, Amir, et al. "Style aligned image generation via shared attention." CVPR. 2024.
>
> [3] Cao, Mingdeng, et al. "Masactrl: Tuning-free mutual self-attention control for consistent image synthesis and editing." ICCV. 2023.

---

### Official Review · Reviewer_W4R9 · 2025-10-31

**Soundness:** 4
**Presentation:** 4
**Contribution:** 3
**Rating:** 6
**Confidence:** 4

**Summary:**

This paper presents a systematic analysis of DDIM inversion in diffusion models, focusing on the discrepancy between the original Gaussian noise and the latent representation of DDIM inversion. The authors identify that inversion errors primarily accumulate in early steps and are particularly pronounced in plain image regions. Based on this observation, the paper proposes a simple method that replaces the first inversion steps with forward diffusion noise.

**Strengths:**

- The paper provides a thorough and insightful diagnosis of the underlying causes of latent distortions introduced during inversion.
- The experiments are conducted across a diverse set of diffusion models, supporting the generality of the findings.
- The proposed forward-diffusion replacement is conceptually simple, easy to integrate, and empirically improves both interpolation smoothness and the diversity of editing outcomes.

**Weaknesses:**

- The analysis focuses on diffusion-based models and does not provide evidence that similar problems occur in flow-matching models (e.g., FLUX, Stable Diffusion 3)
- The paper evaluates mainly interpolation and text-guided editing, but does not explore other inversion-driven applications (e.g., local editing, style transfer).

**Questions:**

- Have the authors tested whether similar inversion-induced correlation patterns appear in flow-matching models such as FLUX or Stable Diffusion 3?
- Can the proposed method enhance performance when combined with various editing engines such as MasaCtrl or Pix2Pix-Zero? Demonstrating improvements across multiple editing scenarios would further strengthen the paper.
- Tables 4 and 7 suggest that increasing the proportion of forward diffusion replacement decreases reconstruction fidelity, while improving editability. This appears to reflect an inherent trade-off between preserving the original image information and increasing the degree of manipulation. It would be valuable to analyze or formalize this trade-off explicitly.

---

> ### Author Response · Authors · 2025-11-26
> **Response to Reviewer W4R9 (1/N)**
>
> We thank the Reviewer for their comments and questions. Below, please find our response with additional experiments conducted to strengthen our work.
>
> > W1: The analysis focuses on diffusion-based models and does not provide evidence that similar problems occur in flow-matching models (e.g., FLUX, Stable Diffusion 3)
> > Q1: Have the authors tested whether similar inversion-induced correlation patterns appear in flow-matching models such as FLUX or Stable Diffusion 3?
>
> Thank you for this interesting suggestion. Indeed, the inversion procedure can also be incorporated into flow-matching models, e.g., for image editing [1,2,3].
>
> The generative process of flow-matching models can be interpreted as solving ODE numerically with the Euler method: $z_{t-1}=z_t+(\sigma_{t-1}-\sigma_t)\cdot\nu_\theta(z_t,t)$. As described, e.g., in [1] (Section 3.3), the inverse step can be expressed as $z_t=z_{t-1}+(\sigma_t-\sigma_{t-1})\cdot\nu_\theta(z_{t-1},t)$ with an assumption that $\nu_\theta(z_{t},t)\approx\nu_\theta(z_{t-1},t)$.
>
> As this approximation relies on a similar assumption as in the case of DDIM (Equation 3 in our work), we analyzed, with the original implementation from [1], if the ODE Inversion process, similarly, induces correlation patterns in latents. Below, we report image reconstruction error, editing textual alignment, and metrics validating the latents' normality with the Flux model when performing inversion with $T=50$ steps. We show that latents resulting from the ODE Inversion algorithm, similarly to the case of DDIM latents, exhibit correlations and visible deviation from the Gaussian distribution. Importantly, these deviations lead to a significant decrease in prompt alignment when starting the generation process with an editing prompt.
>
> | Prior | Absolute Reconstr. Error $\downarrow$ | LPIPS Reconstr. Error $\downarrow$ | CLIP-T (Edit Prompt) $\uparrow$ | Directional CLIP $\uparrow$ | Correlations | KL Div. to $\mathcal{N}(0;\mathcal{I})$ |
> | - | - | - | - | - | - | - |
> | Gaussian Noise ($x_T$) | 0.000 | 0.000 | 81.49 | 87.94 | 0.135 | 0.002 |
> | ODE Inv. Latent ($\hat{x}_T$) | 0.045 | 0.156 | 56.40 | 55.32 | 0.269 | 0.038 |
>
> Additionally, in the table below, we analyze the discrepancy between original noise and ODE Inversion latents for both plain and non-plain pixels in the input images. Although not as significantly visible as in DDIM latents, ODE Inversion outputs similarly tend to be more erroneous for plain image pixels and less diverse in those areas. To visualize those differences, in Appendix P of the updated manuscript, we have added qualitative examples.
>
> | Prior      | Image region      | Noise Reconstruction Error | Variance |
> |------------|-------------------|----------------------------|----------|
> | Gaussian Noise | plain         | 0.000                      | 0.981    |
> | Gaussian Noise | non-plain     | 0.000                      | 1.008    |
> | ODE Inv. Latent| plain         | 0.308                      | 0.939    |
> | ODE Inv. Latent| non-plain     | 0.261                      | 1.026    |
>
> [1] Avrahami, Omri, et al. "Stable flow: Vital layers for training-free image editing." CVPR. 2025.
>
> [2] Rout, Litu, et al. "Semantic image inversion and editing using rectified stochastic differential equations." ICLR. 2025.
>
> [3] Kulikov, Vladimir, et al. "Flowedit: Inversion-free text-based editing using pre-trained flow models." ICCV. 2025.

---

> ### Author Response · Authors · 2025-11-26
> **Response to Reviewer W4R9 (2/N)**
>
> > W2: The paper evaluates mainly interpolation and text-guided editing, but does not explore other inversion-driven applications (e.g., local editing, style transfer).
>
> We appreciate the suggestion to evaluate our fix with those additional exciting applications. To that end, we have assessed the impact of replacing the erroneous inversion steps with random noise on the task of style transfer.
>
> To that end, we follow StyleAligned [1], which proposes to employ DDIM Inversion and combine it with their attention-sharing approach in the task of Style Transfer from an Input Image. In the Table below, we evaluate the StyleAligned method when combined with Naive DDIM Inversion and with DDIM Inversion incorporating our fix. Methods are evaluated on the StyleDrop [2] dataset. We measure generations' text alignment (with CLIP [4] score), set consistency (pairwise cosine similarities in the dataset with DINO [3] and CSD [5]), and style consistency between generations and the reference image (cosine similarity with DINO and CSD). As presented in the Table, our fix consistently improves the alignment with the target style.
>
>
> | Inversion Method | CLIP Prompt Alignment $\uparrow$ | Set Consistency (DINO) $\uparrow$ | Set Consistency (CSD) $\uparrow$ | Style Similarity (DINO) $\uparrow$ | Style Similiarity (CSD) $\uparrow$ |
> | --- | --- | --- | --- | --- | --- |
> | Naive DDIM | **0.795** | **0.476** | 0.552 | 0.505 | 0.690 |
> | Ours (forward $4\%$) | **0.795** | 0.471 | **0.554** | **0.510** | **0.697** |
>
>
> Additionally, in Appendix O.2, we present reconstructions of real images from StyleDrop with our method. In Appendix O.5, we qualitatively compare Naive DDIM and our version with our fix for style transfer.
>
> [1] Hertz, Amir, et al. "Style aligned image generation via shared attention." CVPR. 2024.
>
> [2] Sohn, Kihyuk, et al. "Styledrop: Text-to-image generation in any style." NeurIPS. 2023.
>
> [3] Caron, Mathilde, et al. "Emerging properties in self-supervised vision transformers." ICCV. 2021.
>
> [4] Radford, Alec, et al. "Learning transferable visual models from natural language supervision." ICML. 2021.
>
> [5] Somepalli, Gowthami, et al. "Investigating Style Similarity in Diffusion Models." ECCV. 2024.
>
> > Q2: Can the proposed method enhance performance when combined with various editing engines such as MasaCtrl or Pix2Pix-Zero? Demonstrating improvements across multiple editing scenarios would further strengthen the paper.
>
> As demonstrated in the response above, our method, predominantly thanks to its simplicity, can be easily integrated with the StyleAligned engine in order to obtain approximate latent encoding for the reference style images.
>
> Additionally, in Appendix O.4, we show examples of image editing, when vanilla DDIM Inversion and our method are combined with the MasaCtrl editing engine. We present examples across three tasks from PIEBench for text-based real image editing: object replacement, attribute editing, and object removal. We observe that replacing the first $4\%$ of DDIM Inversion steps with forward diffusion leads to more successful edits in prompt adherence, while not observing degradation in consistency to input images.

---

> ### Author Response · Authors · 2025-11-26
> **Response to Reviewer W4R9 (3/N)**
>
> > Q3: Tables 4 and 7 suggest that increasing the proportion of forward diffusion replacement decreases reconstruction fidelity, while improving editability. This appears to reflect an inherent trade-off between preserving the original image information and increasing the degree of manipulation. It would be valuable to analyze or formalize this trade-off explicitly.
>
> We fully agree that the proposed simple solution introduces a trade-off between preserving the original image information and improving the editability. However, this trade-off is highly favorable. We discuss it in section 4 and Table 4, where we show that replacing the initial 4% of inversion steps reduces the latent's KL Divergence from 11.57 to near-ideal 0.25, while the absolute reconstruction error increases only slightly: from 0.05 to 0.07. At the same time, for the Deepfloyd IF model, we observe a crucial improvement in latent quality (KL Div drop from 608.25 to 0.48) without increasing the reconstruction error at all. To further formalize this trade-off, we have highlighted it in the revised version of the manuscript.
>
> Additionally, in Appendix N.1 (Table 14), you can find our analysis on how an increasing number of replaced steps impacts reconstruction quality and latent editability when a larger number of inversion steps (i.e.,$T=1000$) is used. In such a case, replacing the first two inversion steps with forward diffusion ($0.2\%$ of steps) yields higher quality latents with on-par image reconstruction error. In Appendix N.2 (Figure 22), we also qualitatively present some failure cases when image reconstruction is worse when using our fix.
>
> Finally, we agree that a more thorough exploration of this trade-off can improve our work. In the tables below, we present how the number of inverstion steps substituted with forward step, impacts the image reconstruction error (MAE, LPIPS, and SSIM metrics) and latent editability (their correlations and KL Divergence from Gaussian Noise) for: Deepfloyd IF, DiT, and Stable Diffusion XL.
>
> ### Deepfloyd IF
> | #steps replaced (%) | MAE$(x_0, \hat{x}_0)$ $\downarrow$ | LPIPS$(x_0, \hat{x}_0)$ $\downarrow$ | SSIM$(x_0, \hat{x}_0)$ $\uparrow$ | Correlation in $\hat{x}_T$ $\downarrow$ | $KL(x_T, \hat{x}_T)$ $\downarrow$ ($\times 10^2$) |
> | --- | --- | --- | --- | --- | --- |
> | 0 (0%) | 0.073 | 0.030 | 0.878 | 0.643 | 60.449|
> | 1 (2%) | 0.069 | 0.037 | 0.854 | 0.057 | 0.934 |
> | 2 (4%) | 0.071 | 0.038 | 0.845 | 0.050 | 0.352|
> | 3 (6%) | 0.074 | 0.040 | 0.830 | 0.050 |0.346 |
> | 4 (8%) | 0.078 |0.043 |0.813 | 0.049 |0.341 |
> | 5 (10%) | 0.082 | 0.047 | 0.796 | 0.049 |0.338 |
> | 10 (20%) | 0.099 | 0.066 | 0.713 | 0.049 |0.344 |
> | 20 (40%) | 0.131 |0.113 |0.556 | 0.049 | 0.360|
> | 30 (60%) | 0.169 | 0.179 |0.394 | 0.049 |0.367 |
> | 40 (80%) | 0.233 | 0.279 |0.204 | 0.049 |0.370 |
> | 50 (100%) | 0.487 | 0.437 | 0.009 | 0.049 | 0.374 |
>
> ### Diffusion Transformer (DiT)
> | #steps replaced (%) | $MAE(x_0, \hat{x}_0)$ $\downarrow$ | $LPIPS(x_0, \hat{x}_0)$ $\downarrow$ | $SSIM(x_0, \hat{x}_0)$ $\uparrow$ | Correlation in $\hat{x}_T$ $\downarrow$ | $KL(x_T, \hat{x}_T)$ $\downarrow$ ($\times 10^2$) |
> | --- | --- | --- | --- | --- | --- |
> | 0 (0%) | 0.052 | 0.063 | 0.839 | 0.159 | 1.118 |
> | 1 (2%) | 0.070 | 0.097 | 0.741 | 0.038 | 0.011 |
> | 2 (4%) | 0.085 | 0.125 | 0.658 | 0.036 |0.010 |
> | 3 (6%) | 0.097 | 0.151 | 0.594 | 0.036 | 0.023 |
> | 4 (8%) | 0.107 | 0.173 | 0.544 | 0.037 | 0.036 | 0.035
> | 5 (10%) | 0.116 | 0.195 | 0.505 | 0.037 | 0.041|
> | 10 (20%) | 0.154 | 0.280 | 0.375 | 0.038 | 0.040|
> | 20 (40%) | 0.231 | 0.437 | 0.235| 0.037 | 0.020 |
> | 30 (60%) | 0.353 | 0.595| 0.145| 0.037 | 0.017 |
> | 40 (80%) | 0.521 | 0.710 | 0.071| 0.037 | 0.017 |
> | 50 (100%) | 0.628 | 0.750 | 0.029 | 0.037 | 0.018|
>
> ### Stable Diffusion XL
> | #steps replaced (%) | $MAE(x_0, \hat{x}_0)$ $\downarrow$ | $LPIPS(x_0, \hat{x}_0)$ $\downarrow$ | $SSIM(x_0, \hat{x}_0)$ $\uparrow$ | Correlation in $\hat{x}_T$ $\downarrow$ | $KL(x_T, \hat{x}_T)$ $\downarrow$ |
> | --- | --- | --- | --- | --- | --- |
> | 0 (0%) | 0.027 | 0.099 | 0.814 | 0.166 | 0.008 |
> | 1 (2%) | 0.029 | 0.106 | 0.790 | 0.151 | 0.006 |
> | 2 (4%) | 0.035 | 0.137 | 0.716 | 0.120 | 0.003 |
> | 3 (6%) | 0.038 | 0.155 | 0.685 | 0.117 | 0.003 |
> | 4 (8%) | 0.041 | 0.171 | 0.663 | 0.116 | 0.003 |
> | 5 (10%) | 0.043 | 0.183 | 0.646 | 0.116 | 0.002 |
> | 10 (20%) | 0.051 | 0.230 | 0.588 | 0.115 | 0.002 |
> | 20 (40%) | 0.065 | 0.313 | 0.515 | 0.115 | 0.002 |
> | 30 (60%) | 0.082 | 0.389 | 0.460 | 0.115 | 0.002 |
> | 40 (80%) | 0.112 | 0.464 | 0.464 | 0.115 | 0.002 |
> | 50 (100%) | 0.159 | 0.541 | 0.541 | 0.115 | 0.002 |
>
>
> We appreciate your feedback, which motivated us to demonstrate the generalizability of our method to flow-matching models. We hope these new results address your questions and justify increasing the score.

---

> ### Comment · Reviewer_W4R9 · 2025-11-28
>
> Thank you for your detailed response. The new experiments adequately address the concerns I raised, and I appreciate the effort you put into clarifying the contribution. I will keep my original score.

---

### Author Response · Authors · 2025-12-03
**General response from authors**

## Message to Area Chair

Dear Area Chair,

We understand you have a heavy workload with many papers to assess, particularly given the recent disruptions to the platform. We are writing to provide a concise summary of our rebuttal, as the system blockade unfortunately prevented a discussion phase with 3 of our 4 reviewers.

We dedicated the rebuttal period to a tremendous experimental effort, prioritizing computationally intensive requests (e.g., analysis on new architectures like FLUX and SDXL, evaluation on new tasks like Style Transfer) to thoroughly satisfy the reviewers' curiosity. Regrettably, due to the system freeze, only one reviewer could respond before the cutoff.

We respectfully ask you to consider the following points in your decision:

1. **Borderline reject reviews.** Both Reviewers Q9MZ and Mq2S gave "Borderline" ratings but expressed a great interest in the work and willingness to accept it if specific inquiries were resolved. Since they could not respond due to the blockade, we ask you to verify that we have met their conditions:

    - Reviewer Q9MZ (Score: 4) explicitly stated: "Addressing the raised concerns could affect my final rating". They asked for an ablation on the trade-off between reconstruction fidelity and editability and additional qualitative results. To this end:

        - We added the requested ablation on the trade-off between reconstruction and editability as presented in the tables in the comment and in the Appendix N.2. We demonstrated that replacing just the first 4% of inversion steps with random noise reduces the latent's KL divergence from 11.57 to a near-ideal 0.25 (a massive gain in editability) while increasing the absolute reconstruction error only negligibly (from 0.05 to 0.07). This proves the trade-off is highly favorable.
        - We also included extensive qualitative results (Appendix O), including real image reconstruction, editing, new task of style transfer and additional experiments (Appendix N.3) showing that our fix also improves inversion with larger guidance scales.

    - Reviewer Mq2S posed a significant number of curiosity-driven questions regarding the method's generalizability and practical utility. We addressed this extensive list in full:
        - **Additional architectures/paradigms.** On top of our original experimentation with 6 diffusion models (including LDMs, DiTs and DeepFloyd models), we have added more results with SOTA generative models such as Stable Diffusion XL and FLUX (Flow Matching), which confirm that our findings spans also to the similar formulations, and even larger architectures.
        - **Real image editions.** We have added comprehensive evaluations showing how our simple fix can be combined with SOTA methods for real image editions or style transfer for which we used StyleDrop and PIE-Bench datasets. The results presented in Section 4.4 and Appendices O.4 and O.5 of the updated submission confirm that our solution further improves their performance.
        - **New Metrics.** Reviewer Mq25 suggested Directional CLIP for better editing assessment. We implemented this metric, which quantitatively confirmed our method outperforms standard inversion.

2. Reviewer W4R9 (Score: 6) was the only reviewer able to respond before the system locked. They confirmed: "The new experiments adequately address the concerns I raised... I appreciate the effort". This signals that our new experimental data is robust and convincing.


**Conclusion.** Our work contributes a rigorous empirical diagnosis of a fundamental issue in diffusion inversion, latent correlations in smooth regions, and proposes a simple, effective fix to resolve it. We have fully satisfied the explicit condition set by Reviewer Q9MZ to raise their score and resolved the extensive inquiries of Reviewer Mq2S with significant new experimental evidence on SOTA architectures (FLUX/SDXL). We hope you can assess our rebuttal in their absence and see that the paper is now well above the acceptance threshold.

Thank you for your time and efforts in these difficult circumstances. Please note that **the updated PDF now includes additional experiments and clarifications motivated by Reviewers comments**.

## Message to Reviewers
We are grateful for the time you dedicated to assessing our work. We deeply regret that platform issues prevented a proper discussion, as your inquiries were extremely insightful and we were eager to engage further (e.g., on the notation ambiguity in related works raised by Mq2S). Your constructive feedback has been invaluable, as it directly strengthened our submission and the updated PDF now includes additional experiments and clarifications motivated by your comments. Finally, we wish to explicitly reassure you that we did not exploit the system bug to discover your identities, and you remain strictly anonymous to us.

---

### Meta-Review · Area_Chair_WPho · 2026-01-09

**Summary:**

This paper focuses on DDIM inversion in diffusion models, addressing the known issue of latent representations deviating from Gaussian noise. Its main contributions are: (a) the observation that this deviation arises from the initial inversion steps (corresponding to the final steps of the forward noising process), and (b) a simple heuristic to mitigate it by replacing a small number of initial DDIM inversion steps with forward noising steps. The authors provide a series of experiments that empirically validate these claims.
Reviewers recognise several strengths in the paper. Namely, they observe that it is easy-to-follow, making interesting observations, and contains  an extensive experimental section that significantly improved during the rebuttal phase. The paper builds on a known empirical observation proposing a simple trick to address the issue. As noted by reviewers Mq2S and H3z8, a convincing theoretical explanation of the observed phenomenon could offer deeper insights and enable a more principled derivation of the proposed fix, or even alternative solutions. However, I believe that current version of the paper makes sufficient contributions to the field and provides interesting insights. Therefore, I recommend its acceptance at ICLR.

**Reviewer Concerns:**

The reviewers expressed concerns about the empirical validation of the proposed approach, which the authors largely addressed in their rebuttal by adding further experiments. Specifically, they included evaluations on Stable Diffusion XL and FLUX, incorporated different editing approaches and inversion-driven applications, and added experiments using additional metrics, thereby addressing reviewer Mq25’s comments. Finally, the authors partly addressed the concerns of reviewers Mq2S and H3z8 regarding the lack of a solid theoretical justification for the observed phenomenon and a principled derivation of the proposed fix.

**Reviewer Scores:**

Reviewer W4R9 initially assigned a score of 6 and expressed their intention to maintain it following the rebuttal. Reviewers Q9MZ and Mq2S both gave scores of 4. Reviewer Q9MZ has been inclined to raise their score to 6. The authors made provided extensive responses to reviewers' Mq2S comments, and it is likely that they would increase their score. Finally, reviewer's H3z8 score was 4. The reviewer  highlighted issues with the theoretical grounding of the method, concerns that were partly addressed during the rebuttal, providing some motivation behind their proposed approach.  That being said, most likely they would either increase their score or  keep it unchanged.

---

### Decision · Program_Chairs · 2026-01-26

Accept (Poster)